# Importance of different parameterization changes for the updated dust cycle modelling in the Community Atmosphere Model (version 6.1)

Longlei Li[1], Natalie M Mahowald[1], Jasper F Kok[2], Xiaohong Liu[3], Mingxuan Wu[4], Danny M Leung[2], Douglas S Hamilton[1], Louisa K Emmons[5] Yue Huang[2,7,8], Jun Meng[2], Neil Sexton[1], and Jessica Wan[6]

[1]Department of Earth and Atmospheric Sciences, Cornell University, Ithaca, NY, United States
[2]Department of Atmospheric and Oceanic Sciences, University of California, Los Angeles, CA, United States
[3]Department of Atmospheric Sciences, Texas A&M University, College Station, TX, United States
[4]Atmospheric Sciences and Global Change Division, Pacific Northwest National Laboratory, Richland, WA, United States
[5]Atmospheric Chemistry Observations and Modeling Laboratory, National Center for Atmospheric Research, Boulder, CO, United States
[6]Scripps Institution of Oceanography, University of California San Diego, La Jolla, CA, USA
[7]Earth Institute, Columbia University, New York, NY 10025, USA
[8]NASA Goddard Institute for Space Studies, New York, NY 10025, USA

*Correspondence to*: Longlei Li (ll859@cornell.edu)

**Abstract.** The Community Atmosphere Model (CAM6.1), the atmospheric component of the Community Earth System Model (CESM; version 2.1), simulates the lifecycle (emission, transport, and deposition) of mineral dust and its interactions with physio-chemical components to quantify the impacts of dust on climate and the Earth system. The accuracy of such quantifications relies on how well dust-related processes are represented in the model. Here we update the parameterizations for the dust module, including those on the dust emission scheme, the aerosol dry deposition scheme, the size distribution of transported dust, and the treatment of dust particle shape. Multiple simulations were undertaken to evaluate the model performance against diverse observations, and to understand how each update alters the modeled dust cycle and the simulated dust direct radiative effect. The model-observation comparisons suggest that substantially improved model representations of the dust cycle are achieved primarily through the new more physically-based dust emission scheme. In comparison, the other modifications induced small changes to the modeled dust cycle and model-observation comparisons, except the size distribution of dust in the coarse mode, which can be even more influential than that of replacing the dust emission scheme. We highlight which changes introduced here are important for which regions, shedding light on further dust model developments required for more accurately estimating interactions between dust and climate.

## 1 Introduction

Mineral dust accounts for most aerosol mass in the Earth's atmosphere and plays an important role in different aspects of the coupled Earth-Human-Climate system. For example, dust modifies the radiative budget and atmospheric dynamics via direct, semi-direct, and indirect interactions with radiation (Sokolik and Toon, 1996; Miller and Tegen, 1999; Pérez et al., 2006; Li and Sokolik, 2018a) and clouds (DeMott et al., 2003; Rosenfeld et al., 2001; Shi and Liu, 2019). In addition, the deposition of mineral dust perturbs the energy budget by darkening snow and glacial ice sheets directly due to the relatively darker color of dust particles (Skiles et al., 2018; Sarangi et al., 2020) and indirectly by providing nutrients

(e.g., phosphorus) to snow algae (Mccutcheon et al., 2021). Dust deposited onto land and ocean can also affect the biogeochemistry by adding nutrients (iron and phosphorus) and/or pollutants to ecosystems (Martin et al., 1990; Swap et al., 1992; Shinn et al., 2000; Tie and Cao, 2009; Mahowald, 2011; Mahowald et al., 2017, 2010; Hamilton et al., 2020).

To quantify the climate and biogeochemical impacts of dust, accurately reproducing the dust cycle (e.g., emission, transport, deposition, etc.) with models is required. However, previous studies have shown substantial differences between the modeled dust cycle and observations (e.g., surface dust concentration, and dust deposition) (Albani et al., 2014; Wu et al., 2020a). These uncertainties in the dust cycle modeling, as well as uncertainties in optical properties due primarily to dust size and mineral composition suggest a large uncertainty in estimating the dust direct radiative effect (Kok et al., 2017; Li et al, 2021).

The difficulty in modeling dust results primarily from a limited understanding of the processes that control the emission, aging, and removal of dust during transport (Sokolik et al., 2001). Past studies have documented a nonlinear response of dust emission to the soil surface state and meteorological fields (Kok et al., 2012), strong regional variation of the erodible soil composition (Claquin et al., 1999; Journet et al., 2014), complex chemical and physical aging of dust during transport (Cwiertny et al., 2008; Usher et al., 2003) at varied time and spatial scales, a wide range of dust particle size (Mahowald et al., 2014), and irregular shape of dust aerosol particles (Reid et al., 2003a; Wang et al., 2015). These complexities impose a great challenge to parameterizing dust-related processes (e.g., dust emissions and dust deposition) and thus to accurately simulating the dust cycle in climate models. In addition, in situ or station-based measurements of dust aerosols are highly limited at both temporal and spatial scales, which makes representation of those measurements challenging, especially considering the episodic character of dust events (Mahowald et al., 2009). As such, the modeling community is still moving toward better parametrizing the different phases of the dust cycle.


To account for regional variations in dust composition and the resultant dust optical properties in estimating the dust direct radiative effect, several common and radiatively important minerals found in dust from major dust sources were introduced to the Community Atmosphere Model versions 4 (CAM4) and 5 (CAM5) (Scanza et al., 2015) and migrated to CAM6.1 (Li et al., 2021), which are the atmosphere components of the Community Earth System Model (CESM: version 1 and 2, respectively). Including the ability to resolve dust speciation along with the addition of an atmospheric iron cycle module (Scanza et al., 2018; Hamilton et al., 2019) facilitates the study of dust impacts on biogeochemical cycles (Hamilton et al., 2020).



As one of the widely used climate models, the Community Atmosphere Model (CAM) contains several weaknesses in modeling the dust cycle. For example,

1) the default scheme in CAM6.1 (Zender et al., 2003; Dust Entrainment And Deposition DEAD model, referred as DEAD) relies on an empirical geomorphic dust source function, created based on satellite retrievals of dust source regions, to model dust emissions;


2) the current default CESM2.1 is using the dry deposition scheme Zhang et al. (2001; Z01 hereafter) developed for particle deposition over smooth and non-vegetated surfaces. This scheme, however, underemphasizes the interception loss, the mechanism of which is less influential over the other surfaces such as grassland. The use of the Z01 in the


current default CESM2.1 is, thus, very likely overestimating the dry deposition velocity of fine-sized aerosols (diameter < 1.0 μm; referring to the geometric diameter herein unless stated otherwise) and slightly underestimating that of aerosols with diameter > 5.0μm (Wu et al., 2018), especially over non-vegetated surfaces (Petroff and Zhang et al., 85    2010);

3) one of the changes from CAM5 to CAM6.1 was replacing the size distribution of aerosols in the coarse mode in CAM5 with the one that has a much narrower width in CAM6.1 (Table 1). This change was to accommodate stratospheric aerosols in the coarse mode (e.g., volcanic sulfate) compared to an early officially released version of this 90    model (Mills et al., 2016). A recent model evaluation against satellite retrievals (Wu et al., 2020b) suggest that CESM2.1-CAM6.1 worsened the dust cycle representation and stands out in simulating the relative importance of wet to dry deposition, compared with the other global climate models or model versions, such as CESM1-CAM5, due partially to the narrow coarse geometric standard deviation;

4) dust aerosol are typically aspherical particles in shape. The dust asphericity could lengthen the dust lifetime by ~20% compared to modeling dust as spherical particles (Huang et al., 2020). Still, CAM6.1 simulates dust as spherical particles, though the impact of dust asphericity on optical depth and resulting direct radiative effect of dust (Kok et al., 2017) has been previously introduced to CAM6.1 (Li et al., 2021).

Correspondingly, this paper describes several updates to the dust representation in CAM6.1 on the four aspects and evaluates whether and for what conditions they improve the dust model comparison to observations in the present climate. Specifically, we

1) replace DEAD with a new more physically based dust emission scheme, Kok et al., (2014a; referred as BRIFT) 105    previously developed for the climate models within the framework of DEAD. This scheme performs well against observations in CESM-CAM4 (Kok et al., 2014b) without the aid of the empirical geomorphic dust source function;

2) replace Z01 with the dry deposition scheme developed by Petroff and Zhang et al., (2010) (PZ10 hereafter) to mediate the overestimation of the dry deposition velocity of fine-sized aerosols;

3) revert size distribution of dust aerosol particles in the coarse mode to the one previously employed in CAM5;

4) account for the lifetime effect of dust asphericity by decreasing the modeled gravitational settling velocity.

These updates are based on up-to-date knowledge of the dust cycle and are thus more physically realistic than the default dust parameterizations in CAM6.1/Community Land Model (version 5; CLM5).

**Table 1.** Mode parameters for the Modal Aerosol Module version 4 (MAM4) used in CAM5 (CAM5 size) and CAM6.1 (CAM6 size) by default: geometric standard deviations (σ) and initialized geometric mean diameter (GMD) and its ranges. Values in parentheses if 120    present are for CAM6.1 cells without parentheses are kept the same between CAM5 and CAM6.1.

| Mode (note order) | σ | Initialized | Lower bound GMD (μm) | Upper bound GMD (μm) |
| --- | --- | --- | --- | --- |

| | GMD (μm) | | | |
|---|---|---|---|---|
| Primary carbon (a4) | 1.6 | 0.050 | 0.010 | 0.10 |
| Aitken (a2) | 1.6 | 0.026 | 0.0087 | 0.052 |
| Accumulation (a1) | 1.8(1.6) | 0.11 | 0.054 | 0.44 |
| Coarse (a3) | 1.8(1.2) | 2.0(0.90) | 1.0(0.40) | 4.0(40) |

We organize the paper as follows: Sect. 2 describes the model (Sect. 2.1- 2.4), the modifications we made to the model (Sect. 2.5), and the experiment we conducted (Sect. 2.6) under present climate conditions to achieve our purpose. Section 3 presents the observation and semi-observation for model evaluation in current climate. Section 4 describes metrics used to assess the model performance. Section 5 evaluates the performance of the updated model by comparing simulated dust properties (e.g., surface dust concentrations, deposition fluxes, vertical distribution, and size distribution of transported dust) against measurements, retrievals, and model-observation integration (Sect. 5.1), quantifies the influence of each modification on those simulated dust properties (Sect. 5.2), documents the influence of those modifications on the estimate of the dust direct radiative effect (Sect. 5.3), and compares these changes in order to recommend which are the most important for other models to consider (Sect. 5.4). Section 6 shows the difference between the bulk- and speciated-dust models on the dust cycle modeling and the resultant dust climatic effects. Furthermore, we discuss limitations in the model-observation comparison in Sect. 7, and discussions and conclusions in Sect. 8.

## 2 Model descriptions

We used CAM6.1 (Sect. 2.1), embedded within the National Center for Atmospheric Research CESM2.1, to simulate the dust cycle in all the numerical experiments. This section describes bulk- (Sect. 2.2) and speciated-dust model (Sect. 2.3), dust optical properties and radiation flux diagnostics in CAM6.1 (Sect. 2.4), and our modifications to the base code (Sect. 2.5): the new dust emission scheme and change to the aerosol dry deposition and optics to include dust asphericity. Two sets of simulations with offline dynamics were conducted (Table 2; Sect. 2.6) using bulk (no composition distinguished between particles) and speciated dust. A total of nine experiments were conducted to evaluate the performance of each development that a future version of official model release would likely include on reproducing the dust cycle against that of the current schemes and observations. Five out of the nine experiments quantify how the size treatment for transported dust affects the dust cycle modeling. We do not evaluate the model performance on simulating the dust cycle in the preindustrial considering the scarcity of measurements relative to the current climate (Mahowald et al., 2010).

**Table 2.** Simulations performed in this study for years 2006-2011. Treatment of dust tracer: speciated dust with separate tracers (MINE: mineralogy), or no dust speciation (bulk); the dust emission scheme: Zender et al., (2003a; DEAD) or Kok et al., (2014a; BRIFT); with or without accounting for the lifetime effect of dust asphericity (Asp versus Sph); dry deposition scheme: Zhang et al., (2001; Z01) or Petroff and Zhang (2010; PZ10); parameters for size distribution taken from the released version of CAM5 and CAM6.1 (see Table 1 for CAM5 and CAM6 size, respectively); additional test on dust size distribution using the coarse-mode σ=1.2 from the released version of CAM6.1 and the rest parameters (e.g., boundaries of the geometric mean diameter) from the released version of CAM5; meteorology field nudged toward reanalysis data (offline) for 2000s climate; dust tuning parameter includes the CAM namelist variable (dust_emis_fact) and b used in the calculation of the threshold gravimetric water content (see Sect. 2.5.1). The variable $f_{clay}$ denotes the clay fraction in CLM5. CAM6.1 and CAM6.α in bold refer to the default model and proposed new model versions, respectively, with bulk dust. Note negligible influence on the dust cycle modeling and corresponding DRE by changing the size parameters of the accumulation mode between CAM5 and CAM6 size.

| Exp. | Case names | Dust model | Dry dep. | Lifetime effect of dust asphericity | Emi. scheme | Dust size distribution | Dust tuning parameters (dust_emis_fact; b) | Comments |
|---|---|---|---|---|---|---|---|---|
| 01 | **CAM6.1** | Bulk | Z01 | No (Sph) | Zender [2003a] | Default CAM6 size (Table 1) | 0.91; $1/f_{clay}$ | Officially released version |
| 02 | NEW_EMIS | Bulk | Z01 | No (Sph) | Kok [2014a] | Default CAM6 size (Table 1) | 28; $1/f_{clay}$ | Control for size tests |
| 03 | NEW_EMIS_SIZE | Bulk | Z01 | No (Sph) | Kok [2014a] | Default CAM5 size (Table 1) | 28; $1/f_{clay}$ | Changing the coarse-mode size distribution; influence quantified by comparing this with Exp. 02 |
| 04 | NEW_EMIS_SIZE_WIDTH | Bulk | Z01 | No (Sph) | Kok [2014a] | Default CAM6 size but with width of the coarse-mode size distribution from defaulted CAM5 size | 28; $1/f_{clay}$ | No change to size parameters for the other modes; influence quantified by comparing this with Exp. 02 |
| 05 | **CAM6.α** | Bulk | PZ10 | Yes (Asp) | Kok [2014] | Default CAM5 size | 3.6; 1.0 | New bulk dust model |
| 06 | MINE_BASE | Mineralogy | Z01 | No (Sph) | Zender [2003a] | Default CAM5 size | 1.6; $1.0/f_{clay}$ | Baseline for quantifying the impact of each modification |
| 07 | MINE_NEW_EMIS | Mineralogy | Z01 | No (Sph) | Kok [2014a] | Default CAM5 size | 3.6; 1.0 | Changing the dust emission scheme: influence quantified by comparing this with Exp. 06 |
| 08 | MINE_NEW_EMIS_SHAPE | Mineralogy | Z01 | Yes (Asp) | Kok [2014a] | Default CAM5 size | 3.6; 1.0 | Experiment for changing the dust emission and shape |
| 09 | CAM6.α _MINE | Mineralogy | PZ10 | Yes (Asp) | Kok [2014a] | Default CAM5 size | 3.6; 1.0 | New mineralogy dust model: combined influence of the new emission scheme, PZ10, and dust asphericity quantified by comparing this with Exp. 02 |

## 2.1 Aerosol representation

We use the Modal Aerosol Model version 4 (MAM4) in the CESM2.1-CAM6.1 (Liu et al., 2016). We consider both the default DEAD dust emission scheme (Zender et al., 2003) in the current officially released version of CAM6.1 model as well as that of Kok et al., (2014a) (Sect. 2.5.1).

CAM6.1 simulates the advection, deposition, and aerosol microphysics (e.g., coagulation and nucleation) during transport via Module Aerosol Model (version 4: MAM4) using four log-normal size modes (Liu et al., 2016): accumulation (containing sulfate, secondary organic matter, primary organic matter, black carbon, sea salt, and soil dust), Aitken (containing dust, sulfate, sea salt, and secondary organic matter), coarse (containing dust, sea salt, and sulfate), and a primary carbonaceous mode (primary organic matter and black carbon). Within each mode, aerosol tracers are transported as an internal mixture of the species present, while aerosol species from different modes are externally mixed. Also advected in each of the four modes is the number concentration of aerosol particles (Liu et al., 2016), allowing an effective radius to be calculated and the effect of aerosol-cloud interactions to be diagnosed. The removal of dust aerosols is mainly through dry deposition and wet deposition, including in- and below-cloud processes, as detailed in Neale et al. (2010). In the formation of precipitating clouds, dust particles can serve as cloud condensation nuclei (CCN) and/or ice nucleating particles (INPs) and thus can be removed via nucleation scavenging (Zender et al., 2003). In addition, the model accounts for the in-cloud scavenging of dust in the Aitken mode by Brownian diffusion, but neglects

the other scavenging processes (Easter et al., 2004), which are relatively slow (Pruppacher and Klett, 1997), such as thermophoresis. Below the cloud, dust particles can be removed by the sub-cloud scavenging. This sub-cloud scavenging of dust aerosols follows a first order loss as the product of the precipitation flux, dust mass mixing ratio, and the scavenging efficiency (Dana and Hales, 1976), for example. The wet deposition rate thus depends on the hygroscopicity of dust (=0.068; Scanza et al., 2015) as CCN/INPs and the prescribed scavenging coefficient (Neale et al., 2010), both of which are currently constant with respect to the dust size (and composition for speciated dust) in CAM6.1. This size independency of the scavenging coefficient may be an oversimplification, since measurements suggest that it can vary intensively on an order or two even within a size mode (Wang et al., 1978).

The geometric standard deviation (σ) of each mode is prescribed and default values for CAM5 and CAM6.1 are given in Table 1, along with the initialized geometric mean diameter (GMD), based on which the model predicates the GMD online, and its ranges. Note that the current default CAM6.1 employs a narrow coarse-mode size distribution but a broad boundary width (high bound minus low bound), likely resulting in the GMD bounds less in effect, compared to that in CAM5. The narrower set of the coarse-mode size distribution was designed to accommodate for stratospheric aerosols (e.g., volcanic sulfate) (Mills et al., 2016), but was not previously compared to dust aerosol observations in detail.

## 2.2 Bulk dust modeling

Parameterization of the default dust emissions in DEAD generally follows the dust mobilization mechanism developed by Marticorena and Bergametti (1995) (referred as DEAD hereafter as well). As a component of CESM2, the CLM initiates dust entrainment once the near-surface friction velocity exceeds the soil threshold friction velocity, which primarily depends on the physical characteristics of the soil (e.g., soil moisture content, and grain size distribution) and land cover (Kok et al., 2012; Shao, 2008). The downwind transfer of wind momentum to the surface soil to produce dust emissions is assumed to be completely prevented by vegetation when the leaf area index (LAI) exceeds a threshold value, 0.3 $m^2$ $m^{-2}$ (Mahowald et al., 2006a). Below the threshold value, the fraction of a grid cell capable of releasing dust aerosols is parameterized as an inverse and linear function of LAI (Mahowald et al., 2006a). The inhibition of soil moisture on dust deflation, and thus dust emission, activates when the near-surface soil gravimetric water content exceeds a threshold value, determined by the static mass fraction of the clay soil, and is parameterized in the land model according to a semi-empirical relation (Fécan et al., 1999).

The size distribution of the emitted dust is derived using the brittle fragmentation theory developed by Kok (2011b) distributing 0.1%, 1.0%, and 98.9% percentage of dust mass into Aitken, accumulation, and coarse modes, respectively, independent of the friction velocity upon dust emissions (Kok, 2011a).

## 2.3 Speciated dust aerosol modeling

The bulk dust model (Sect. 2.2) has previously been modified to speciate the bulk dust into eight mineral tracers, which allows more detailed optical properties as a function of minerals (Scanza et al., 2015; Li et al., 2021). Using the approach of Claquin et al. (1999), Li et al. (2021) estimated a mean mineralogical composition in the soil at each model grid cell for the minerals illite, kaolinite, montmorillonite, hematite, quartz, calcite, feldspar, and gypsum (Fig. S2 of Li et al., 2021). These minerals represent the most common classes for clay- (soil grain diameter < 2 μm including the first 5 minerals) and silt-sized (diameter between 2-63 μm including the last 5 minerals) soil categories (Claquin et al., 1999).

As detailed in Scanza et al. (2015) and Li et al. (2021), additional modifications include: 1) the mineral components in soil types of Gypsic Xerosols and Yermosols, Gleyic and Orthic Solonchaks and salt flats were normalized to unity; 2) the same amount of hematite in the clay- and silt-sized categories was prescribed with equal and opposite change to the illite percentage; 3) the nearest neighborhood algorithm was applied to fill in the grid cells for dust emission; and 4) the soil mineralogy was converted to that of the dust aerosol following the brittle fragmentation theory (Kok, 2011b), as detailed in Scanza et al. (2015).

The distribution of the mass flux for each mineral into the three emission modes follows that of the bulk dust modeling (Sect. 2.2). The sum of the masses of the 8 considered minerals equals the total bulk dust mass without dust speciation. Each of the mineral aerosols are treated as a separate tracer in the same manner as bulk dust, experiencing advection, deposition, and aerosol microphysics (e.g., coagulation).

## 2.4 Dust optical properties and radiation flux diagnostic

We show results of the direct radiative effect calculations from two code versions: one with the bulk dust and the other with speciated dust. Aerosol optical properties (e.g., single scattering albedo, asymmetry factor) of the internal mixture in an aerosol mode are parameterized based upon the complex refractive index (CRI) of the mixture, which is calculated as the volume-weighted CRI of each component, including water (Ghan and Zaveri, 2007) in that mode. The wet size due to growth of aerosol particles by adsorbing water vapor follows the κ-Kohler theory with a time-invariant hygroscopicity for each aerosol species (Petters and Kreidenwei, 2007). CAM6.1 computes the net radiative flux based on the radiation fluxes diagnosed for each model layer at 14 shortwave and 16 longwave spectral bands per model hour. The direct radiative effect by dust aerosols under all-sky conditions is then determined by calculating the difference of the net radiative flux with and without dust at the top of the atmosphere under all-sky conditions. We augmented the longwave radiative effect from the model by 51% to account for the dust scattering (Dufresne et al., 2002). The DRE efficiency, which we used to evaluate the model performance on simulating the dust optical properties, is defined as the ratio of dust DRE to dust optical depth (DOD) under clear conditions. This study does not consider the indirect radiative effect which is subject to substantially larger uncertainty due to the complexity involved in cloud microphysics (IPCC, 2021).

## 2.5 Changes to the dust parameterizations in CAM6.1/CLM5

The model developments introduced in this section are closely related to the three major components of the dust cycle (emission and removal mechanism) and the radiative effects. Specifically, we incorporate into CAM6.1 a relatively new dust emission scheme originally developed by Kok et al., (2014a, b), a dry deposition scheme developed by Petroff and Zhang (2010) and incorporated in CAM5 by Wu et al., (2018), and the influence of dust non-spherical shape on the removal rate of dust aerosol particles (Huang et al., 2020).

### 2.5.1 Dust emission schemes

The vertical flux of dust emitted by wind erosion in a model grid cell is represented by

$$\varphi_d = \lambda S_f F_{d,CLM5}, \qquad (1)$$

where λ is a global tuning factor, $S_f$ is the source function that shifts the dust emission to the most erodible sources, such as the Bodélé depression in North Africa (Zender et al., 2003a), and $F_{d,CLM5}$ is the vertical emission flux predicted by the dust emission scheme in CLM5 (Kok et al., 2014a).

As part of the DEAD scheme (Zender et al. 2003a), dust sources are strongly associated with the erodible soils (Ginoux et al., 2001). These source regions are parameterized using information contained in the time invariant geomorphology map (Zender et al., 2003) which was optimized (Albani et al., 2014) to match the observed dust optical depth (DOD).

The vertical dust emission in CLM5 occurs when the friction velocity ($u_*$) exceeds the threshold friction velocity ($u_{*t}$) which is parameterized in DEAD as

$$F_{d,CLM5} = \begin{cases} C_{MB}\chi f_{bare} \dfrac{\rho_a}{g} u_*'^3 \left(1 - \dfrac{u_{*t}'^2}{u_*'^2}\right)\left(1 + \dfrac{u_{*t}'}{u_*'}\right), \, u_*' > u_{*t}' \\ 0, \, u_*' \le u_{*t}' \end{cases} ,(2)$$

Where CMB equals; $f_{bare}$ is the bare soil fraction, $\rho_a$ is the atmospheric density; $u_*'$ is the friction velocity; $u_{*t}'$ is the threshold friction velocity; the sandblasting efficiency $\chi$ is written as a function of the clay fraction ($f_{clay}$)

$$\chi = 10^{13.4 f_{clay}{}^{-6}}. (3)$$

Kok et al. (2014a) developed a new dust emission scheme for climate models based on the brittle fragmentation theory (Kok, 2011b), which avoided the use of such a static soil erodibility map (the source function, $S_f$) while improving the accuracy of dust cycle modeling (Kok et al., 2014b); although even dust modeling with DEAD can be improved if optimized against observations (Kok et al., 2021). Improvements are likely achieved because, compared to that in DEAD, the dust emission in BRIFT tends to be more sensitive to the soil's threshold friction velocity and thus to the 270 surface physical conditions when soil becomes more erodible, owing to the introduced dust emission coefficient (Kok et al., 2014a) and the new method of calculating the threshold gravimetric water content in the top soil layer (see Eq. 4 of Kok et al., 2014b). Below we briefly introduce the new dust emission scheme.

In this new scheme the vertical dust emission in CLM5 is expresses as

$$F_{d,CLM5} = \begin{cases} C_d f_{bare} f_{clay} \dfrac{\rho_a(u_*^2 - u_{*t}^2)}{u_{st}} \left(\dfrac{u_*}{u_{*t}}\right)^{C_\alpha \frac{u_{*st} - u_{*st0}}{u_{*st0}}}, u_* > u_{*t}. \\ 0, u_* \le u_{*t} \end{cases} (4)$$

where $\rho_{a0}$ is the atmospheric density; $u_{*st}$ is the threshold friction velocity standardized according to atmospheric density and the standard atmospheric density of $\rho_{a0}$=1.225 kg m$^{-3}$,

$$u_{*st} = u_{*t} \sqrt{\dfrac{\rho_a}{\rho_{a0}}}; (5)$$

and $u_{*st0}$ is the minimal value of $u_{*st}$ equaling 0.16 m s$^{-1}$.

Since the presence of non-erodible roughness elements, such as rocks, is not considered in CLM5,

$$u_* = u'_*, \quad (6)$$

$$u_{*t} = u'_{*t}. \quad (7)$$

Because of the neglection of the non-erodible elements, $u_{*t}$ is mostly determined by soil moisture content, which means that the augmentation factor of $u_{*t}$ is:

$$f_{*t} = \begin{cases} \sqrt{1 + 1.21(w - w')^{0.68}}; w > w' \\ 1; w \leq w' \end{cases} \quad (8)$$

Where w and w' are soil moisture content and the threshold gravimetric water content of the top soil layer in percentage.

Fécan et al. (1999) parameterized the threshold gravimetric water content (w) of the top soil layer by

$$w' = b\left(17 f_{clay} + 14 f_{clay}^2\right), \quad (9)$$

where b is a tuning factor.

Equations (8) and (9) are also used in DEAD with an equivalent tuning factor b set to be $f_{clay}^{-1}$ which in BRIFT is set as unity. The clay fraction is taken from the FAO(2012) soil database (see Fig. S1 of Kok et al., 2014).

The dust emission coefficient, Cd, in Eq. 4 is expressed as

$$C_d = C_{d0} exp\left(-C_e \frac{u_{*st} - u_{*st0}}{u_{*st0}}\right), \quad (10)$$

where $C_{d0}$ equals 4.4×10$^{-5}$; $C_e$ equals 2.0.

### 2.5.2 Dry deposition schemes

**The default dry deposition scheme, Z01**

As is typical among aerosols dry deposition resistance models, CAM6.1 includes parameterizations of gravitational settling ($V_g$), aerodynamic ($R_a$) and surface resistance ($R_s$).

The gravitational settling is parameterized following

$$V_g = \frac{\rho_d g C_c d_p^2}{18\mu}, \quad (11)$$

where $\rho_d$ is the particle density (unit: kg m$^{-3}$), $d_p$ is the particle diameter (unit: m), g the acceleration of gravity, $C_c$ is the Cunningham correction factor as a function of $d_p$ and the mean free path of air molecules, μ is the viscosity coefficient of air.

Aerodynamic resistance is parameterized following

$$R_a = \frac{\ln(Z_R/Z_0) - \varphi_H}{\kappa u_*}, \quad (12)$$

where $Z_R$ is the reference height; $Z_0$ is the roughness length; $\varphi_H$ is the stability function, κ is the von Karman constant set as 0.4; and u* is the friction velocity.

The surface resistance dominates over aerodynamic resistance under turbulent conditions, and is written as

$$R_s = \frac{1}{\varepsilon_0 u_* (E_B + E_{IM} + E_{IN}) R_1}, \quad (13)$$

where $\varepsilon_0$ is an empirical constant set as 3.0; $R_1$ is the factor to represent particle rebound; $E_B, E_{IM}, E_{IN}$ are collection efficiencies due to Brownian diffusion, impaction, and interception, parametrized respectively as:

$$E_B = Sc^{-\gamma}, \quad (14)$$

$$E_{IM} = \left(\frac{St}{St + \alpha}\right)^\beta, (15)$$

$$E_{IN} = 0.5 \left(\frac{d_p}{A}\right)^2, (16)$$

where Sc is the Schmidt number, defined as the ratio of the kinematic viscosity of air (v) to the particle Brownian diffusivity (D); γ depends on land use categories, typically ranging between [0.50, 0.67]; α depends on the land use categories; β=2; A is the characteristic radius of collectors depending on land use categories; St is the Stokes number parameterized following

$$St = \begin{cases} \frac{V_g u_*}{gA}; \text{ Vegeted surfaces} \\ \frac{V_g u_*^2}{v}. \text{ Smooth surfaces or surfaces with bluff roughness elements} \end{cases} \quad (17)$$

According to Eq. 13, the surface resistance consists of three processes, two applicable to all land types (Brownian diffusion and impaction), and one only to non-smooth surfaces (interception). All the three processes are a function of aerosol size through empirical coefficients constrained by matching the modeled dry deposition velocity with field and laboratory measurements.

The dry deposition velocity then has the form of

$$V_d = V_g + \frac{1}{R_a + R_s}. \quad (18)$$

With more observations available to constraint these coefficients, the default Z01 (Zhang et al., 2001) used in CAM6.1 was found to greatly overestimate dry deposition rates for fine particles (diameter < 1 μm: Aitken and accumulation mode) and slightly underestimate (relative to the large change with fine particles) the rates for coarse particles (diameter around 1 or 2 μm) (Petroff and Zhang, 2010; Wu et al., 2018; Farmer et al., 2020).

**The new dry deposition scheme, PZ10**

The new scheme (PZ10; Petroff and Zhang, 2010) uses a quite different formula to calculate the dry deposition scheme as follows:

$$V_d = V_{drift} + \frac{1}{R_a + 1/R_{ds}}, \quad (19)$$

where the drift velocity,

$$V_{drift} = V_g + V_{phor}. \quad (20)$$

Therefore, this new scheme includes the effect ($V_{phor}$) of different physical processes (thermophoresis, diffusiophoresis, and electricity) occurring between water, ice, and snow surfaces and the air immediately above them, which can result in a downward flux of particles (the phoretic effect; Petroff et al., 2008). PZ10 accounts for such effects of thermophoresis and diffusiophoresis for particle deposition over the three surface types by assigning constant values of $5\times10^{-5}$ m s$^{-1}$ to water and $2\times10^{-4}$ m s$^{-1}$ to ice and snow surfaces, which allows the scheme to better reproduce the available measurements than Z01 (Petroff and Zhang, 2010). This constant is set to zero for all the other surface types. The phoretic effect tends to dominate deposition of fine particles over Brownian diffusion under low wind conditions (friction velocity less than ~ 11 cm s$^{-1}$). Because of the reduced Brownian diffusion efficiency compared to Z01, PZ10 corrects the high bias seen in Z01 for the deposition of fine particles (Emerson et al., 2020; Petroff and Zhang, 2010; Wu et al., 2018).

PZ10 parametrizes the gravitational settling velocity $V_g$ in the same as Z01 but the aerodynamic resistance ($R_a$) in a different formula as well.

For non-vegetated surfaces,

$$R_a = \frac{1}{\kappa u_*} \left[ \ln\left(\frac{Z_R - d}{Z_0}\right) - \Psi_h\left(\frac{Z_R - d}{L_o}\right) + \Psi_h\left(\frac{Z_0}{L_o}\right) \right], (21)$$

where d, and $L_o$ are the canopy height, the displacement height of the canopy, and the Obhukov length, respectively; $\Psi_h$ is the integrated form of the stability function for heat.

The surface dry deposition velocity is expressed as

$$V_{ds} = u_*(E_{gb} + E_{IT}), (22)$$

Where $E_{IT}$ is the efficiency of collections by turbulent impaction; $E_{gb}$ represents Brownian diffusion, written as

$$E_{gb} = \frac{Sc^{-2/3}}{14.5}\left[\frac{1}{6}\ln\frac{\left(1+\sqrt[3]{Sc}/2.9\right)^2}{1-\sqrt[3]{Sc}/2.9+\left(\sqrt[3]{Sc}/2.9\right)^2}+\frac{1}{\sqrt{3}}\tan^{-1}\frac{2\left(\sqrt[3]{Sc}/2.9\right)-1}{\sqrt{3}}+\frac{\pi}{6\sqrt{3}}\right]^{-1}, (23)$$

For vegetated surfaces,

$$R_a = \frac{1}{\kappa u_*}\left[\ln\left(\frac{Z_R-d}{h-d}\right)-\Psi_h\left(\frac{Z_R-d}{L_o}\right)+\Psi_h\left(\frac{h-d}{L_o}\right)\right], \quad (24)$$

where h is the displacement height of the canopy.

The expression for surface deposition velocity is written as

$$V_{ds} = u_*E_g\frac{1+\left(\frac{Q}{Q_g}-\frac{\delta}{2}\right)\frac{\tanh\eta}{\eta}}{1+\left(Q_g+\frac{\delta}{2}\right)\frac{\tanh\eta}{\eta}}, \quad (25)$$

where $E_g$ is the collection efficiency on the ground below the canopy.

$E_g$ includes Brownian diffusion and the turbulent impaction, $E_{gt}$, given as

$$E_{gt} = 2.5\times10^{-3}C_{IT}\tau_{ph}^2, \quad (26)$$

where $C_{IT}$ equals 0.14, and $\tau_{ph}$ is the dimensionless particle relaxation time.

 Q and $Q_g$ in Eq. 25, are given as

$$Q = LAI\cdot\frac{(E_B+E_{IN}+E_{IM})\,U_h/u_*+E_{IT}}{I_{mp}(h)/h}, \quad (27)$$

$$Q_g = \frac{E_gh}{I_{mp}(h)}, \quad (28)$$

where LAI is the two-sized leaf area index; $E_B$, $E_{IN}$, and $E_{IM}$, are Brownian diffusion, interception, and inertial impaction.

In Eq. 25,

$$\eta = \sqrt{\delta^2/4 + Q} \,, (29)$$

where the aerodynamic extinction coefficient δ is expressed as

$$\delta = \left(\frac{LAI \cdot k_x}{12\kappa^2(1 - d/h)^2}\right)^{1/3} \phi_m^{2/3} \left(\frac{h - d}{L_o}\right), (30)$$

where $k_x$ is the inclination coefficient of the canopy elements; and $\phi_m$ is the nondimensional stability function for momentum.

### 2.5.3 Dust asphericity

To account for the influence of dust asphericity on the gravitational settling velocity, we first calculated the asphericity factor γ (defined as the ratio of the gravitational settling velocity of aspherical dust to that of spherical dust) offline based on a combination of observed dust shape parameters ($F_s$) previously compiled by Huang et al. (2020) following

$$\gamma = \frac{2}{F_s^{1/3} + 1/F_s^{1/3}}, \quad (31)$$

Fs is parameterized by the dust shape parameters

$$F_s = \frac{D_g^3}{L^{2.3} \bullet W^{0.7}}, \quad (32)$$

where $D_g$ is the volume-equivalent diameter of the dust particle defined by three axes (L: length, W: width, and H: height, respectively) of the ellipsoid having a form of

$$D_g = \sqrt[3]{LWH}. \quad (33)$$

The orientation of particles during gravitational settling determines the drag coefficient. Equation 29 assumes that during settling, aerosol particles randomly orientate. This assumption is reasonable, since for dust falling in the Earth's atmosphere, 1) the Reynolds number, Re << 1 (Kok et al, 2012), and, especially, 2) CAM6 does not simulate super coarse dust particles (diameter > 10 μm), for which such an assumption may introduce high errors. A previous study (Bagheri and Bonadonna 2016) suggests that this approximation of the influence of the dust asphericity on the gravitational settling velocity is accurate and reliable with a mean and the maximum errors of 2.4%, and 33.9%, respectively. Equations 31-33 indicate a range of γ between 0 and 1. When a dust particle becomes less ellipsoidal γ is getting closer to 1.

In the Stokes regime (Kok et al., 2012), where the gravitational settling of dust usually occurs, the terminal velocity of spherical (sph) and ellipsoidal (asp) dust is approximated as

$$V_{g,sph} = \frac{g\rho_p}{18\mu} D_g^2, \quad (34)$$

and

$$V_{g,asp} = \gamma V_{g,sph}, \quad (35)$$

respectively, where g is the gravitational constant (~9.8 m s$^{-2}$), $\rho_p$ is the dust density (~2,500 kg m$^{-3}$), $\mu$ is the dynamic viscosity of air (1.81×10$^{-5}$ Pa s).

In this calculation, we also assume that the dust shape parameters are independent of the size of dust aerosol particles. Therefore, a constant revision of the dust gravitational settling velocity (the calculated value in the model by default is for spherical aerosols) due to dust asphericity by multiplying the velocity by $\gamma$ was applied to dust species in the three modes that contains dust aerosols (Aitken, accumulation, and coarse). The size independency assumption of dust asphericity follows the recent observational evidence that there does not exist a statistically significant relationship between the shape parameters (aspect ratio and height-to-width ratio) and dust sizes (Huang et al., 2020). Measurements made at different locations show that the shape parameters (e.g., aspect ratio; Fig. 3 of Huang et al., 2020), which we used to calculate $V_{g,asp}$, change for dust during transport. But, because of highly limited measurements of dust shape parameters, we subjectively divided the dust coverage into "close-to-source", "short-range", and "long-range" zones and calculated the asphericity factor $\gamma$ for each of the zones, the global map of which is shown in Fig. S1, ranging between 0.82 and 0.93. In the regions where the shape parameter measurement is sparse or unavailable, such as those in the Southern Hemisphere, the shape parameters from the global median are used instead to calculate the asphericity factor yielding a value of 15%. We acknowledge limitation of the methodology here to account for the lifetime effect of dust asphericity, anticipating improvements on modeling this effect when more high-quality dust shape measurements become available.

Matching modelled DOD to observations requires the model to account for the dust asphericity, which acts to enhance the mass extinction efficiency of particles, particularly in the coarse mode (Kok et al., 2017). This enhancement in the mass extinction efficiency due to the dust asphericity is not included in the current version of CESM2.1 but will be incorporated into a future officially released CESM2.1 version. According to calculations of Kok et al., (2017), the dust mass extinction efficiency at the visible band due to dust asphericity is approximately 16% and 28% higher for non-spherical particles than for spherical particles in the fine (accumulation plus Aitken) and coarse modes, respectively. Consequently, the model requires lower dust emissions to achieve a global DOD of ~0.030 compared to simulations without considering dust asphericity. The shape effect on the mass extinction efficiency may also explain the difference between the global mean DOD in Aerosol Comparisons between Observations and Models (AEROCOM; median: 0.023) (Huneeus et al., 2011) and that in Ridley et al. (2016) (0.03±0.005) near the visible band. We have included the enhanced dust mass extinction efficiency due to dust asphericity in our previous studies (e.g., Li et al., 2021, Kok et al.,2021),

which suggests that the inclusion of this enhanced dust mass extinction efficiency would reduce the overestimation of the surface concentration (Kok et al.,2021). Here we do not investigate its impact on the simulated dust cycle.

**2.6 Experiment design**

Table 2 lists the simulations designed for the present study. In all simulations, the CAM6.1 with different modifications is configured as a stand-alone model where the atmosphere is coupled to active land and sea ice models, and to a data ocean and slab glacier models. Each simulation in these sets was performed at the spatial resolution of $1.25° \times 0.9° \times 56$ (longitude by latitude by vertical layers) using a data ocean for years 2006-2011, with the simulated data for the last five years used for analysis. In addition, the meteorology field (horizontal wind, air temperature T, and relative humidity) was

nudged toward the Modern-Era Retrospective analysis for Research and Applications, Version 2 (MERRA-2) at a 6-hour relaxation time scale. The anthropogenic emissions were taken from the Climate Model Intercomparison Program (CMIP6) inventory for the year 2000 (Eyring et al., 2016). The enhancement of the mass extinction efficiency of aerosol particles by dust asphericity is included in all the simulations, since we do not attempt to quantify how this enhancement impacts the simulated dust cycle. An offline sensitivity test (Table S1) supports the use of unity tuning factor to calculate

the threshold gravimetric water content which we employed in the experiments for quantifying influence of each modification (speciated dust simulations listed in Table 2).

NEW_EMIS serves as the baseline simulation for quantifying the impact of the coarse-mode size change on the dust cycle modeling. NEW_EMIS_SIZE completely and NEW_EMIS_SIZE_WIDTH partly reverted the coarse-mode size

distribution to that used in CAM5 (Table 1). The other changes to the width of the accumulation mode and the bounds of the simulated GMD online impose negligible impacts on the dust cycle modeling, thus, we did not construct sensitivity tests on them in this study. We investigate how the incorrect dust size distribution influences the dust cycle modeling and the estimate of dust DRE in the bulk-dust model rather than in the speciated-dust model, because this incorrect size distribution has been employed in previous studies using the officially released bulk-dust CAM6 only and not in any

study using the speciated-dust CAM. It is also reasonable to make all the quantifications in the model that use a correct dust size distribution. Therefore, we reverted the dust size distribution in all the speciated-dust runs to that configured in CAM5.

We quantified the impact of each of the modifications (Z01 to PZ10, spherical to aspherical dust, and DEAD to BRIFT)

on the simulated dust cycle and DRE by differentiating corresponding results in the paired simulations that contain identical developments except for the targeted modification. Specifically, we quantified the impact of changing (1) Z01 to PZ10 by taking the difference between the simulation with Z01 (MIN_NEW_EMIS_SHAPE) and that with PZ10 (CAM6.α_MIN), (2) spherical to aspherical dust between the simulation with special dust (MINE_NEW_EMIS) and that with spherical dust (MIN_NEW_EMIS_SHAPE), and, (3) DEAD to BRIFT  between the simulation using DEAD

(MINE_NEW_EMIS) and that using BRIFT (MINE_BASE).

Note that there are many ways to conduct sensitivity studies, which could lead to slightly different results. We added the modification on top of the previous change to understand how the simulated dust cycle evolves while updating the model (MIN_BASE) toward the most advanced version (CAM6.α_MIN). This may not hinder a clean comparison of the effect

of each development, since the "interaction" between the existing and the newly introduced parameterizations appears weak (Fig. S2).

With the dust tuning applied toward the similar global mean DOD of ~0.030 the modeled dust cycle (i.e., burdens, concentrations, loadings, and deposition fluxes) would be similar between the bulk- and speciated-dust models that are nudged toward identical offline dynamics and using the same dust size distribution (see Sect. 6). The quantified effect of each of the modifications would thus be similar if using the bulk dust model instead (Fig. S2), except that the modeled dust optical properties (e.g., single scattering albedo) by the bulk and speciated dust models would differ considerably, resulting in considerably different dust DRE (Scanza et al., 2015) and DRE efficiencies between NEW_EMIS (CAM6.α) and MINE_NEW_EMIS (CAM6.α_MINE). A comparison of the bulk- and speciated-dust models on simulating dust DRE had been previously documented (Scanza et al., 2015). This study includes the speciated-dust runs, because we want to verify as well if the updates help improve the agreement with the observed dust DRE efficiency in the dust speciated model which could better represent the spatial variation of the dust optical properties.

We tuned CAM6.1, NEW_EMIS, CAM6.α, MINE_BASE, and MINE_NEW_EMIS, following Albani et al., (2014), by modifying a CAM namelist variable, dust_emis_fact, such that the simulated global mean DOD is ~0.030 at the visible band centered at 0.53 µm (hereafter unless stated otherwise), an estimate obtained by an integrated analysis of the AERONET-based measurements, bias-corrected satellite-retrievals, and a model ensemble (Ridley et al., 2016). We prefer to tuning the model to reproduce the global mean DOD, 0.030, because DOD is currently the best estimate of global dust quantities, compared to the others (i.e., dust concentrations). It turns out that doing so can also reasonably reproduce the other quantities with no need of a regional tuning. MINE_NEW_EMIS requires the dust tuning to use a much larger tuning parameter (dust_emis_fact=3.6; Table 2), than MINE_BASE (dust_emis_fact=1.6), because, otherwise, if using the same dust_emis_fact as in DEAD, the dust emissions in BRIFT would lead to an unrealistically high global mean DOD (>~0.5).

The dust tuning was not applied to NEW_EMIS_SIZE and NEW_EMIS_SIZE_WIDTH, but the emission in which was kept identical to NEW_EMIS, to see how changes in the transported dust size distribution affects the DOD calculation. Because of the rough linearity among DOD, DRE, and dust burdens (Liao and Seinfeld, 1998; Mahowald et al., 2006b), when comparing surface dust concentrations, dust loadings, and deposition fluxes, we rescaled each of them using the same factor to achieve the global mean DOD ~0.030. For the other cases (MINE_NEW_EMIS_SHAPE and CAM6.α _MINE), as will be seen, the global mean DOD only changes slightly within the uncertainty range (0.025-0.035; Ridley et al., 2016). The model retuning is, thus, not required.

## 3 Observational datasets for model evaluations

Tables 3-5 summarize available datasets used to evaluate the model performance, detailed descriptions about each datum, and how they are used in the model-data comparison. Due to limitations in precisely matching the period and locations between model results and data, the evaluations focus on checking if models can capture overall features of the measured/observed/retrieved dust cycle and the corresponding dust DRE efficiency. We summarize limitations going beyond this mismatch on period and location and common in all the model-data comparisons in Sect. 7.

### 3.1 Surface dust concentrations and dust aerosol optical depth from AERONET

We used monthly surface dust concentration data that Albani et al. (2014) compiled from measurements made using high-volume filter collectors at the University of Miami Ocean Aerosol Network and station-based data that have been previously compiled on annual averages (Mahowald et al., 2009; Zuidema et al., 2019). Because the model only simulates dust <10 μm (the cut off value of aerosol size) in diameter, the Albani et al. (2014) compilation has been processed to estimate the flux of dust below the size cut-off according to reported or assumed parameters (e.g., geometric standard deviation) for the size distribution of transported dust. Simulated DOD is compared to Aerosol Robotic Network (AERONET) retrievals, which were subject to data quality control and station selection based on the dust dominance in the reported AOD. The data quality control includes a minimum of 10 days per month that contains valid retrievals, the annually averaged Angstrom Exponent <1.2 (the larger the value, the smaller the aerosol size), and a full coverage of the data availability through a year within the observation period (Albani et al., 2014).

### 3.2 Surface dust deposition fluxes

The dust deposition flux data used here were those that Albani et al. (2014) compiled from publications (Tegen et al., 2002; Ginoux et al., 2001; Lawrence and Neff, 2009; Mahowald et al., 2009) for the present-day climate. Since the model only simulates dust <10 μm (the cut off value of aerosol size) in diameter, the Albani et al. (2014) compilation was processed to estimate the surface concentration of dust below the size cut-off, according to reported or assumed parameters (e.g., geometric standard deviation) for the size distribution of transported dust.

**Table** 3. Observed/retrieved cycle for dust model evaluations including optical depth, surface mass concentrations, surface deposition fluxes, and wet deposition percentages. AERONET: Aerosol Robotic Network; MODIS: Moderate Resolution Imaging Spectroradiometer; AOD: aerosol optical depth; DOD: dust optical depth.

| Dust properties | Representative locations | Platform/Instruments | Levels | Time periods | References | Comments |
|---|---|---|---|---|---|---|
| Dust optical depth | Filtered AERONET sites (see Fig. 1a of this study) | Sun photometers | All height levels | 2003-2013 | Albani et al. (2014) | 1) Data quality control; 2) Months selected containing data for at least 10 days; 3) Years selected have a full 12-months coverage; 4) non-dust aerosols filtered out based on the Ångström exponent and single scattering albedo |
| | Regional averages | Multiple satellite platforms and models | All height levels | 2004-2008 | Ridley et al. (2016) | Seasonal value obtained by combining four global climate models with multiple satellite aerosol products that were bias corrected using station-based AERONET data |
| | Terra/Aqua tracks; Regional averages | MODIS | All height levels | 2003-2015 | Pu et al. (2020) | 1) Non-dust aerosols filtered out based on AE and single scattering albedo; 2) An empirical function that relates DOD to AOD and the Ångström exponent |
| Surface mass concentrations | See Fig. 1d of this study | High-volume filter collectors | Near ground surface | 1991-1994 | Prospero and Nees (1986) Prospero and Savoie (1989) | This study uses both monthly data and period averaged climatology |

| | | | | | | |
|---|---|---|---|---|---|---|
| Surface deposition fluxes | See Fig. 1g of this study | Sampling filters | At and/or near ground surface | See references | Tegen et al. (2002); Ginoux et al. (2001); Lawrence and Neff (2009); Mahowald et al. (2009) | Data compiled by Albani et al. (2014) and has been processed to get the mass fraction of dust below 10 µm based on reported size parameters, such as geometric standard deviation; see Albani et al. (2014) for details |
| Wet deposition percentages | Ten sites; see Table 7 of this study (1st column) | See references | At and/or near ground surface | See references | R.Arimoto et al. (1985); Uematsu et al. (1985); Arimoto et al. (1990); Hillamo et al. (1993); Jickells et al. (1998); Wagenbach et al. (1998); Wolff et al. (2006) | Data compiled by Mahowald et al. (2011b) |

## 3.3 Size distributions of dust aerosol

Most of the remotely sensed, size-resolved dust volume retrievals used here were taken from the AERONET Level 2.0 Almucantar Retrievals (Version 2), which is reported for 22 size bins with bimodal size distribution and ellipsoid shape of aerosol particles (Dubovik et al., 2000). This data overestimates dust mass in the submicron size range and has possible contamination by non-dust aerosols (Albani et al., 2014; Dubovik et al., 2000; Mahowald et al., 2014). We, therefore, only retain the super micron fraction of dust in the comparison, even though AERONET may underestimate

the mass of dust between 1-10 µm in diameters (McConnell et al., 2008). The data processing procedure is detailed in Albani et al., (2014). Near North Africa, we also compare the modelled size distribution of dust aerosols with measurements from Otto et al. (2007) taken in the vicinity of the Canary Islands, from Ryder et al. (2013) by aircraft with a track between the Canary Islands and Mauritania/Mali, and from Ryder et al. (2018) near Cape Verde.

**Table** 4. Measured/retrieved dust size distribution for model evaluation. AERONET: Aerosol Robotic Network; DustCOMM: Dust Constraints from joint Observational-Modelling-experiMental analysis.

| | | | | | |
|---|---|---|---|---|---|
| AERONET sites | Sun photometers | Near ground surface | 2003-2013 | Holben et al. (1998); Dubovik et al. (2000) | 1) AERONET Level 2.0 Almucantar Retrievals (Version 2); 2) data reported for 22 size bins with bimodal size distribution and ellipsoid shape of aerosol particles (Dubovik et al., 2000); only the super micron fraction of dust in the comparison used, even though AERONET may underestimate the mass of dust between 1-10 µm in diameters (McConnell et al., 2008) |
| Near the Canary Islands | See Table 1 of Otto et al. 2007 | At flight heights: 2700, 4000, 5500, 7000 m | June-July in 1997 | Otto et al. (2007) | Data obtained from Fig. 3 of Adebiyi et al. (2020) |
| Along flight tracks between the Canary Islands and Mauritania/Mali near Cabo Verde | See Table 3 (second column) of Ryder et al. 2013 for details | At flight heights between 0-3000 and 0-6000 m | June in 2011 | Ryder et al. (2013) | Data obtained from Fig. 3 of Adebiyi et al. (2020) |
| DustCOMM/global | Joint observation and models | All height levels | See Adebiyi et al. (2020) | Adebiyi et al. (2020) | Data obtained from Fig. 5a of Adebiyi et al. (2020) |

## 3.4 The direct radiative effect efficiency of dust

The modeled direct dust DRE efficiency (ratio of dust DRE to DOD) is compared to satellite-based observations under clear-sky conditions at the top of the atmosphere. These observations include:


1) longwave dust DRE efficiency derived over North Africa by Zhang and Christopher (2003) for September 2000 based on measured longwave (5-200 µm) fluxes at the top of the atmosphere from the Multi-angle Imaging Spectroradiometer (MISR), and the Clouds and the Earth's Radiant Energy System (CERES) instrument and AOD at 0.55 µm from MODIS;


2) the shortwave (0.3-5 µm) dust DRE efficiency obtained by Li et al.(2004) with the measured shortwave flux from CERES and AOD at 0.55 µm from MODIS near North Africa (15-25ºN, 45-15ºW) in the summer to winter months between 2000 and 2001;

3) the shortwave (0.3-5 µm) dust DRE efficiency that Patadia et al. (2009) derived using a 1-D radiative transfer model, radiative fluxes from CERES, and AOD at 0.55µm from MISR and Ozone Monitoring Instrument (OMI) over the high-reflective regions (surface albedo and 0.55 µm > 0.35) of Saharan desert (15-30ºN, 10ºW-30ºE) for the summer months in 2005 and 2006.

**Table** 5. Retrieved dust radiative effect efficiency for model evaluation. CERES: Clouds and the Earth's Radiant Energy System; TOA: top of the atmosphere; JJA: June, July, and August; AOD: aerosol optical depth; MISR: Multi-angle Imaging SpectroRadiometer; OMI: Ozone Monitoring Instrument; NDJ: November, December, and January; MODIS: Moderate Resolution Imaging Spectroradiometer; CALIPSO: Cloud-Aerosol Lidar and Infrared Pathfinder Satellite Observations; MFRSR: MultiFilter Rotating Shadowband Radiometer; SEVIRI: Spinning Enhanced Visible and Infrared Imager; GERB: Geostationary Earth Radiation

Budget; AERONET: Aerosol Robotic Network; MPL: Micro-Pulse Lidar; AERI: Atmospheric Emitted Radiance Interferometer; SMART: Surface-sensing Measurements for Atmospheric Radiative Transfer; AMJ: April, May, and June.

| | | | | | |
|---|---|---|---|---|---|
| Sahara Desert [15°–30°N, 10°W–30°E] | Satellite CERES and model | TOA | JJA, 2005-2006 | Patadia et al. (2009) | Shortwave (0.3-5 µm); clear sky; AOD from MISR and OMI |
| Tropical Atlantic [15°–25°N, 15°–45°W] | Satellite CERES | TOA | JJA/NDJ, 2000-2001 | Li et al. (2004) | Shortwave (0.3-5 µm); clear sky |
| Tropical Atlantic [10°–30°N, 20°–45°W] | Satellite CERES, and model | TOA | JJA, 2007-2010 | Song et al. (2018) | Shortwave; clear sky; modelled AOD with constraints from MODIS/CALIPSO |
| Atlantic Ocean [0°–30°N, 10°–60°W] | Satellite CERES | TOA | JJA, 2000-2005 | Christopher and Jones (2007) | Shortwave; clear sky; AOD from MODIS |
| Mediterranean basin [35.5°N, 12.6°E] | Satellite CERES | TOA | September, 2004-2007 | Di Biagio et al. (2010) | Shortwave; clear sky; AOD from MFRSR |
| North Africa [15°–35°N, 18°W–40°E] | Satellite CERES | TOA | September, 2000 | Zhang and Christopher (2003) | Longwave (5-200 µm); clear sky; AOD from MODIS/MISR |
| West Africa [16°–28°N, 16°–4°W] | Satellite SEVIRI and GERB | TOA | JJA, 2006 | Brindley and Russell (2009) | Longwave; clear sky; AOD from AERONET and MISR |
| Niger-Chad [15°–20°N, 15°–22°E] | Satellite SEVIRI and GERB | TOA | JJA, 2006 | Brindley and Russell (2009) | Longwave; clear sky; AOD from AERONET and MISR |
| Sudan [15°–22°N, 22°–36°E] | Satellite SEVIRI and GERB | TOA | JJA, 2006 | Brindley and Russell (2009) | Longwave; clear sky; AOD from AERONET and MISR |
| Egypt/Israel [23°–32°N, 23°–35°E] | SEVIRI and GERB | TOA | JJA, 2006 | Brindley and Russell (2009) | Longwave; clear sky; AOD from AERONET and MISR |

| North Libya [27°–33°N, 15°–25°E] | Satellite SEVIRI and GERB | TOA | JJA, 2006 | Brindley and Russell (2009) | Longwave; clear sky; AOD from AERONET and MISR |
|---|---|---|---|---|---|
| South Libya [23°–27°N,15°–25°E] | Satellite SEVIRI and GERB | TOA | JJA, 2006 | Brindley and Russell (2009) | Longwave; clear sky; AOD from AERONET and MISR |
| Sahara Desert [15°–30°N, 10°W– 30°E] | Satellite CERES | TOA | JJA, 2005-2006 | Yang et al. (2009) | Longwave; clear sky; AOD from MISR and OMI |
| Tropical Atlantic [10°–30°N, 20°–45°W] | Satellite CERES and model | TOA | JJA | Song et al. (2018) | Shortwave; clear sky; modelled AOD with constraints from MODIS/CALIPSO |
| Atlantic Ocean [0°–30°N, 10°–60°W) | Satellite CERES | TOA | JJA, 2000-2005 | Christopher and Jones (2007) | Longwave; clear sky; AOD from MODIS |
| Cape Verde [16.7°N, 22.9°W] | Models | TOA | September, 2006 | Hansell et al. (2010) | Longwave; clear sky; AOD from MFRSR, MPL, CALIPSO, and AERI |
| Zhangye, China [39°N, 101°E] | Ground-based SMART | TOA | AMJ | Hansell et al. (2012) | Longwave; clear sky; AOD from MFRSR, MPL, CALIPSO, and AERI |

## 3.5 Other datasets

In addition to the abovementioned observations, we compare our results to datasets which combine model simulations and observations. Specifically, we compare 1) the modeled transported dust size distribution with that from the Dust Constraints from joint Observational-Modelling-experiMental analysis (DustCOMM) (Adebiyi et al., 2020) in global average; 2) regional dust deposition fluxes with the semi-observational data that were inverted based on an integration of a global model ensemble and quality-controlled observational constraints on the transported dust size distribution, extinction efficiency, and regional DOD (Kok et al., 2021a). This set of semi-observational data was shown to compare better with the high-quality measurement than model ensemble means or any individual model (Kok et al., 2021a); and 3) regional DOD seasonally with the estimates of Ridley et al. (2016), who obtained DOD by combining four global climate models with multiple satellite aerosol products that were bias corrected using station-based AEROENT data.

## 4 Model assessment metrics

Metrics used to evaluate the model performance against observations include the root mean square error (RMSE) and correlation efficient (Kendall's τ or Spearman's Correlation). Both the Kendall's τ and Spearman's Correlation are non-parametric methods which do not require a distribution of the data, such as Gaussian or normal. For dust deposition, loadings correlations calculated are to assess how well models reproduce both their regional climatology mean or one-time observation and the seasonal cycles. However, because of a lack of reliable monthly data, assessments for the dust DRE efficiency, DOD from Rideley et al. (2016), and percentages of wet deposition in the total deposition are on spatial variability based on the regional climatology mean or one-time observations. We tested the correlation significance of the metrics at the statistical confidence level of 95%. For the dust DRE efficiency and percentages of wet deposition, some domains only have a range available, such as, Sahara Desert (15º-30ºN, 10ºW-30ºE) in the longwave spectral range. For those domains, a mean of the low and high boundaries of the range is used in the calculation of the Spearman's Correlation and the corresponding significance test.

## 5 Results

Each of the modifications made to CAM6.1 (described in Sect. 2.5) is relevant to the modeled dust cycle, and, thus, relevant to the estimate of dust climatic impacts (e.g., direct radiative effects). The proposed new (CAM6.α) and default model versions (CAM6.1) simulated a similar (Fig. S3a: relative change ~16%; CAM6.α relative to CAM6.1) global mean dust loading of 24 and 29 Tg, respectively, and DOD of 0.032 (Fig. S3c: relative change < 1.3%) (Table 6). Comparing to the recent estimates that include very coarse dust which are not included in this model, the dust loadings here are well within the range of 22-30 Tg in Kok et al. (2021a) (Table 1 of their study), and are close to the 30 Tg in Adebiyi and Kok (2020a). But globally CAM6.α shows 54% more dust deposition than in CAM6.1 (Fig. S3b). The general spatial distributions of the relative change of dust loadings, deposition fluxes, and DOD are similar, though the magnitude of this change differs for some regions (e.g., North Africa, India).

**Table 6.** Simulated annual emission, loading, surface concentration, deposition, lifetime, DRE (all-sky conditions) and DREE (DRE efficiency; all-sky conditions) of dust speciated by mineralogy and bulk dust CAM6 with offline dynamics. The longwave direct radiative effect by dust was augmented by 51% (Dufresne et al., 2002) to account for dust scattering which is not represented in CAM by default. DOD and SSA shown are for the CAM6.1 visible band centered at 0.53 μm. The global mean dust SSA was calculated over model pixels where DOD/total AOD>0.5 as previously did (Scanza et al., 2015; Li et al., 2021). CAM6.α and CAM6.1 in bold represent the proposed new and default model versions, respectively.

| Cases | Emissions (Tg a$^{-1}$) | Dust loadings (Tg) | Surf conc. (μg cm$^{-3}$) | Deposition (Tg a$^{-1}$) | Lifetime (days) | DOD | Dust SSA | SW DRE (W m$^{-2}$) | LW DRE (W m$^{-2}$) | Net DRE (W m$^{-2}$) | Net DREE (W m$^{-2}$ τ$^{-1}$) |
|---|---|---|---|---|---|---|---|---|---|---|---|
| **CAM6.1** | 2421 | 29 | 38 | 2427 | 4.3 | 0.032 | 0.918 | -0.50 | 0.20 | -0.30 | -9.4 |
| NEW_EMIS | 1606 | 22 | 25 | 1609 | 4.9 | 0.030 | 0.931 | -0.66 | 0.42 | -0.24 | -8.0 |
| NEW_EMIS_SIZE | 1621 | 11 | 14 | 1622 | 2.4 | 0.013 | N/A | -0.39 | 0.30 | -0.094 | -7.2 |
| NEW_EMIS_SIZE_WIDTH | 1612 | 11 | 14 | 1613 | 2.4 | 0.019 | 0.936 | -0.51 | 0.31 | -0.20 | -11 |
| **CAM6.α** | 2891 | 24 | 25 | 2893 | 3.0 | 0.030 | 0.911 | -0.45 | 0.19 | -0.26 | -8.7 |
| MINE_BASE | 4456 | 27 | 41 | 4459 | 2.2 | 0.035 | 0.897 | -0.38 | 0.24 | -0.14 | -4.0 |
| MINE_NEW_EMIS | 2910 | 25 | 26 | 2912 | 3.1 | 0.029 | 0.900 | -0.29 | 0.23 | -0.06 | -2.1 |
| MINE_NEW_EMIS_SHAPE | 2914 | 26 | 27 | 2916 | 3.2 | 0.030 | 0.900 | -0.30 | 0.24 | -0.06 | -2.0 |
| CAM6.α_MINE | 2869 | 24 | 25 | 2871 | 3.1 | 0.031 | 0.896 | -0.31 | 0.24 | -0.07 | -2.3 |

### 5.1 Evaluation of model performance and improvements on the dust cycle modeling

### 5.1.1 Dust emissions

To achieve the global mean DOD of ~0.030, CAM6.α requires a dust emission of 2891 Tg a$^{-1}$ (Table 6), which falls below the estimate of 3400-9100 Tg a$^{-1}$ by Kok et al. (2021a; their Table 1) that accounts for dust between 0.1-20 μm in diameter and above the median, 1123 Tg a$^{-1}$, reported in AEROCOM phase I (Huneeus et al., 2011). The dust emission in CAM6.1 is also much lower than their estimate: 2421 Tg a$^{-1}$, which is, however, higher than the previous estimate (1490 Tg a$^{-1}$) with the same emission scheme (DEAD) and dust size range (<10 μm) but using the binned method (Zender et al., 2003).

There are no dust emission estimates from observations at a global-scale coverage. We thus infer the model performance on simulating dust emissions using model-data comparisons on the surface dust concentration and deposition flux.

However, such an evaluation of emission is probably achievable only when the observation site is close to the dust source. Otherwise, the reasoning would become incorrect, because of probable additional errors from the model representation on processes of dust transport and deposition, and interaction of dust with non-dust aerosols (e.g., sea salt and biomass burning). As will be seen in Sect. 5.1.3 and 5.1.4, in most of the grid cells containing the observational sites in North Africa, all experiments overestimated the deposition fluxes (Fig. 1g) and the surface dust concentrations (at Bani). This might suggest that, when turning the global DOD toward ~0.030, the model with the current settings and modifications probably overestimated dust emissions from North African sources, which is also shown in Kok et al. (2021a) using an integrated model ensemble and observational constraints. The smoother distribution of the dust emission in BRIFT than DEAD is due primarily to the use of the source function in DEAD that shifts dust emissions toward the most erodible soil, while in BRIFT, the near-surface friction velocity frequently exceeds the calculated threshold wind fraction velocity, causing dust to emit at more grid cells.

The locally emitted dust from the high-latitude region (> 50º N and < 40º S) in CAM6.α constitutes ~1.6% of the global total emitted dust flux, which is below the estimate of ~5% (2-3% for each hemisphere) derived from field and satellite observations (Bullard et al., 2016; Bullard, 2017). Especially for the northern high-latitude region, where local dust sources may dominate the near surface dust concentrations (Groot Zwaaftink et al., 2016), CAM6.α substantially underestimated its contribution to the global dust (<0.1%). This underestimation is what we expected, since the new scheme is designed to simulate dust emissions in low-latitude regions predominantly from the impact of saltators (Kok et al., 2012) and thus may not well capture the high-latitude dust emissions which occur through different physical processes. In comparison, despite missing dust sources > 60º S (Fig. 2a), CAM6.1 may overestimate the contribution of the high-latitude dust emission to the global dust total emission (8.0%). We attribute the much higher dust emission in the southern high-latitude region in CAM6.1 primarily to the higher emission from the South American sources (i.e., the Patagonian Desert) than in CAM6.α. This much higher dust emission is not due to local dust emissions from the Antarctic, because the local emission in the Antarctic though exists (Delmonte et al., 2013; Meinander et al., 2021; and Fig. 2b), it is weaker in strength (the contribution percentage <0.01%) than Patagonian Deserts (Fig. 2b), and the two models (CAM6.1 and CAM6.α) also simulated a percentage contribution of dust emission from the Antarctic sources comparable to each other. Since both dust emission schemes are far from perfect in reproducing the percentage contribution to the global dust emission and thus probably the high-latitude dust loadings, especially in the Arctic (i.e., Fig. 1e of Shi and Liu, 2019) where dust aerosol could impose big impact on polar clouds (Shi et al., 2021), a regional tuning of the local emission in the high-latitude regions is needed to better quantify the dust-cloud and dust-radiation interactions there.

### 5.1.2 Climatology annual means of dust optical depth, surface concentrations, and deposition fluxes

Over 90% of the measurement sites, all models reproduced the climatology of DOD from AERONET retrievals, the surface concentration, and deposition within a factor of ten (Fig. 1 and Fig. S4), with the spatial correlation between the models and observations statistically significant. Analysis of the spatial correlation (Pearson; R) and root mean square error (RMSE except for the surface dust concentration) suggests a substantial and statistically significant improvement in simulating DOD close to source region (Fig. 1c versus Fig. 1b: R=0.63 versus 0.41 for CAM6.α and CAM6.1, respectively, in log space; RMSE=0.30 versus 0.41 in log space). Note we obtained the RMSE in log space which removes the dominant influence of stations with high DOD (i.e., sites in North Africa and Middle East). So, the reduced

bias is because the new model better captures DOD over North Africa and Australia. Compared to the improvement in

DOD, the modifications do not notably better improve modeling the surface dust concentrations (Fig. 1f versus Fig. 1e: R=0.86 versus 0.75 for CAM6.α and CAM6.1, respectively; similar RMSE≈0.70 in both models) and dust deposition (Fig. 1i and Fig. 1h; R: 0.78 versus 0.69 and RMSE=0.85 versus 0.97). This is because the model's ability to simulate DOD, especially close to source regions, is subject to fewer potential errors than for surface dust concentration and deposition, which also require the model to simulate a correct vertical distribution. Therefore, the model's ability to

reproduce DOD close to source region appears to have improved at most of the sites (33 out of a total of 36 sites; especially in Australia as shown in Fig. 1a), but this improvement did not propagate to simulations of the dust surface concentrations (Fig. 1d: improvement at 24 out of a total of 47 sites) and deposition (Fig. 1g: improvement at 62 out of a total of 108 sites).

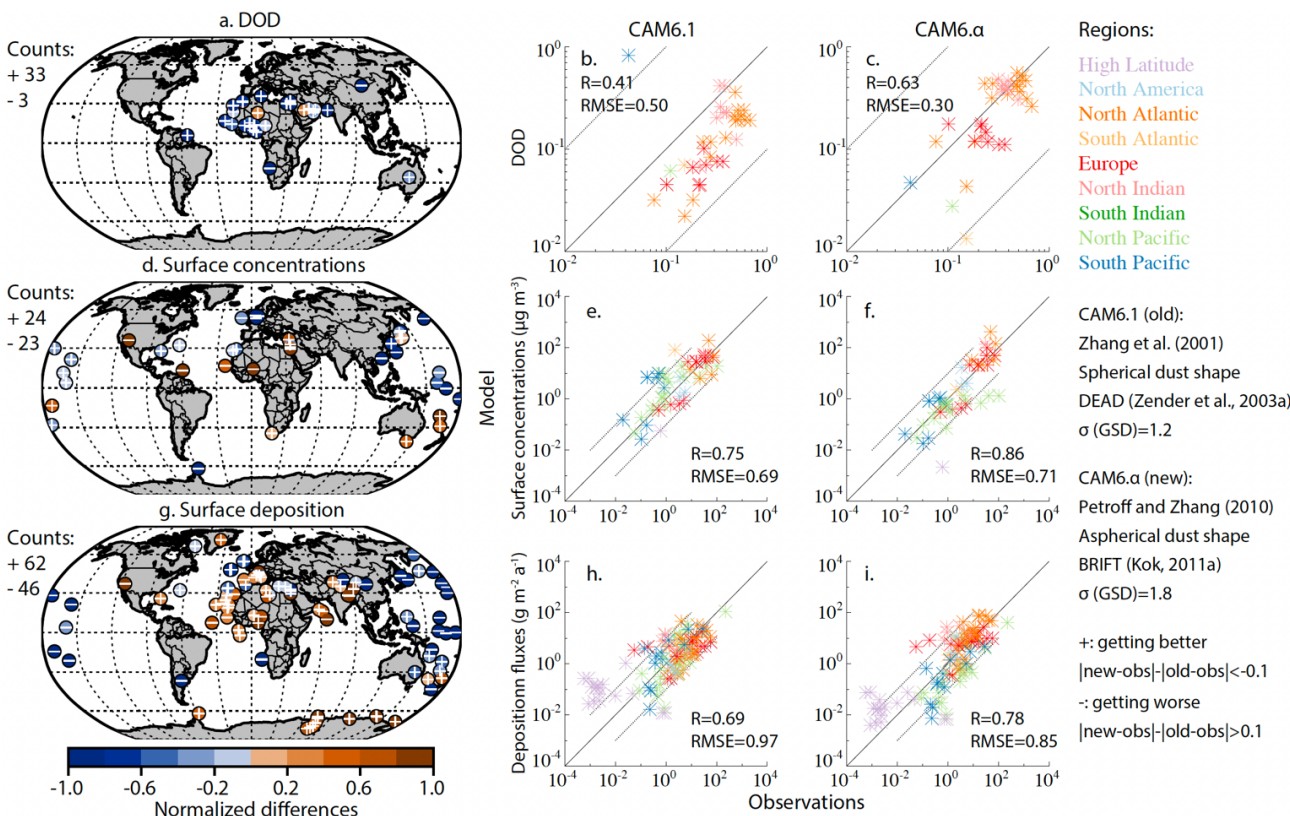

**Figure 1.** Model-observation (AERONET) comparison for DOD (dust optical depth) at the visible band centered at 0.53 μm (a, b, and c), dust surface concentrations (d, e, and f), and surface deposition fluxes (g, h, and i). Colored dots in a, d, and g show the difference between the proposed new model (CAM6.α) and observations. White symbols indicate the new model CAM6.α improves (plus sign) or worsens (minus sign) the model-observation comparison over that between the default model (CAM6.1) and observations with the

metric included in the bottom right-hand corner of the figure. Numbers listed in a, d, and g are counts of the number of improved or worsen stations. The spatial correlation coefficients between model (CAM6.1: b, e, and h; CAM6.α: c, f, and i) and observations were calculated based on the annual mean values in log space (the log of each model and observational value was taken before calculating the correlation coefficient, since the values span several orders of magnitude except DOD). Dash lines in the scatter plot show 10:1 or 1:10 lines.

### Dust emission flux rate (kg m⁻² s⁻¹)

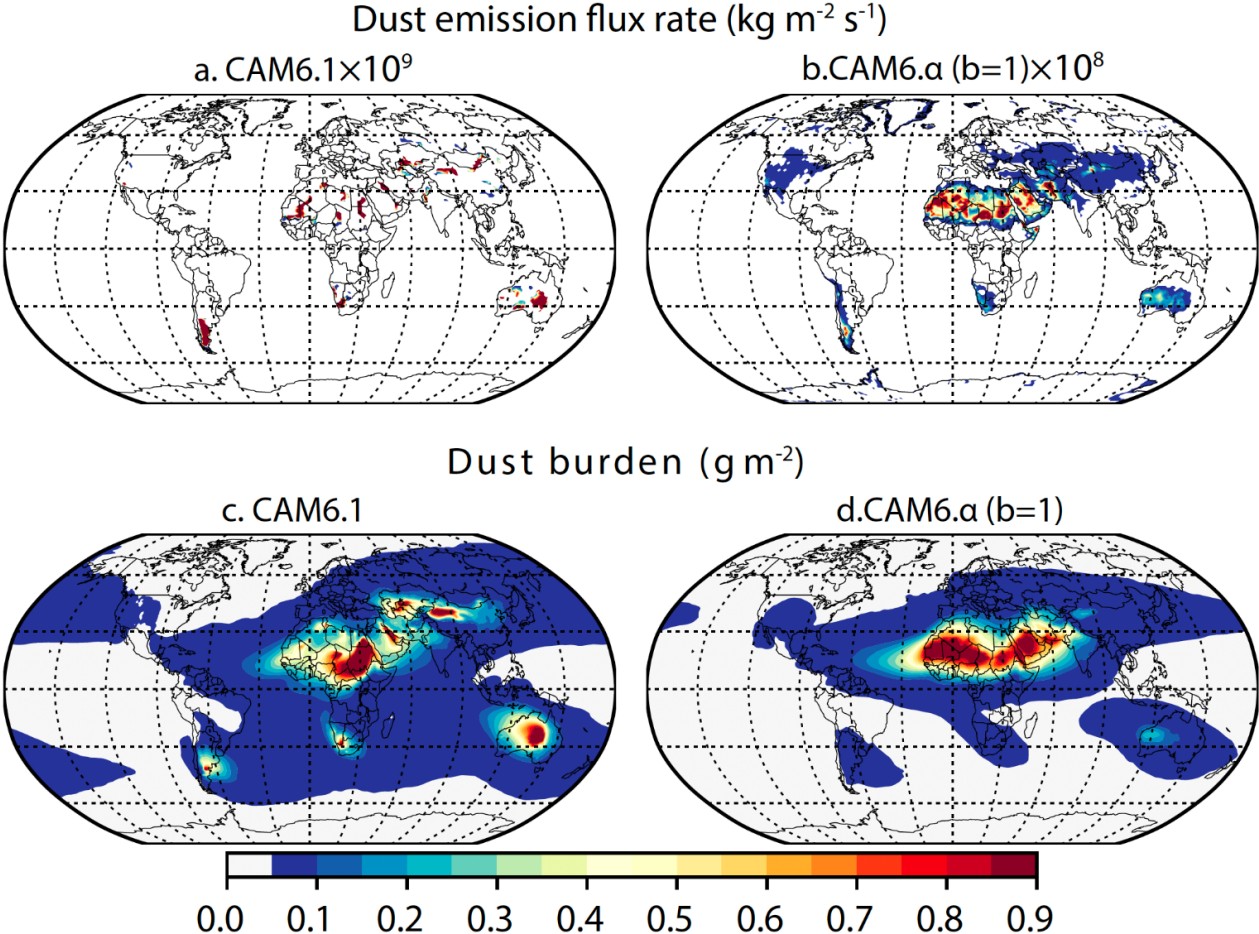

**Figure 2.** Dust emission flux rate (kg m$^{-2}$ s$^{-1}$; panels a and b) and dust burdens (panels c and d) simulated in default CAM6.1 (panels a and c; dust emission flux rate rescaled up by $10^9$) and new model CAM6.α with the threshold gravimetric water content calculated following Fécan et al. (1999) using unity tuning factor (b=1 in panels b and d; dust emission flux rate rescaled up by $10^8$ in panel b).

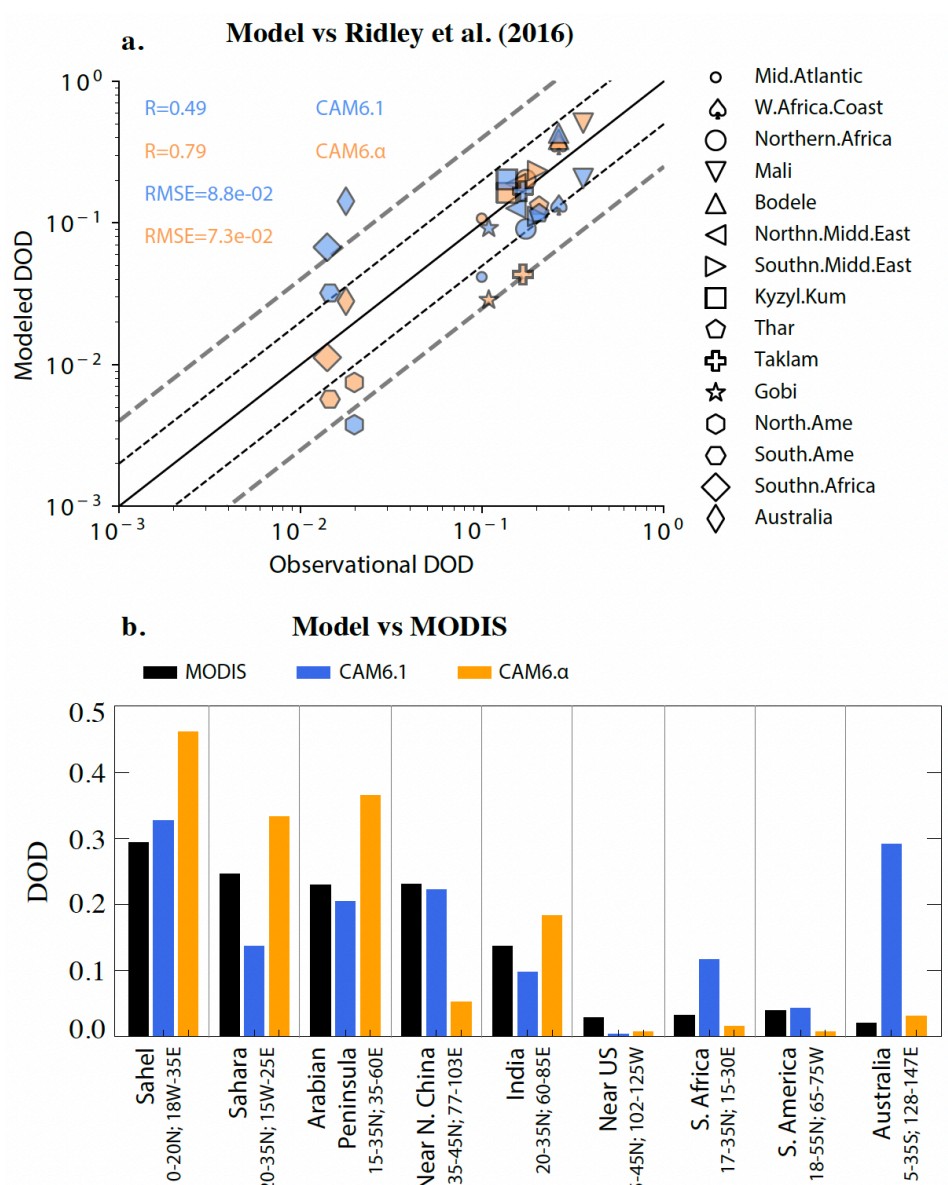

**Figure 3.** Modeled DOD in CAM6.1 (blue) and CAM6.α (orange) in comparison with that from Ridley et al. (2016) at sub regions as defined in their Fig. 1 and from MODIS retrievals (b) at sub regions (see x-axis labels). Both correlations, shown as the Kendall's τ in panel (a), are statistically significant at the 95% confidence level. Black and grey dash lines in panel (a) represent a factor of 2 and 4 differences.

Improvements are also seen if the climatologic DOD is compared to regional averages of the observationally constrained DOD in Ridley et al. (2016) (Fig. 3a). The new model CAM6.α substantially improved the modeled DOD, increasing the correlation (Kendall's τ coefficient) from 0.49 to 0.79 and reducing RMSE from 0.088 to 0.077, compared to CAM6.1. Spatially, CAM6.α better captures the regional DOD averaged over Australia and South Africa, which is consistent with comparison to the regional MODIS DOD (Fig. 3b). Over Taklamakan and Gobi deserts, however, the new model greatly underestimated the regional DOD compared to both estimates from Ridley et al. (2016) (Fig. 3a) and MODIS DOD (Fig. 3b; near northern China), whereas the default CAM6.1 works better, due very likely to lower dust emissions in the source regions in CAM6.α than in CAM6.1 (Fig. 2b versus Fig. 2a). Comparing with both datasets suggests that the new model may overestimate the regional DOD over North Africa and Middle East within a factor of two. Despite the imperfect math on the period between data and model, this overestimated regional DOD probably results from the retuning method,

which provides more credits to dust emission from North Africa and Middle East. This overestimated DOD in the model near the source regions resulting from the tuning method may also partly explain the imperfect match between the modeled and AERONET-based DOD (Fig. 1a).

    The underestimation in the surface dust concentration and overestimation in deposition occurring at several sites (near

the El Djouf; near the Antarctic from our model in all cases) is noteworthy. At some sites, such as King George in the Antarctic (62ºS, 58ºW), this phenomenon has been previously revealed by studies with multiple model ensemble mean or individual models, including an earlier version of CAM, model-data integrated study (Kok et al., 2021a), in the results of models other than CAM6.1, such as GFDL Atmospheric Model (version 2) (Li et al., 2008), and in earlier versions of CAM (Albani et al., 2014). We suggest that the phenomenon occurs likely in part due to 1) model errors in simulating

dust wet and dry deposition which is substantially larger than in simulating DOD and surface concentrations (Kok et al., 2021b): in addition to errors in and dust emissions, and the parameterization of the dry and wet deposition schemes, MAM4 in CAM6.1 represents dust transport as an internal mixture with other species (e.g., sea salt) in the accumulation and coarse modes (Liu et al., 2016), which may have unduly increased the particle size and hygroscopicity, and, thus, the removal rate (dry and wet) of dust during transport to the sites (i.e., King George); 2) the possible misrepresentation of

dust sources in the Southern Hemisphere in the model. With current emission sources, the increase of the emission rate with BRIFT from Patagonia compared to DEAD slightly mediated the underestimated dust surface concentration at King George. A further increase of the dust emission may help reduce the underestimation of dust deposition in land and the surface concentration at King George, but it would then exacerbate the bias in simulating the surface deposition at that site; 3) the limited observation period which could result in the climatology representative issue, considering the episodic

character of dust events. This limitation due to observation period may be particularly important for observed dust in the Southern Hemisphere where the dust quantities tend to be more episodic than in the Northern Hemisphere (Mahowald et al., 2011b).

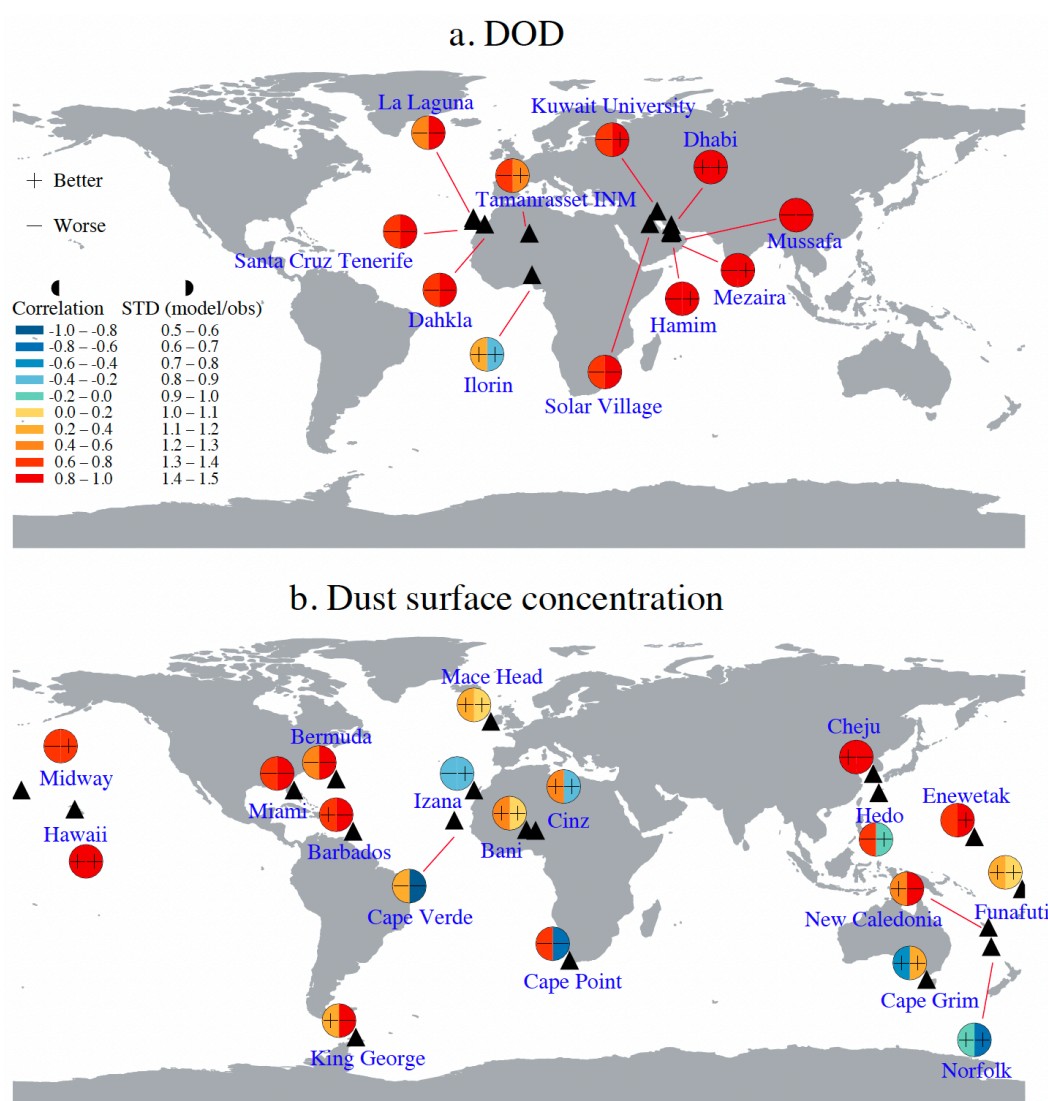

**Figure 4.** Modeling performance for the seasonal cycle of DOD (a) and dust surface concentrations (b) by CAM6.α (new model) against in situ (site names listed in the figure) measurements, relative to the performance of CAM6.1 (default model) against in situ measurements (improvement, degradation, no change indicated by "+", "-", and none characters, respectively). Colored left and right semi-circles represent Kendell's τ coefficient between CAM6.α and observations and the ratio of standard deviation (CAM6.α over observation; each normalized by annual mean values), respectively.

As to the relative importance of dry and wet deposition, we find that the dust wet deposition may dominate the total deposition of dust, especially in the remote oceanic area (Fig. S5a; Table 7), and thus affects the long-range dust transport. The models tend to overestimate the observed percentages of the wet deposition (Table 7). This overestimation could be due partly to the internal mixing assumption of dust aerosol with sea salts which increases hygroscopicity of the aerosol mixture during transport. Correcting the coarse-mode distribution, following we suggest (Table 1), does not help improve the model performance (Table 7). With that said, a recent study has shown that the CMIP6 models overestimate the precipitation frequency, particularly for the light precipitation (0.1-20 mm per day) (Na et al., 2020). It could be the same reason – the unduly simulated precipitation frequency in CAM6.1 – that explains the overestimated importance of wet deposition, compared to the observations we have here. Therefore, future model changes on the cloud physics that reduce the light precipitation frequency may help better simulate the transport of dust aerosols across zones where

frequent precipitation occurs (i.e., the ITCZ zone). Considering the limited observations on the partitioning of the dust total deposition between dry and wet processes, however, we cannot draw a concrete conclusion that CAM6.1 overestimates the wet dust deposition fluxes (Table 7).

**Table 7.** Percentage (%) of wet deposition. Observations compiled by Mahowald et al., (2011b) from data at Bermuda (Jickells et al.,
1998), Amsterdam Island, Cape Ferrat, Enewetak Atoll (R.Arimoto et al., 1985), Samoa; New Zealand sites (Arimoto et al., 1990); North Pacific sites (Uematsu et al., 1985); Greenland Dye 3 (Hillamo et al., 1993), Coastal Antarctica (Wagenbach et al., 1998), and Dome C of Antarctica (Wolff et al., 2006). RMSE: root mean square error; R: Spearman's Correlation.

| Location | CAM6.1 [RMSE=39%; R=-0.38] | NEW_EMIS [RMSE=39%; R=-0.52] | NEW_EMIS _SIZE [RMSE=37%; R=-0.63] | CAM6.α [RMSE=37%; R=-0.31] | MINE_BASE [RMSE=34%; R=-0.45] | MINE_NEW _EMIS [RMSE=35%; R=-0.29] | CAM6.α_MINE [RMSE=36%; R=-0.38] | Observations |
|---|---|---|---|---|---|---|---|---|
| Bermuda [32ºN, 65ºW] | 92 | 91 | 81 | 87 | 81 | 85 | 87 | 17-70 |
| Amsterdam Island [38ºS, 78ºE] | 88 | 88 | 73 | 81 | 78 | 80 | 83 | 35-53 |
| Cape Ferrat [43ºN, 7ºE] | 92 | 94 | 89 | 86 | 87 | 84 | 86 | 35 |
| Enewetak Atoll [12ºN, 162ºE] | 79 | 73 | 52 | 66 | 58 | 56 | 64 | 83 |
| Samoa [14ºS, 152ºW] | 91 | 91 | 83 | 86 | 83 | 81 | 85 | 83 |
| New Zealand [35ºN, 173ºE] | 89 | 92 | 82 | 87 | 80 | 85 | 88 | 53 |
| North Pacific[a] [4º-28ºE, 162º-158ºW] | 62-90 | 71-91 | 48-80 | 53-85 | 46-80 | 48-80 | 56-84 | 75-85 |
| Greenland [65ºN, 44ºE] | 82 | 87 | 82 | 86 | 75 | 86 | 84 | 65-80 |
| Coastal Antarctica [76ºN, 25ºW] | 96 | 92 | 68 | 93 | 82 | 87 | 88 | 90 |
| Dome C. Antarctica[b] [75ºN, 123ºE] | 97 | 97 | 95 | 96 | 88 | 89 | 91 | 20[b] |

[a] shown are minimum and maximum of the annual wet percent among the four sites

[b] Non sea salt-sulfate

## 5.1.3 Seasonal cycle of climatology dust optical depth and surface dust concentrations

**Dust optical depth**

Both CAM6.1 and CAM6.α reasonably reproduced the retrieved seasonal cycle at the selected AERONET sites except Ilorin (Fig. S6), where both models greatly underestimated the observed DOD in winter (Fig. S6b). It is possible that non-dust aerosols (e.g., black carbon) transported from the South Africa contaminated the observation, leading to an
artificially high DOD during the winter season at that site.

The new model CAM6.α improved both the temporal correlation based on the monthly values and standard deviation, compared to CAM6.1, only at two (Ilorin, and Dhabi) out of a total of the eleven selected AERONET sites where the measurements cover the whole twelve months in a year (Fig. 4a). Significant improvements on the modeled seasonal
cycle of DOD occurs at Tamanrasset (25ºN, 4ºE). CAM6.α increased the temporal correlation coefficient from 0.42 to

0.82 (Fig. S6e). Despite the improvement, the new model continued largely overestimating the observed DOD at this site, especially in the peak month June (Fig. S6i), resulting in an overestimated annual mean DOD.

Similar results are obtained if the seasonal cycle of DOD is compared to model-data constraints on regional DOD in Ridley et al. (2016) (Fig. S7): spatial correlation analysis on the seasonal mean DOD suggests that the new model CAM6.α substantially improved the modeled DOD in all seasons (Fig. S7) with reduced root-mean-square errors (RMSEs) and higher correlations that are statistically significant (Table S2), compared to simulations using CAM6.1 (i.e., in JJA CAM6.α: R=0.71 and RMSE= 0.085 versus CAM6.1: R=0.48 and RMSE=0.12; Kendall's τ coefficient).

**Surface dust concentrations**

In terms of temporal correlation and standard deviation for assessing the seasonal cycle, the modifications do not uniformly improve the model performance on reproducing the surface dust concentration (Fig. 4b). Only at seven out of the nineteen sites in total (a reduced number of the total sites, compared to that used in the climatology comparison, due to the removal of sites where there is no full coverage of the measurement over the twelve months in a year) - Bani and Cinz in North Africa, Mace Head in the North Atlantic, Cape Grim in Australia, and Hawaii in the North Pacific (Fig. 4b) - the modifications result in improvements using both metrics. Examining a third metric, the difference between modeled and observed surface concentration in specific months, we have thirteen of the nineteen sites where at least half a year shows improvement. Still, the new model overestimated the surface concentration of dust at many of those thirteen and other sites during most months in the year (Fig. S8-9). This overestimation is particularly pronounced for Cape Verde, likely mainly because of the strong dust emission in western North Africa using BRIFT compared to DEAD. The new model produced significant improvement in terms of all the three metrics at Bani (14ºN, 3ºE; Fig. S9i), increasing the temporal correlation from 0.21 (insignificant at the 95% confidence level) between CAM6.1 and the observation to 0.58 (significant at the same confidence level) between CAM6.α and the observation.

**5.1.4 Size distribution of transported dust**

We show the simulated size-resolved dust mass compared to AERONET retrievals and in situ measurements in Fig. 5. In general, the new model CAM6.α with the mode size distribution from CAM5 better reproduced the retrieved atmospheric size distribution than the default CAM6.1 with the size distribution from CAM6.1 over most sites. Only at 2 sites (La Laguna: 28ºN, 17ºW; and Puerto Rico: 18ºN, 67ºW) the mass size distribution from CAM6.α becomes worse than from CAM6.1. Compared to CAM6.α, CAM6.1 tends to carry more dust in mass with the diameter >~5.0 μm, which also overshot AERONET retrievals in that size range (Fig. 5). The bias in CAM6.1 could be even higher for mass of dust >~5.0 μm, considering that AERONET retrievals might have a bias towards fine dust when compared to in situ measurements (McConnell et al., 2008).

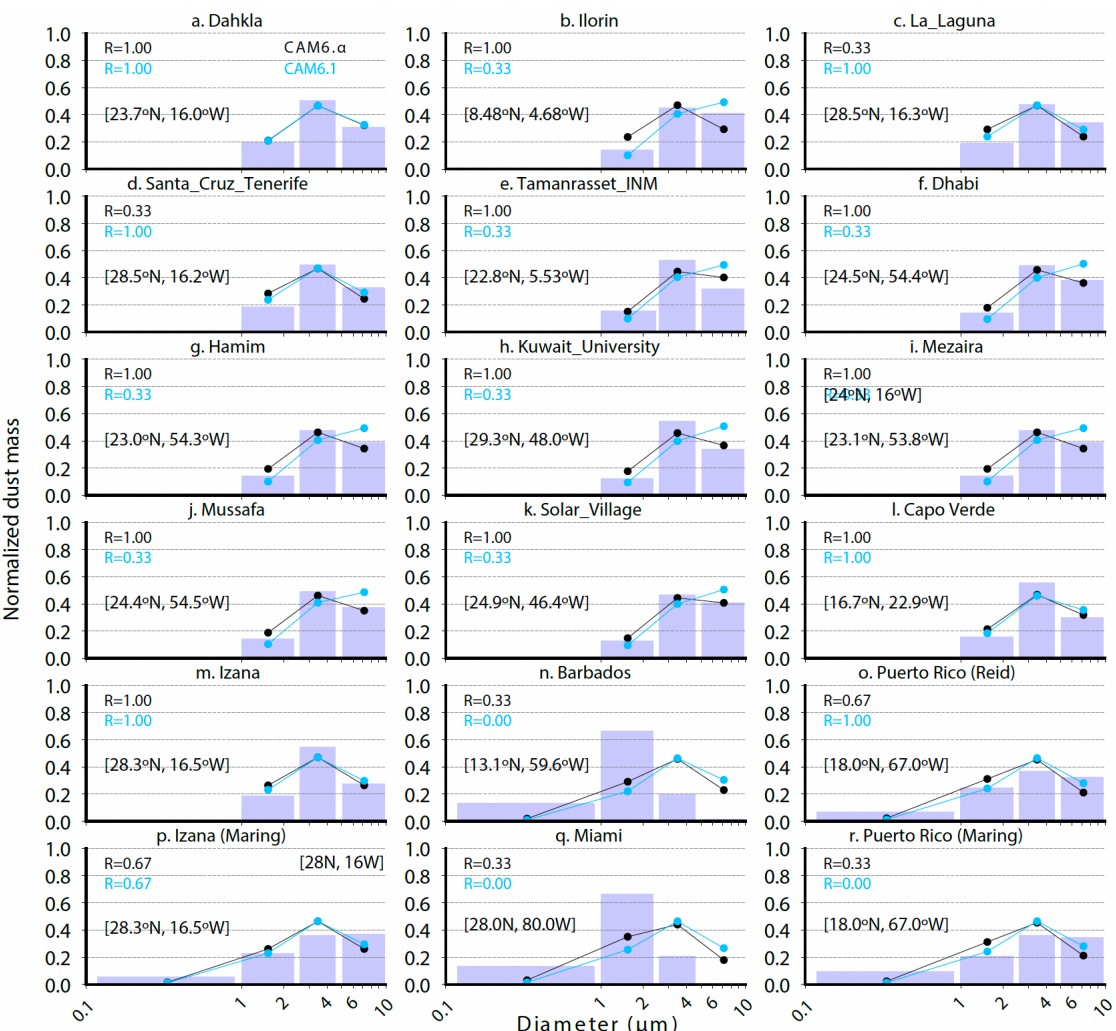

**Figure 5.** Modelled and observed atmospheric size-resolved dust mass in the geometric diameter range of 1-10 µm at AERONET
stations. Numbers in each plot indicate the Kendall's τ coefficient between model and observations (blue bars). The model runs here
include the one using the old model with the mode size parameters from CAM6 by default (CAM6.1 in cyan) and the other one using
the new model with the mode size parameters from CAM5 (CAM6.α in black). Both runs were using the offline dynamics.

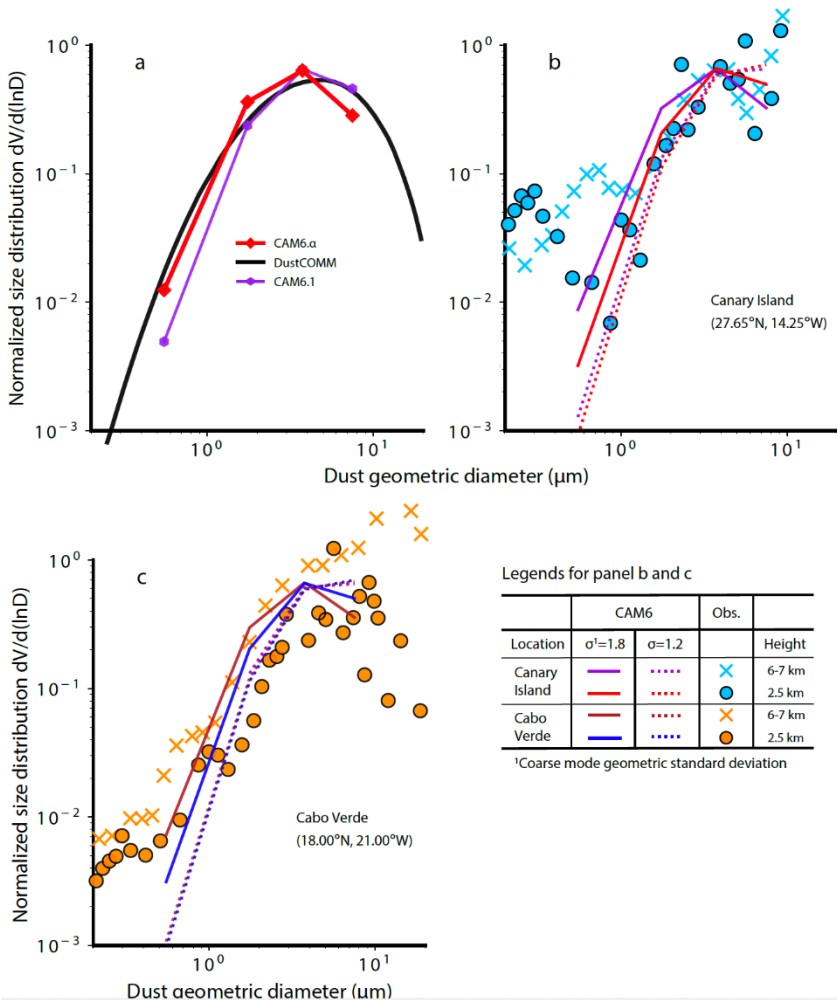

**Figure 6.** Normalized size distribution of dust between 0.2 and 10 μm diameter in the global average (a), near Canary Island (blue colors in b; dot: 2.5 km; x: 6-7 km; data for June/July 1997 from Otto et al., 2007), and near Cabo Verde (orange colors in c; dot: 2.5 km; x: 6-7 km; data for August 2015 taken from Ryder et al., 2018). The default model, CAM6.1: (purple line); the new model, CAM6.α: (red line); semi-observations: DustCOMM (black line) inverted based on an integration of a global model ensemble and quality-controlled observational constrains on the transported dust size distribution, extinction efficiency, and regional DOD with data

taken from Adebiyi et al. (2020). We chose the model layers and grid cells that are closest to the location and atmospheric height, as well as the months, where and when the measurements were made for comparison.

When comparing global mean model results to those from DustCOMM (Adebiyi et al., 2020) (Fig. 6a), generally, CAM6.α better reproduced the atmospheric size distribution (dV/dlnD) than most of the other models (e.g., WRF-Chem:

Weather Research Forecasting-Chemistry) (Adebiyi et al., 2020) in the full size range, and CAM6.1 for dust < 2 μm in diameter (Fig. 6a). Like most climate models shown (Adebiyi et al., 2020), CAM6.1 tends to underestimate coarse dust with the diameter greater than ~5 μm and the model currently excludes dust with diameter > 10 μm. The size distribution from CAM6.1 compares well with the DustCOMM result for dust > 2 μm in diameter. But it greatly underestimated the fine dust fraction (diameter < 2 μm) which CAM6.α can better capture due primarily to the more correct gravitational

settling velocity modeled by using the new dry deposition scheme.

We then evaluated the model's performance in reproducing the size distribution measurements at the high-atmosphere levels (2-5 and 6-7 km) near the Canary Island (Fig. 6b) and Cabo Verde (Fig. 6c) where Ryder et al. (2018) and Otto et

al. (2007) for transported dust (for the model-observation comparison at other atmospheric levels, see Fig. S10). Overall, CAM6.α better reproduced the size distribution at the higher atmospheric level (6-7 km) than CAM6.1, but CAM6.α substantially overestimated dust mass at the lower atmospheric level (2-5 km) compared to the measurements where CAM6.1 performed better. As also suggested in the global size distribution comparison, CAM6.1 simulated more dust > 5 µm and less dust < 5 µm than CAM6.α. However, both models underestimated the observed mass fraction of dust in that size range at the high-atmosphere level (6-7 km) near the Cabo Verde. The models also fail to capture the change in the size distribution between the two atmospheric levels that the measurements suggest. It is worth noting that the measurements are from single campaigns or flights that may have representative issues not reflecting the climatological size and vertical distributions of dust aerosols (i.e., limited by the space and time coverage).

## 5.2 Impacts of each modification on the dust cycle modeling

This section details the relative importance of each modification to the modeled dust properties (loading, and/or other dust variables). We show in Sect. 5.2.1-5.2.3 the results on the global mean and spatial distribution, and in Sect. 5.2.4 how the modifications affect the dust properties on the regional mean basis.

### 5.2.1 Dust emission schemes: BRIFT versus DEAD

The dust emission in MINE_NEW_EMIS using BRIFT (2910 Tg a$^{-1}$) is 35% lower than in MINE_BASE using DEAD (4456 Tg a$^{-1}$), primarily due to the lower DOD (0.035 versus 0.029) and higher dust lifetime in the former (3.1 and 2.2 days) (Table 6). The relative strength of dust emission for different sources also differs between DEAD and BRIFT, as Kok et al. (2011b) documented based on CAM4. The comparison between the two emission schemes here on the spatial distribution of the dust emission largely remains as in CAM4. For example, the preferential source function of Zender et al. (2003b) used in DEAD simulates most of the emission in the central part of North Africa (e.g., the Bodélé depression) (Fig. 2a). In comparison, the dust emission coefficient in BRIFT (Eq. 10) and the new method of calculating the threshold gravimetric water content of the topsoil layer (Eq. 9; see values for the tuning parameter "b" in Table 2) shifts the main dust emission in North African source westward and southward into the dust source belt (Kok et al., 2011b). This shifting in BRIFT, compared to DEAD, tends to have the dust emission to occur in the wind erodible areas that satellite-based retrievals suggest (Ashpole and Washington, 2013; Ginoux et al., 2012), though the retrieval of dust beneath clouds are unavailable which may lead to a missing of potential dust sources that satellite retrievals cannot detect, for example, dust emissions occur at the presence of deep convection (Engelstaedter and Washington, 2007; Marsham et al., 2013). The much lower dust emission in Taklamakan and Gobi deserts in China relative to that from North Africa using BRIFT, concerning that comparison using DEAD (Fig. 2b), is likely due to the high soil moisture simulated in CAM6.

Another pronounced difference in the modeled dust emission occurs in less erodible areas (i.e., North America, South Africa, Australia), where BRIFT tends to decrease the emission flux compared to using DEAD, an opposite response than that simulated for the North African sources. Such as in Australia, both schemes simulate the maximum in dust emission from the Great Artesian Basin and the Murray-Darling Basin, but BRIFT reduces the dust emission there, bringing a better agreement on the climatological DOD with AERONET observations than DEAD. However, BRIFT, using the unity tuning factor to calculate the threshold gravimetric water content, simulates high dust emissions in western Australia instead of central and eastern Australia as previously documented (Ginoux et al., 2012). Sensitivity

tests suggest that using inversed clay fraction can likely better capture the spatial emission pattern in Australia (Fig. S11). In Patagonia, as Kok et al., (2014b) found based on CAM4 simulations, using BRIFT in CAM6.1 substantially increases the dust emission compared to DEAD. In addition, BRIFT simulates the dust emission from a source in northern Chile (the Atacama Desert) and the high-latitude area, where no dust emits in DEAD.

Due to the southwestward shifting of dust emission in BRIFT to the "real" dust belt in North Africa (Sect. 5.2.1), dust aerosol particles experienced stronger vertical transport (not shown, Li et al. *in prep*) by near-surface convergence that controls the annual cycle of North African dust (Engelstaedter and Washington, 2007). The lifetime of dust thus tends to be higher for aerosol particles experiencing strong convection, which uplifts them high above the surface (Cakmur et al., 2004), increasing the dust lifetime from 2.2 days in MINE_BASE to 3.1 days in MINE_NEW_EMIS. This lifetime changing mechanism in turn indicates the importance of accurately simulating convergence-related convection (i.e., haboob) (Marsham et al., 2011) and where the dust emission occurs for dust transport modeling, especially the cross-Atlantic/Pacific (Prospero, 1999; Prospero et al., 2020) and -equatorial transport (Kok et al., 2021a; Li et al., 2008), which, currently, the models do not well represent.

In response to the change in dust emissions due to shifting from DEAD to BRIFT, the global annual mean dust deposition and loadings decreased by 35% and 7%, respectively (Table 6). Considering the lower global DOD in BRIFT than in DEAD (0.035 versus 0.029), differences between the global annual mean dust deposition in BRIFT and DEAD would become smaller, if we rescaled the global annual mean dust deposition and loadings offline using factors to make the global mean DOD in the two experiments exactly equal 0.030. The change in the total dust deposition and loading (burdens as well; see Fig. 2d versus Fig. 2c) has a similar spatial distribution (Fig. 7a versus Fig. 7b): a great increase of dust deposition and loading primarily in the Southern Ocean, the Middle East, the western Atlantic Ocean, and western USA and its downwind areas, and a great decrease in the Pacific Ocean due to reduced dust emissions in East and Central Asia (Fig. 2b); near Greenland, BRIFT simulates more dust deposition and slightly less dust loadings, owing to the local dust emission that occurs in BRIFT (Fig. 2b) but not in DEAD (Fig. 2a) and the ability of transporting further in BRIFT because of the increased lifetime of dust (Table 6).

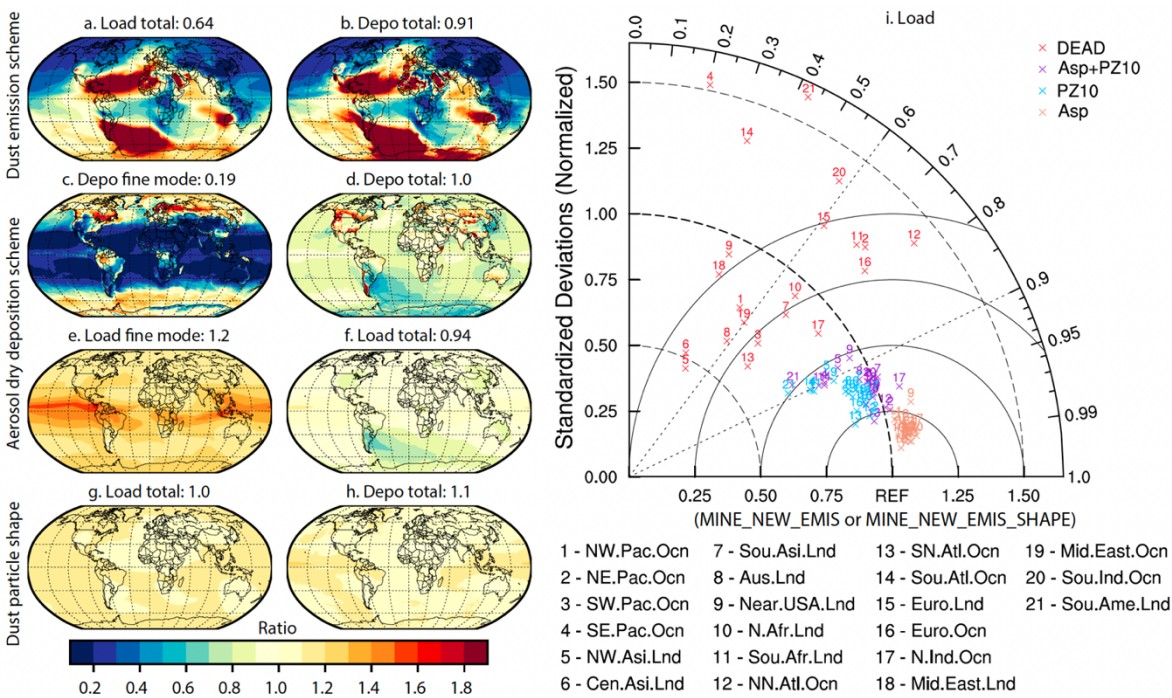

Figure 7. Impacts of the dust emission scheme (a and b: ratio of BRIFT to DEAD), aerosol dry deposition scheme (c-f: ratio of PZ10 to Z01), and dust shape (g and h: ratio of ellipsoidal to spherical dust) on the modeled dust deposition (total: a, d, and g; fine mode: c), and dust loading (total: b, f, and h; fine mode: e). The Taylor diagram (i) compares dust loading in 21 sub-regions defined in Fig. S12. In panels i, DEAD shows comparison between MINE_BASE and MINE_NEW_EMIS; Asp+PZ10 between CAM6.α_MINE and MINE_NEW_EMIS; PZ10 between CAM6.α_MINE and MINE_NEW_EMIS_SHAPE; the Kendall's τ temporal correlation and the standard deviation were obtained based on monthly values with the seasonal cycle removed.

Interestingly, we also find considerable changes to the simulated mass fraction of dust minerals between using BRIFT and DEAD (Fig. S13). These changes are as expected, given the redistributed "hot spots" where dust emission occurs by switching to BRIFT (Fig. 2) and the grid-dependency of the soil mineralogy that we used to initialize the dust speciation (Fig. 1 of Scanza et al., 2015 or Fig. S2 of Li et al., 2011). This change in the simulated mineral mass fractions of dust matters for quantifying the dust shortwave DRE at the top of the atmosphere (hematite) (Balkanski et al., 2007; Li et al., 2021; Sokolik and Toon, 1999), the cloud-aerosol interaction (feldspar) (Atkinson et al., 2013), and biogeochemistry effect (irons) (Mahowald et al., 2011a). It, thus, deserves quantifying how the shift of dust sources changes the simulated mineral content of iron-bearing minerals including hematite and illite, and feldspar in the dust.

The results suggest that BRIFT simulated ten times more hematite than DEAD in terms of mass (or volume) fraction in the Northern Hemisphere (BRIFT: 1.0%; DEAD: 0.098%) and 25% less in the Southern Hemisphere (BRIFT: 1.2%; DEAD: 1.6%). Such a decreasing of the simulated mass fraction of hematite aerosol in the Southern Hemisphere is due primarily to reduced dust emission from the Australian deserts, the soil of which enriches iron oxides (Claquin et al., 1999; Journet et al., 2014). BRIFT also shifts the dust emission westward in Australia where the soil abundance of hematite is lower than in the Australian deserts. Similarly, the increased mass fraction of hematite aerosol in the Northern Hemisphere can be partially attributed to the reduced dust emission from East Asia. The change is also evident (increase with the relative change > 30%) to feldspar in the South Ocean (Fig. S13d) and to calcite in the North Pacific Ocean

(decreasing; Fig. S13f), which may have implications for the amount of the ice nucleation by mineral dust since this
nucleation could be dominated by feldspar in mixed-phase clouds (Atkinson et al., 2013).

### 5.2.2 Dust deposition schemes: PZ10 vs Z01

The comparison of the dry deposition velocity between PZ10 and Z01 is size dependent. Because of the reduced dry deposition velocity in the fine mode, moving to PZ10 from Z01 greatly decreases the dry deposition of fine-mode (accumulation plus Aitken) dust within the low-to-mid latitude regions (between 40º S and 40º N; Fig. 7c; PZ10:Z01<0.3; similar in the accumulation mode only as Fig. S14b suggests). Since most dust mass is in the coarse mode, the small change of dust deposition in this mode, because of the slightly larger dry deposition velocity in the coarse mode in PZ10 than in Z01, results in a slight change in the total dust deposition in the low-to-mid latitude regions (Fig. 7d or Fig. S15b). Even for dust deposition in the fine mode, the increased wet deposition by using PZ10 (such as in the accumulation mode shown in Fig. S16b) offsets the reduced dry deposition in the low-to-mid latitude regions, resulting in a negligible change spatially and on global average (not shown). In the South Ocean (downwind of the Patagonian deserts), a decrease of dry deposition fluxes causes more fine-mode dust aerosol particles near the source regions, which then become cloud-borne. The increased cloud-borne particles in turn increase the possibility of horizontal transport and release of particles by the cloud droplet evaporation, leading to an increase of the dry deposition flux at the downwind regions (Fig. 7c). But the reduced dust deposition in the coarse mode dominates over the increased dry deposition flux at the downwind regions, leading to a considerable decrease of total dust deposition by >30% (relative change; Fig. 7d).

Compared to Z01, PZ10 increased the global mean dust loading in the fine mode by ~20% (Fig. 7e). Particularly in the tropics, such an increase in the remote areas can be over 60%, though the dust abundance there is low. The slight decrease of dust in the coarse mode dominates the change in the total dust loading, resulting in a slight decrease of the global mean total dust loading by 6% (Fig. 7f). Correspondingly, the global mean DOD remains almost the same between the simulations using PZ10 and Z01 (Table 6).

### 5.2.3 Dust asphericity

The overall change to the spatial distribution of dry deposition induced by dust asphericity is not as important as the change induced by changing to the dry deposition scheme PZ10. The model simulated a similar overall spatial distribution of dust deposition at the surface between modeling dust as spherical and ellipsoid shaped particles (Fig. 7h). The lower gravitational settling velocity when modeling dust as ellipsoids induced a considerable change to dust deposition only locally within remote areas: the South Pacific, western and eastern equatorial Pacific, and downwind of Patagonia, an increase of the dry deposition by up to 30% (MINE_NEW_EMIS_SHAPE versus MINE_NEW_EMIS). In comparison, little change to the dust deposition by dust asphericity occurs near/over major dust source regions. This contrast in the changes in the dry deposition flux between close-to-source and remote areas suggests that including dust asphericity could potentially mediate the overestimated dust emission from source regions (e.g., North Africa), because dust asphericity could enlengthen the lifetime in the atmosphere and thus it takes less amount of dust to have same amount of dust loadings and DOD as spherical shape assumption does.

**5.2.4 Dust size representation**

The removal rates of dust aerosol particles by both dry and wet deposition highly depends on their size (Mahowald et al., 2014). Since most of dust loadings are in the coarse mode, changing parameters of the coarse-mode size distribution ($\sigma$, initialized GMD, and the prescribed minimal and maximum boundaries within which the modeled GMD can vary, Table 1) from $\sigma$=1.2 to 1.8 halved the lifetime of dust (lifetime=4.9 days versus 2.4 days; Table 6). This reduction of dust lifetime is primarily due to the change in $\sigma$ of the coarse mode rather than the initialized GMD and its boundaries, as we obtained almost the same dust lifetime (~2.4 days) between experiments with different parameters for dust size distribution but identical $\sigma$=1.8 (NEW_EMIS_SIZE versus NEW_EMIS_SIZE_WIDTH; Table 6).

We also notice a different DOD simulated by NEW_EMIS_SIZE (DOD=0.013) and NEW_ EMIS_SIZE_WIDTH (DOD=0.019). The prescribed GMD boundaries does not affect the simulated dust loadings and DOD, because the predicted GMD in the model varies little. We can, therefore, derive that the initialized GMD itself, is also relevant to simulated DOD, but its influence (relative change=20%) is second to that of changing the coarse-mode $\sigma$. Thus, it is the increased $\sigma$ of the coarse mode that explains the reduced dust loadings (22 versus 11 Tg in NEW_EMIS and NEW_EMIS_SIZE, respectively; Table 6; Fig. 8b) and DOD (0.030 versus 0.013 Tg in NEW_EMIS and NEW_EMIS_SIZE, respectively; Table 6). This impact of changing the coarse-mode $\sigma$ is also greater than that of the other modifications (e.g., speciating dust or changing the dust emission scheme from DEAD to BRIFT) on the simulated dust lifetime which appears trivial (e.g., dust lifetime increased by 0.6 days only by changing to the new emission scheme). Correspondingly, given a similar emission rate, changing the coarse-mode $\sigma$ affects DOD most, compared to the other modifications we made.

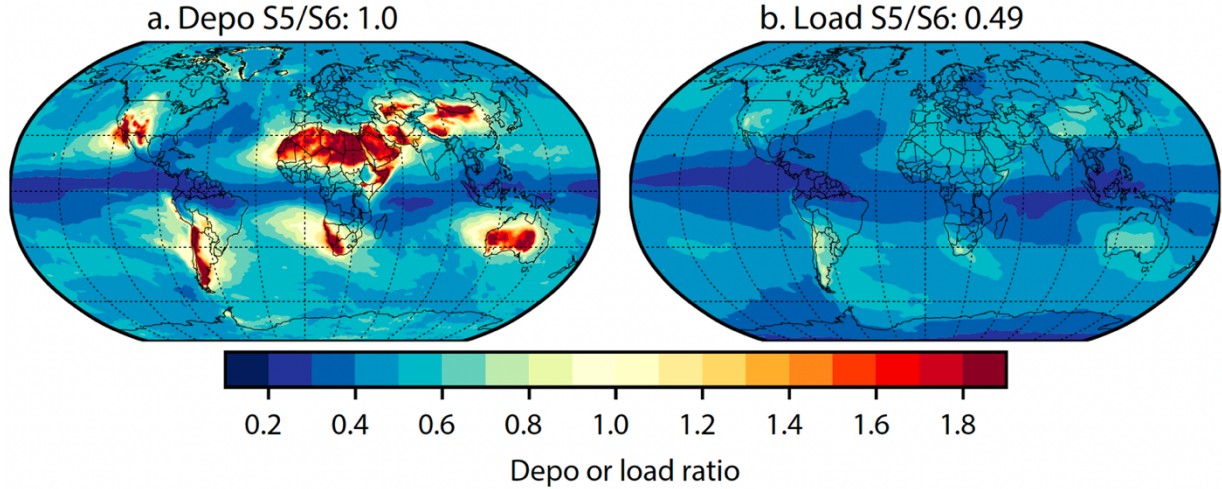

**Figure 8.** Impact of changing the coarse-mode geometric standard deviation ($\sigma$) for transported dust aerosol on the modeled dust surface deposition fluxes (depo) and column loading (load): ratio of NEW_EMIS (S5: $\sigma$ =1.8) to NEW_EMIS_SIZE (S6: $\sigma$ =1.2) (see Table 2 for case names). Numbers on the top of the plot show ratios on global average.

**5.2.5 Impacts of the modifications on the regional mean basis**

The regional analysis over 21 selected sub-regions (Fig. S12 for definition) suggests that, over most of those sub-regions, the simulated dust loading/deposition flux using the model under different modifications (PZ10, and/or dust asphericity) except to the dust emission scheme closely correlates (temporal correlation coefficient > 0.85 based on monthly values) with that in the reference case MINE_NEW_EMIS (Fig. 7i). In addition to slightly increasing dust loading, introducing

dust asphericity to the model slightly increases the temporal variability of the modeled dust loading, while replacing Z01 with PZ10 slightly decreased the variability of the simulated dust loading with respect to the reference case generally in nearly all the 21 sub-regions (Fig. 7i). The combined effect of the two modifications on this temporal variability is more determined by the choice between PZ10 and Z01 than dust asphericity.

Using different dust emission schemes changes the regional dust loading/deposition flux the most among those modifications in terms of the standard deviation (the BRIFT/DEAD ratio>1.25 or <0.75 in many regions) and temporal correlation (low-to-moderate temporal correlation between 0.15 and 0.85) (Fig. 7i). Particularly, the strong regional contrast on the dust loading/deposition exists in the northwest Asia (region 5: the BRIFT/DEAD ratio<0.5 and temporal correlation<0.5), Central Asia (region 6: the BRIFT/DEAD ratio<0.6 and temporal correlation<0.5), southeastern Pacific Ocean (region 4: the BRIFT/DEAD ratio>1.5 and temporal correlation=~0.2), and southern America (region 21: the BRIFT/DEAD ratio>1.5 and temporal correlation<0.5).

## 5.3 Dust direct radiative effect

CAM6.$\alpha$ yields a global mean net dust DRE of $\sim$-0.26 W m$^{-2}$ (shortwave plus longwave; longwave has been augmented by 51% to include dust scattering), which is slightly less cooling than in CAM6.1 ($\sim$-0.30 W m$^{-2}$). But the net dust DRE can strongly differ between the two model versions at regional scales (Fig. S17a). For example, CAM6.$\alpha$ suggests more warming (difference > 2 W m$^{-2}$ in amplitude) near Australia due to reduced dust loadings (or DOD) (Fig. 7a) and hematite mass fraction (Fig. S17a), and more cooling (difference > 2 W m$^{-2}$ in amplitude) in downwind regions of North Africa primarily due to increased dust loadings (Fig. 7a). The opposite change in one region relative to another, however, cancels out at the global scale, resulting in a negligible net DRE change (-0.04 W m$^{-2}$). The following subsections evaluate the model performance on reproducing the observed dust DRE efficiency (Sect. 5.3.1) and quantify the impact of each modification on the estimate of dust DRE and its efficiency (Sect. 5.3.2 and 5.3.3).

### 5.3.1 Dust direct radiative effect efficiency

All model versions as shown in Fig. 9 have difficulty in reproducing the shortwave dust net DRE efficiency under clear-sky conditions (Fig. 9a). In the shortwave spectral range, the new model, CAM6.$\alpha$, does not show improvement, in general. It works better in reproducing the retrievals only in the Atlantic Ocean (10º-30ºN, 20º-45ºW) in the summer and at a site in the Mediterranean basin (33.5ºN, 12.6ºW) in September. In the longwave spectral range (Fig. 9b), the dust DRE efficiency in the new model, CAM6.$\alpha$, agrees better with retrievals than that in CAM6.1, likely mainly owing to the improved representation of the dust cycle. It worth noting that, in addition to uncertainty due to the imperfect representation of the spatial distribution of dust aerosols (Fig. 1), the different spectral ranges in the model and the satellite-based sensors and radiation parameterization in the model (Jones et al., 2017) may also contribute to the difference between dust DRE efficiency from the model and observations.

All the modifications do not change the global mean longwave efficiency (Table 6), except that BRIFT yields the global mean net efficiency value that substantially differs in the shortwave spectral range compared to DEAD (MINE_NEW_EMIS: -2.1 W m$^{-2}$ $\tau^{-1}$ versus MINE_BASE: -4.0 W m$^{-2}$ $\tau^{-1}$; Table 6).

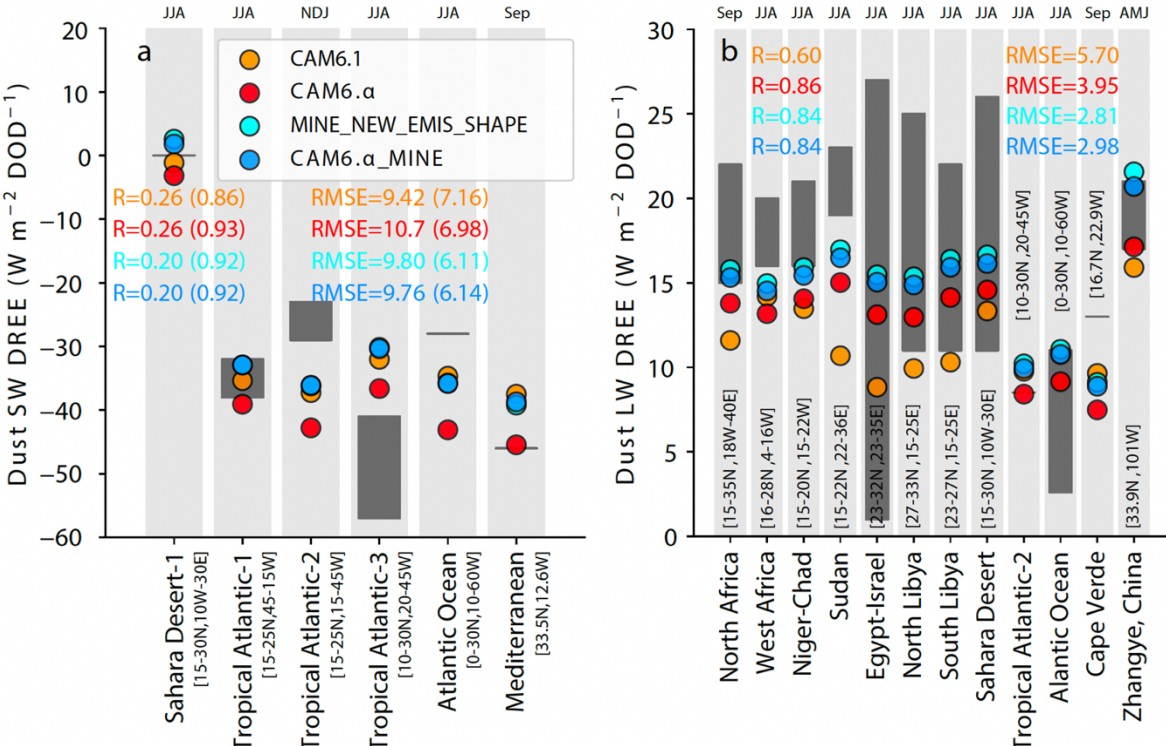

**Figure 9.** Modelled and observed dust direct radiative effect efficiency in the shortwave (SW) and longwave (LW) spectral ranges under clear conditions at the TOA over the sub-domains (shown in the inserted map and location described below) in April-June (AMJ), summer (JJA), fall (NDJ), and September (Sep) for the 2000s climate. The radiative effect efficiency is defined as the ratio of the radiative effect to DOD, so has units of W m$^{-2}$ $\tau^{-1}$. Included cases from left are CAM6.1, CAM6.α, MINE_NEW_EMIS_SHAPE, CAM6.α _MINE. Colored numbers show correlation coefficient (R) and the root mean square error (RMSE) between the model and retrievals in the SW (a) / LW (b) spectral ranges or in both spectral ranges (numbers in parenthesis in Panel a).

### 5.3.2 Impacts of dust asphericity, dry deposition scheme, and dust emission scheme

The dust asphericity introduced negligible (relative change < 10%) impacts on the global net dust DRE, and PZ10 enhanced the net dust cooling by ~18% relative to that using Z01 (Table 6). Regionally, the slightly higher/lower dust loading or DOD due to dust asphericity only slightly enhanced/weakened the warming over land (Fig. 10a; e.g., North African land; net DRE: 0.97 and 1.1 W m$^{-2}$ for MINE_NEW_EMIS and MINE_NEW_EMIS_SHAPE, respectively; the single scattering albedo at the visible bands ~0.90 for both runs, not shown) / ocean (e.g., downwind of North Africa). PZ10 simulated a slightly enhanced cooling relative to Z01 almost everywhere (Fig. 10b; e.g., south northern Atlantic Ocean, net DRE: 0.72 and 0.76 W m$^{-2}$ for MINE_NEW_EMIS_SHAPE and CAM6.α _MINE, respectively).

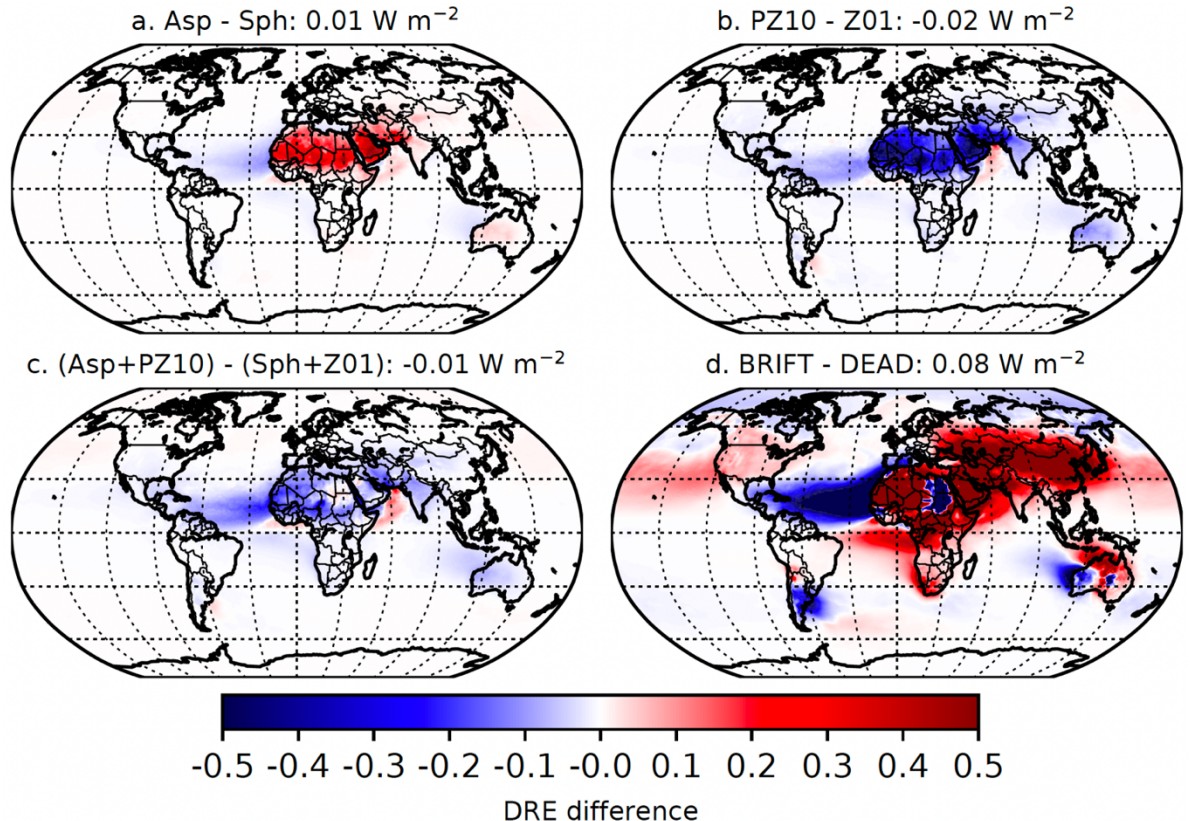

**Figure 10.** Spatial distribution of the net (shortwave plus longwave) direct radiative effect difference at the top of the atmosphere under all-sky conditions in current climate between model results using non-spherical (Asp) and aspherical dust (Sph) (a), PZ10 and Z01 (b), Asp+PZ10 and Sph+Z01 (c), and BRIFT and DEAD (d). The longwave radiative effect was augmented by 51% to account for the dust scattering. Numbers shown in each panel title represents annual mean difference in global average.

Calculations suggest a regionally strongly contrasted change to net dust DRE when shifting from DEAD to BRIFT (Fig. 10d), but the enhanced cooling in one region (i.e., the downwind Atlantic Ocean of North Africa: BRIFT: -0.76 W m$^{-2}$; DEAD: -0.64 W m$^{-2}$) and warming in another (i.e., western Africa) cancel out, resulting in a weaker global dust cooling, -0.08 W m$^{-2}$ (Table 6). These regional dust DRE differences primarily result from the regional changes to DOD/dust loadings in response to the spatial change in dust emissions, especially for non-Australian sources. Near Australia, the reduced DOD (Fig. 3) and the hematite mass fraction (Fig. S13a), which is negligible for dust from North Africa, contribute to the reduced cooling in East Asia using BRIFT relative to DEAD.

**5.3.3 Sensitivity to the size distribution**

In NEW_EMIS_SIZE, the dust DRE at the shortwave bands at the top of the atmosphere under all-sky conditions is ∼-0.39 W m$^{-2}$ (Table 6). In contrast, NEW_EMIS yields approximately 70% and 62% stronger cooling effects of -0.66 W m$^{-2}$ by mineral dust. We attribute this strong shortwave cooling in NEW_EMIS primarily to the greatly overestimated mass fraction of fine dust, which is more scattering than coarse dust. The other parameters, such as the GMD bounds of the coarse mode is also relevant to the shortwave dust DRE calculation, inducing a change of 0.12 W m$^{-2}$ (NEW_EMIS_SIZE minus NEW_EMIS_SIZE_WIDTH), which is only slightly smaller than 0.15 W m$^{-2}$ (NEW_EMIS_SIZE_WIDTH minus NEW_EMIS) resulting from the σ change (from 1.2 to 1.8) (Table 6). Compared to

its influence at the shortwave bands, the size change only slightly affected the longwave dust DRE calculation (relative change < 30%).

Spatially, differences (less cooling; absolute difference > 3.5 W m$^{-2}$) on shortwave dust DRE caused by the size change (from S6 to S5) mainly appear over areas close to the non-reflective dust source regions (e.g., ocean regions adjacent to North Africa and the Middle East, where annual surface albedo at visible band < ~0.2) (Fig. S18a). The coarse mode size change from S6 to S5 systematically reduced the longwave warming over all grid cells (Fig. S18c) primarily due to the σ

change, as the other parameters enhanced the warming effect instead (Fig. S18d).

## 5.4 Relative importance of each modification

Figure 11 compares the relative importance of each modification on the modeled dust quantities and the dust DRE at grid cell scales and on the global average. Overall, replacing the size distribution of dust aerosol and the dust emission scheme with new ones are more influential on the modeled quantities of dust (DOD, burden, and deposition) and its DRE

estimate, compared to the other modifications. At model grid cell scales, this is especially true for close-to-source regions: the size change dominates over all the others to be the most important factor in modeling the surface dust concentration which occurs everywhere (Fig. 11a); the choice of dust emission scheme is most important in modeling the dust burden (Fig. 11b) and DOD (Fig. 11c) and in estimating the dust DRE. Dust asphericity can only dominate the change to the modeled dust burden and deposition in the South Pacific Ocean (Fig. 11b, d), where the dust mass is low

relative to close-to-source regions. As for the dry deposition scheme, switching to PZ10 dominated the change to DOD in the Indian Ocean and equatorial northeastern Pacific Ocean (Fig. 11c), and to the dust lifetime at the north polar region (Fig. 11e) where the total dust is more in the fine mode for which PZ10 reduced the dry deposition velocity (Petroff and Zhang, 2010).

On global average (Fig. 11g), the size change is most important in modeling most of the dust quantities, except deposition for which the choice of the dust emission schemes becomes more influential, and in estimating the dust DRE at the top of the atmosphere.

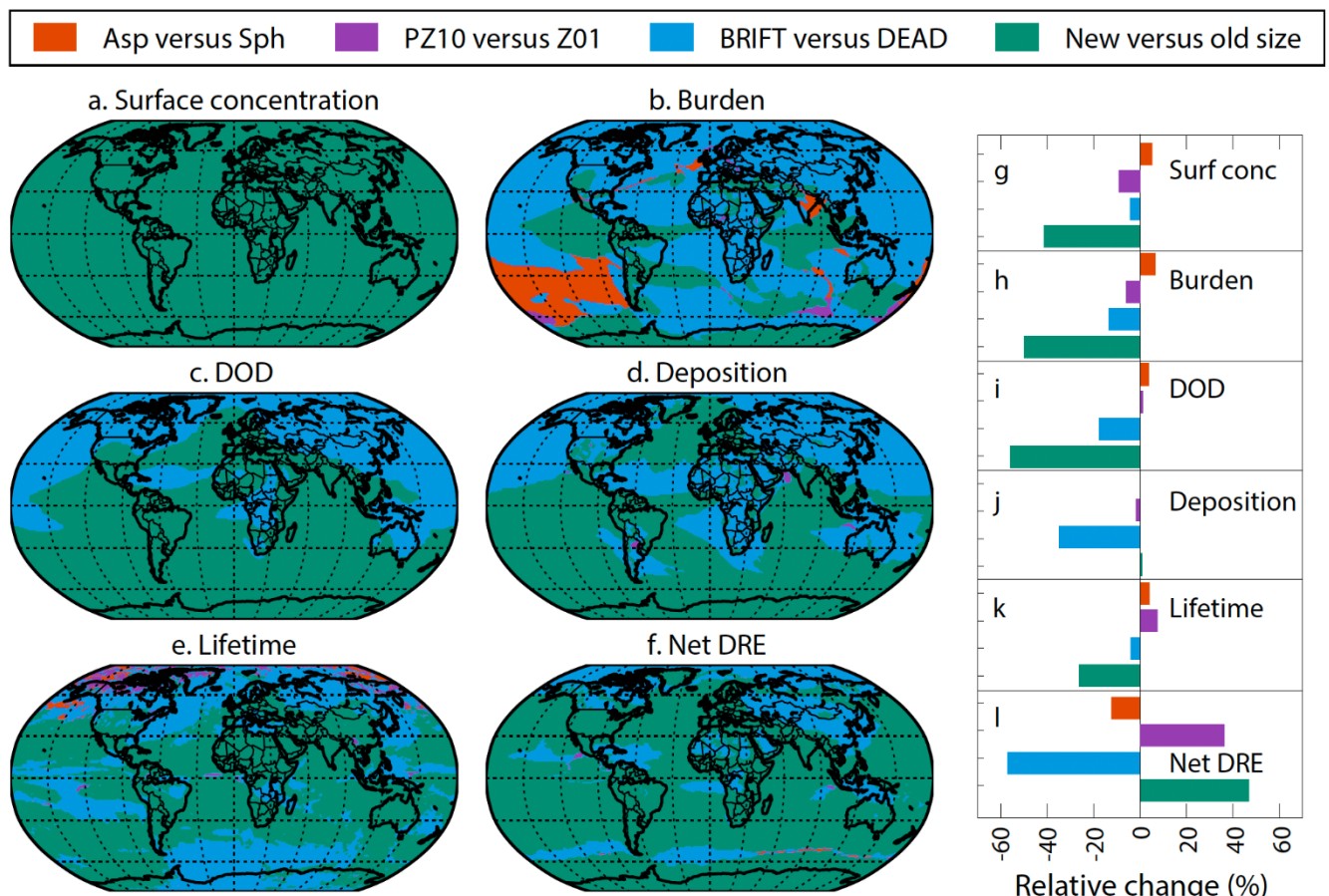

**Figure 11.** Summary of the relative importance of the modifications (spherical, Sph versus non-spherical dust, Asp; default dry deposition scheme, Z01 versus new, PZ10; DEAD versus BRIFT dust emission scheme; and coarse-mode size distribution used in CAM5, CAMS5 versus that used in CAM6.1, CAMS6) at grid cell levels (panel a-f) and in global average (panel g-i) on surface concentration (a and g), burden (b and h), DOD (c and i), surface deposition (d and j), lifetime (e and k), and net DRE (f and i) from simulations with offline dynamics.

## 6. Bulk- versus speciated-dust model

The bulk (CAM6.α) and dust-speciated models (CAM6.α_MINE) simulate a similar dust cycle with the difference between the two types of models orders of magnitude smaller than the dust cycle itself modeled either by CAM6.α or CAM6.α_MIN (e.g., Fig. 12 and 13). This similarity results from several factors:

1) tuning the dust cycle to a global mean DOD of 0.03;

2) nudging both models towards the same meteorology dynamics;

and 3) conserving the dust mass when speciating the dust-aerosols such that summing the mass fraction of each dust species equals unity. For the same reasons, the influence of each of the modifications on the modelled dust cycle quantified using the bulk model instead of the dust-speciated model, as this study used, would be similarly comparable.

What differs remarkably is the modeled dust optical properties between the speciated- and bulk-dust simulations. For example, the speciated-dust model (CAM6.α_MINE) yields a lower global-mean dust SSA than the bulk-dust model

(CAM6.α): 0.896 versus 0.911 (Table 6) at the visible band centered at 0.53 μm. Note that the dust DRE is sensitive to variation of the dust SSA. This lower dust SSA obtained here in the dust speciated model than in the bulk dust model is consistent with the finding of a previous study (Scanza et al., 2015) using an earlier model version (CAM5). Correspondingly, CAM6.α_MINE yields a reduced dust cooling (Table 6) and DRE efficiency (Fig. 9) relative to CAM6.α.

For dust DRE efficiency (Fig. 9), speciating dust in CAM6 tends to reduce the RMSE while retaining the horizontal spatial correlation in either SW (CAM6.α: RMSE=11 W m$^{-2}$ $\tau^{-1}$; R=0.26 versus CAM6.α_MINE: RMSE=10 W m$^{-2}$ $\tau^{-1}$; R=0.20) or longwave (CAM6.α: RMSE=4.0 W m$^{-2}$ $\tau^{-1}$; R=0.86 versus CAM6.α_MINE: RMSE=3.0 W m$^{-2}$ $\tau^{-1}$; R=0.84) or both spectral ranges (CAM6.α: RMSE=7.0 W m$^{-2}$ $\tau^{-1}$; R=0.93 versus CAM6.α_MINE: RMSE=6.0 W m$^{-2}$ $\tau^{-1}$; R=0.92). This comparison suggests that modeling dust as component minerals with the dust size distribution in coarse mode of MINE_NEW_EMIS_SIZE helps improve the model performance relative to modeling dust as a bulk to reproduce the retrieved dust DRE efficiency (Fig. 9a).

The improvement, however, could be artificial because of the combined use of imaginary part of the complex refractive index of hematite (see Fig. 1b of Li et al., 2021) and the volume mixing rule used in the dust speciated model to compute the bulk-dust complex refractive index (Li et al. *in prep.*), leading to artificially more absorptive dust than in the bulk dust model (Fig. 9a and Table 6).

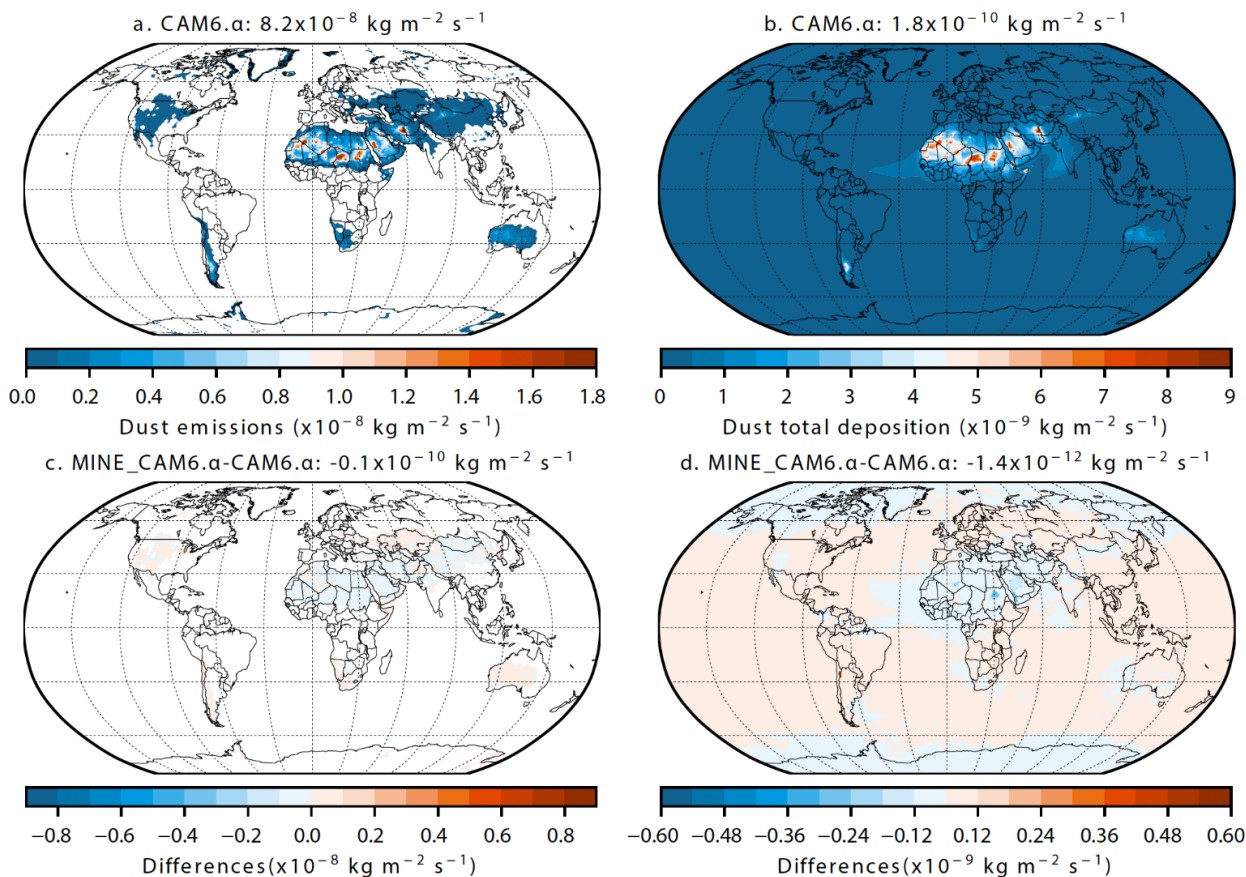

**Figure 12.** Surface dust emissions (a; global annual mean=2891 Tg) and deposition fluxes (b; global annual mean=2893 Tg) simulated by CAM6.α and their differences (c and d; both global annual mean=22 Tg) between CAM6.α_MINE and CAM6.α.

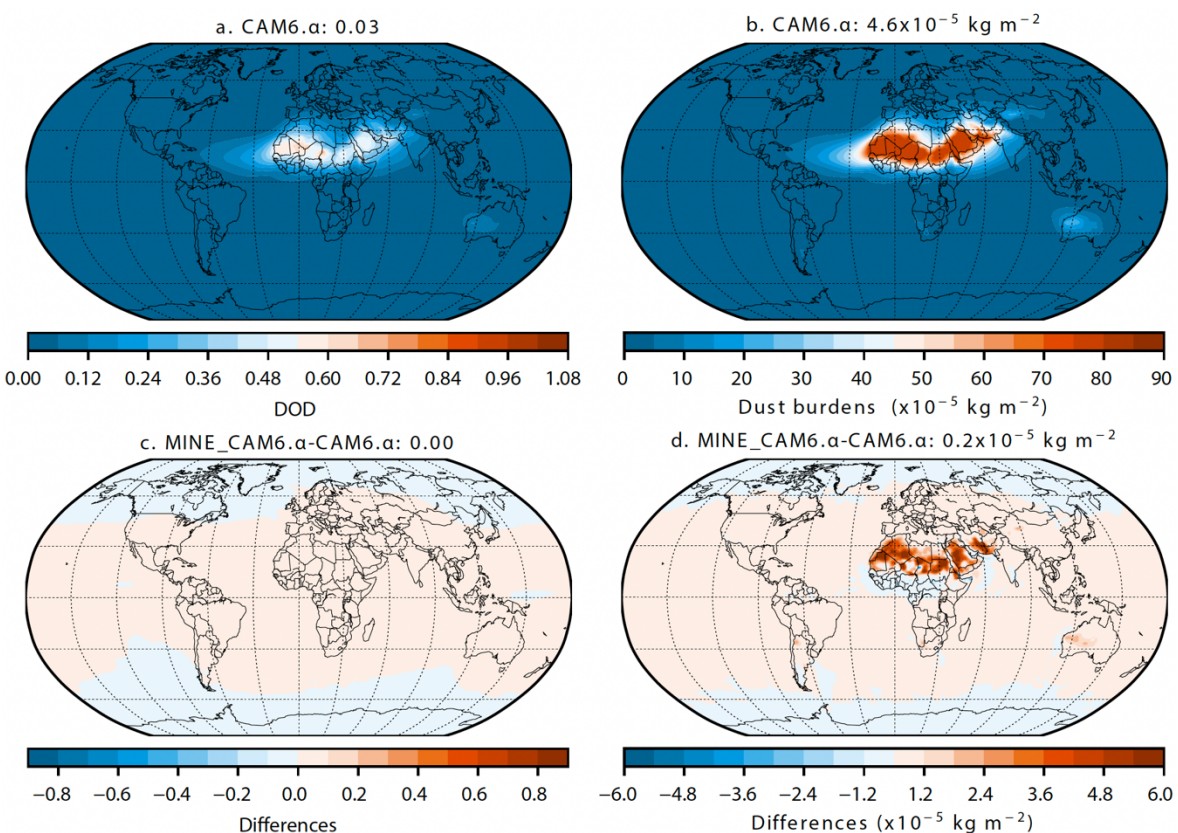

**Figure 13.** The same as Fig. 12 but for DOD (a: global annual mean=0.030 and c: global mean difference=0.001) and dust burdens (b: global annual mean of dust mass=24 Tg and d: global mean difference≈0 Tg), respectively.

## 7 Limitation in the model-observation comparison

There are issues which may affect the model-observation comparison, when interpreting the comparison:

1) the period when the measurements were made not perfectly matching when the simulations were performed for;

2) different representative space volume between the model results and observations; the model results are representative of a colocation in space which is determined by the spatial resolution and often too large compared to the volume that observations represent (Hamilton et al., 2019; Wang et al., 2014). Ground stations measure dust-related quantities using stationary instruments, and aircraft-onboard instrument measures dust along with the flight track;

3), some observations include dust of size $> 10$ μm in diameter (between 10-20 μm; dust particles in this size range are also present over the source regions and regions downwind of North Africa as Ryder et al. found in 2019, but nearly all the observational constraints used in this study do not include those dust particles) which our models do not simulate; this might be an important error source (Adebiyi and Kok, 2020a). On the other hand, the observations of $PM_{10}$ are likely to include only $PM_{6.9}$, because what measured is in aerodynamic not geometric diameters (Huang et al., 2021; Reid et al.,

2003b). Finally, the modelled dust mass is for dust with our own defined mineralogy composition only (Li et al., 2021; Scanza et al., 2015), but the measured mass could likely also include non-dust particles, such as sea salt (Kandler et al., 2011; Zhang et al., 2006), sulfate (Kandler et al., 2007), biomass burning aerosols (Ansmann et al., 2011; Johnson et al., 2008), or other air pollution aerosol (Huang et al., 2010; Yuan et al., 2008). This contamination of non-dust aerosols on the measurement is especially for dust in the fine-mode size where the instrument cannot distinguish dust from the fine-sized non-dust aerosols;

4) some of observations were not made for a period long enough to be taken as a representative of climatology (see Table 3-5); also considering point 1), the model-observation comparison may be subject to change because of interannual variability or the episodic character of dust aerosols (Li and Sokolik, 2018b; Mahowald et al., 2011b);

5) uncertainty in the measurements. In addition to contamination of non-dust aerosols on the measurement of dust, there is also uncertainty due to assumed dust shape and complex refractive index to derive dust size, particularly for particles > 1 µm (Laskin et al., 2006), and error in AERONET AOD retrievals (i.e., the cloud-screening algorithms; Levy et al., 2010) and in the method used to filter out the contribution of non-dust aerosols; note difference exits between clear-sky from observations and all-sky AOD/DOD from the model and aerosol models but the difference is not a considerable error source (tested; not shown);

and 6) the method of selecting AERONET sites may introduce uncertainty because of the possible mismatch between simulated and observed AOD for both dust and non-dust aerosols.

## 8 Concluding remarks and outlook

This study compares how different modelling representations of the dust emission schemes, the aerosol dry deposition schemes, transported dust particle size distributions, and the dust shape treatments affect the modeled dust cycle in CESM2.1-CAM6.1. We evaluated model performance using different combinations of those modifications using offline dynamics by comparing the modeled dust properties (DOD, dust surface concentrations, dust deposition fluxes, atmospheric size distribution of transported dust, and dust direct radiative efficiency at the top of the atmosphere) that are related to the dust lifecycle with (semi-) observations in the current climate. Since the new more physically based dust emission scheme shows substantial improvements on the model-observation comparison and the updated aerosol dry deposition scheme corrects the overestimated fine-mode deposition velocity, future model developments will be focused on introducing both these features into a future official CAM version for the benefit and use of the whole community. Results of this work therefore inform modelers how well these new features will improve model performance in reproducing the dust cycle in CESM.

Our analysis suggests that reverting the geometric standard deviation of the transported dust size distribution (coarse mode) from the default 1.2 to 1.8 imposes the most important change among what we introduced to CAM6.1 to the modeled dust cycle, though the linear assumption between DOD and the other dust quantities based on which we rescaled up the concentrations, deposition, burdens, and DRE of dust in the size distribution simulations. Since the defaulted 1.2 is too narrow to simulate the dust lifetime, in the next released model version, we recommend reverting the geometric standard deviation to 1.8, as in CAM5. This reverse may require a split of representation of dust and the

stratospheric aerosols in the coarse mode, for which the narrow coarse-mode size distribution works better (Mills et al., 2016), and some changes to sea salt.

With the global DOD similarly comparable in different cases because of the retuning we applied or slight impacts by the updates on DOD, the modifications on dry deposition and emission schemes, as well as the gravitational settling due to
dust asphericity only slightly changed the simulated seasonal dust loadings/burden/DOD and deposition. However, regionally, large difference among different model results for dust loadings/burden/DOD and deposition are found. These stem either from the choice of the dust emission schemes (BRIFT versus DEAD) or the width of the coarse-mode size distribution. Consequently, it is due primarily to the inclusion of the new dust emission scheme but not use of the new dry deposition scheme and accounting for dust asphericity that the new model, CAM6.α, shows improvements. It is
worth noting that the results obtained in this study rely on the models with the offline dynamics, which is subject to change while using the predicted meteorology field online.

Overall, with the offline dynamics, the new model, CAM6.α:

1) can better capture the climatology and seasonal variation of DOD at more observational sites than the default model, CAM6.1, bearing in mind the uncertainty in the measurement and in the way that we did the model-data comparison;

2) results in a dust DRE that is regionally substantially different from CAM6.1 (i.e., stronger warming over most land areas except over South America and stronger cooling over the North Atlantic Ocean; Fig. 10d). Though the opposite
change to dust DRE in one region to another partially cancels, (Fig. S17), its influence on the global mean dust DRE remains large (relative change > 55%; Fig. 1l).

Still, there exists large uncertainty in modeling the global and regional dust cycle in comparison with observations. This large uncertainty could partially result from the constants used in the parametrizations that affect the dust emission and
transport processes, such as the critical LAI threshold, the hygroscopicity of dust, and the prescribed scavenging coefficient, though the default values in the model have been used during the past decade in CAM of different versions. In addition, further development and studies focusing on the following processes and dust properties, which the current model does not represent well or omits entirely, may be helpful for further improving the simulation of the dust cycle in CESM:


1) for the dust emission parameterization, the threshold friction velocity calculated in both BRIFT and DEAD does not account for the spatiotemporal variability of the soil properties (i.e., soil grain size distribution and aggregate state; Leung et al., 2021; mainly limited by the sparse information; Kok et al., 2014b) in addition to the soil moisture. The current dust module in CAM6.1 also does not consider the roughness effect due to the presence of non-erodible elements
(i.e., rocks and pebbles) on the threshold velocity calculation (Marticorena and Bergametti, 1995); also, crusted surface layer present at the erodible surface can greatly reduce the wind erodibility by increasing the particle cohesion, and, thus, the dust emission rate, compared to the surface that does not consist of consolidated aggregates (Rice and McEwan, 2001; Rodriguez-Caballero et al., 2022);

2) the models used here did not simulate anthropogenic dust emissions due to human activities (i.e., agricultural practices, such as overgrazing, and fugitive dust from roads and construction), which may constitute a considerable fraction of the total dust emissions (Ginoux et al., 2012). This could likely be a reason for the underestimated dust emission in the northern high-latitude regions (Sect. 5.1.1), for instance, at the Moscow metropolitan area (~56ºN, ~37ºE), one of the most significant northern high-latitude sources generated on paved roads and roadside soils (Kasimov et al., 2020), which the current model does not include.

3) comparisons with the constrained global dust size distribution and measurements downwind of North Africa suggest that the model underestimates dust aerosols in the coarse mode with the geometric diameter > 5 µm and misses aerosol particles with the geometric diameter > 10 µm (Fig. 6). The former happens likely due to an underestimate of dust aerosol particles in that size range upon emissions and/or the removal rate of those particles being too high during transport in the model (Adebiyi and Kok, 2020b; Meng et al., 2022), the reason for which is still under exploration. For the latter, extending the dust size range to include particles with the geometric diameter > 10 µm in CAM6 is a worthy endeavor, such as in Ke et al. (2022).

4) as previously noted (Wu et al., 2018), some of the variables in the dry deposition parameterizations could vary in different seasons for certain land cover and land use types, such as the roughness length, $Z_0$, in Z01 and the displacement height of the canopy, h, in PZ10, for which a fixed climatological mean is used in the models. How accounting for the seasonal variation of those variables in the model can affect the dust cycle modeling deserves further exploration.

5) compared to bulk dust, modeling dust aerosol as component minerals could better reproduce the observed spatiotemporal variability of dust optical properties and thus the dust DRE efficiency (Fig. 9) with the offline dynamics in the present day. But the current atlas of soil mineralogy and the optical properties of key minerals (i.e., iron oxides) contain large uncertainties which should be better quantified in the future, such as that planned for the Earth Surface Mineral Dust Source Investigation (EMIT) and in our ongoing work (Li et al., *in prep*).

The comparison of modeling the global and regional dust cycle with observations itself is limited by the spatial and temporal coverage of observations, especially for high-latitude dust, particularly dust in the Southern Hemisphere. More intensive measurements on concentration, deposition, atmospheric loading, shape parameters, size distribution, and optical properties of dust aerosols at varied spatiotemporal scales would also help better represent dust and project climate changes in the global climate models.

**Data and code availability**

The updated model code, key model results, and observations are available in a publicly accessible repository (https://doi.org/10.5281/zenodo.6989502). DustCOMM data are available at https://dustcomm.atmos.ucla.edu.

**Author contributions (will complete this later)**

LL and NMM designed the study. LL updated the CAM6.1, performed all the simulations, analyzed the model results with comments from NMM and JFK, and wrote the original manuscript with comments from NMM. XL and MW provided the new aerosol deposition code. All authors commented and edited the manuscript.

**Competing interests**

The authors declare that they have no conflict of interest.

**Acknowledgements**

We thank the two anonymous reviewers for their constructive comments that helped to improve the manuscript, and thank Dr. Paul Ginoux for his helpful comments. LL, NMM, and DSH acknowledge assistance from the Atkinson Centre for a Sustainable Future, Department of Energy (DOE) DE-SC0021302. A portion of this work was also supported by the Earth Surface Mineral Dust Source Investigation (EMIT), a NASA Earth Ventures-Instrument (EVI-4) Mission. XL and MW were supported by NASA CloudSat and CALIPSO Science Program (grant NNX16AO94G/80NSSC20K0952). MW was also supported by the U.S. DOE, Office of Science, Office of Biological and Environmental Research, Earth and Environmental System Modeling program as part of the Energy Exascale Earth System Model (E3SM) project. The Pacific Northwest National Laboratory (PNNL) is operated for DOE by the Battelle Memorial Institute under contract DE-AC05-76RLO1830. LL and NMM acknowledge the high-performance computing resources from Cheyenne provided by NCAR's Computational and Information Systems Laboratory (CISL), sponsored by the National Science Foundation.

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
