# Peer review of "Importance of different parameterization changes for the updated dust cycle modelling in the Community Atmosphere Model (version 6.1)"

_Geoscientific Model Development, 2022_

## Referee Comment (RC1)

**Comment on *Importance of different parameterization changes for the updated dust cycle modelling in the Community Atmosphere Model (version 6.1)*, by Li et al. 2022.**

**General comments**

This article presents multiple developments included in the dust cycle representation within the CAM6.1 model and assesses their impact on relevant variables, such as the dust surface concentration, deposition, size distribution, optical depth and direct radiative effect. The work conducted provides relevant information beyond the dust modeling community, as dust has impacts on different features of the atmospheric dynamics and chemistry, the climate and the Earth System. As such, I believe this article is well within the scope of the Geoscientific Model Development journal, it presents novel results, and it deserves publication.

However, in my view, in its current form the reader has to put in a considerable effort to follow the details of the massive amount of work presented.

The authors present nine different experiments: five defining dust as a bulk species and four experiments considering speciated dust. This involves a duplication of experiments in which one (or several) of the new developments are tested, and adds an additional variable to the analysis, making it harder to focus on the specific impact of the new aspects included in the model.

In order to lighten up the contents of the paper, I would recommend splitting the results in two different articles, one focusing on the current developments and their impact on the bulk dust cycle, and another focusing on those improvements that potentially have an impact on the mineralogy (e.g. the changes on the emission scheme).

With respect to the experiments design, the authors could better clarify the criteria used to include the new features in the tests. Instead of relying on a baseline (e.g. CAM6.1), and adding separately to that configuration the different developments (on the emission scheme, dry deposition, size, or asphericity), the authors combine multiple developments in the different experiments. I believe these combinations could hinder a clean comparison of the effect of each development (e.g. looking at Table 4 it is difficult to know which pair of experiments allows disentangling the effect of shape and deposition changes). This issue is accentuated by the fact that the experiments are referenced along the manuscript by different names or acronyms, which further complicates tracing them.

Then, I believe that a fundamental piece of this article is the variety of observations, retrievals, model-derived products and model results that are used for the model evaluation. The modelling community could greatly benefit from the effort done here to compile that information and produce a benchmark for dust properties evaluation at the global scale (in present climate). Unfortunately, these are only presented in the article in a summarized manner (through a table). I would recommend adding in the manuscript at least a discussion on the variables available, their usefulness for modelled dust evaluation and their limitations.

Finally, I would recommend modifying the organization of some of the contents, and re-writing or improving some parts of the text. Also, in some sections, the authors rely excessively on external references, making it difficult to follow the discussion with the information provided in the paper itself. My recommendation would be to restructure or adapt the article contents, such that:

(1) the previous status of the model is clearly defined and the motivation to improve or change the specific dust representation is justified.

(2) the new developments are described in the current paper in a comprehensive manner (i.e. not trusting excessively on the reader to go and check the external references).

(3) the evaluation methodology is explained before the presentation of results, for instance adapting current section 3. It would be particularly useful to identify the multiple metrics that are going to be used for the model evaluation and their purpose (i.e. regional variability, temporal variability, etc.), comment on the dust tuning methodology and its impact on the evaluation metrics (if any), as well as to merge the description of the observations with the comments on section 5 about the limitations of the datasets. Section 5 could be kept to provide an overall assessment of the observations limitations on the main conclusions of the article.

I believe that with these changes, the article would be much easier to follow and it would reach a broader audience.

**Specific comments**

**Introduction**

I believe this section could be slightly re-structured, particularly to better clarify the current model status, justify the need for improvement in the specific aspects that are dealt with in this work, and briefly explain how these are going to be approached.

**2. Model descriptions**

I would recommend starting by describing the aerosol representation in CAM6.1, as it affects both bulk dust, speciated dust and other aerosols simulated in the model.

Please, see my general comment above. Which is the added value of conducting two set of simulations (with bulk and speciated dust) for the purpose of this article (assessing changes due to deposition, emission, size distribution and shape)? If this is not justified, I would focus on this article in the bulk dust experiments, and present the speciated dust experiments elsewhere.

**2.3. Dust optical properties and radiation flux diagnostic.**

Please, take advantage of this section to explain aspects related to the calculation of optical properties and/or radiative variables that are currently explained in the results section (see my comments below on sections 4.3 onwards).

**2.4.2. Dry deposition schemes.**

The original dry deposition scheme is partly described here and partly in the introduction. I would use this section to describe the details on both the previous and the new proposed scheme. At least, I would include here the references to both schemes, and clarify if the empirical coefficients are updated in the new scheme.

**2.4.3. Dust asphericity**

Being this one of the developments listed in the article, it would be worth to include in this section at least the main characteristics of the development (e.g. factor varying according to the source region, and ranging from X to X).

Also, the authors mention the impact of the dust asphericity on optical properties (line 119). In section 4.2.3, they state that CESM2 does not include the enhancement in mass extinction efficiency due to asphericity, but that it is considered in this study (section 4.2.3). I believe the approach used to consider asphericity in the mass extinction efficiency should be clarified and described in this section (2.4.3).

**2.5 Experiment design**

Please, see my general comments related to the experiments' design.

I would recommend to describe first the common model configuration amongst experiments (i.e. configuration of the model components, spatial resolution, period simulated, etc.), and then identify the experiments designed to test the different developments.

**3. Observational datasets for model evaluations**

Please, see my general comment above.

Table 3 now includes a list of different observations, retrievals, and products using a combination of models and observations/retrievals. I believe some of the information of the table, and particularly that included in the column "Comments", is relevant enough to be included in the main text. It would be also helpful to relate the different datasets to the limitations identified in Section 5. If possible, in the Supplement organize the observations by type, e.g. do not mix surface dust concentration with dust optical depth retrievals.

The DustCOMM reference and the models included in the Adebiyi et al. (2020) paper could be mentioned here, as they are used in section 4.1.4 to compare the model size distribution.

**4. Results**

Please, review and re-structure this section, see my general comment above.

I believe using the same set of experiments to discuss all the modifications (either bulk or speciated dust) would help. In addition, a discussion focusing on the different variables, combining the multiple datasets used as a reference, rather than a separate explanation for each comparison could be of benefit. Another strategy to make easier the discussion for the reader, could be to "qualify" the sites / observations by their characteristic trait when explaining the details, e.g. source region, remote station, etc., rather than leaving it to the reader to figure out where the site is or its characteristics.

**4.1.1. Dust emissions**

Why compare the total emission burden with model estimates that go beyond CAM6.1 simulated size range? I believe it would be useful to include comparisons with models that use the same range (e.g. some of the AEROCOM phase I models, Huneeus et al. 2011).

**4.1.2. Climatology annual means of [...]**

The discussion here will greatly benefit from a previous definition of the statistics, metrics, and evaluation, which I would suggest including in Section 3. In that way, the authors could make the discussion in this section lighter.

The authors mention the tuning as a factor affecting the comparison of modelled DOD to MODIS and Ridley et al. (2016) products; however, this is not taken into consideration when AERONET information is used as a target. Could the tuning also have an effect on those results?

Does the dust wet vs. dry deposition balance in their model change with the improvements on size distribution? Could this partly be explained by an overestimation of the finer dust fractions? Or is the representation of modal internal mixtures more relevant to this process?

**4.1.4. Size distribution of transported dust**

Why is the comparison with AERONET presented in the supplement?

**4.2.1. Dust emission schemes**

Please, avoid relying on excessively on external references to explain features observed among the experiments (e.g. lines 561 to 563), summarize them directly here.

What is the impact of the dust tuning on the results? According to section 2.5, both EXP06 (MINE_BASE) and EXP07 (MINE_EMIS) were tuned to match a global DOD of around 0.03. Was that not the case? What does the re-scaling of the DOD mentioned on line 591 refer to?

**4.2.3. Dust asphericity**

The authors state that the dust asphericity could mediate the overestimated dust emission from source regions, is this shown in their experiments? Does the asphericity factor affect differently fine vs coarse particles?

**4.2.4. Dust size representation**

This section is difficult to follow, please, revise.

**4.3. Dust direct ratiative effect.**

Details such as the LW increase by 51% could be explained in section 2.3. I would only mention this again here if the approach used in the different experiments would differ, and thus affect the comparison.

**4.3.1. Dust direct radiative effect efficiency.**

Please, use also section 2.3 to define the net DRE efficiency.

What is the metric used here to define the model performance?

The difference between the experiments with speciated and bulk dust is not exclusively dependent on the developments presented here, but, as the authors mention, attributed to the resulting optical properties for the different representation on the dust.

Does the model diagnose all sky or clear sky DRE (line 730)? Please, clarify this in section 2.3.

**Conclusions.**

The authors mention the effect of dust asphericity on mass extinction efficiency as one of the aspects that produces a larger change in the results, as mentioned above, it is unclear to the reader which is the approach followed to introduce this in the model and/or if it's introduced at all.

I believe it would be useful to include a brief discussion on the implications of reverting the standard deviation changes in the coarse mode for the stratospheric aerosols. If the change was initially introduced to better accommodate those, which would be the recommendation of the authors for the model version to be issued?

The authors comment on potential ways of improving further the dust cycle, however, it is unclear for the reader if those stem from the work performed in this article. I would recommend to highlight the weaknesses detected in this study concerning the dust cycle representation (even after all the improvements included), and link to the appropriate suggested next step to solve that issue.

**Technical corrections**

Please, find below a list of technical corrections that could be applied to the current manuscript version.

L19. Either refer to the CAM6 model in the abstract (as it is in the article title) or change the title to include the CESM model.

L23-24. If possible, outline the main changes included in the different parameterizations (emission, dry deposition, size distribution and dust particle shape).

L26-27. Is it the effect of the size distribution change as large as the change in the dust emission scheme?

L46. Is shape also a factor affecting the uncertainty in dust direct radiative effect?

L63-64. Is it necessary to mention the previous CAM and CESM versions?

L71. Why do the authors mention now the Community Land Model version 5 (CAM6.1/CLM5)? Please, use the same acronym/naming convention all along the article, either CAM6.1 or CAM6.1/CLM5, or at least, mention the full name the first time it appears and explain that from then on it will be referenced as CAM6.1.

L102 (Table 1 caption): MAM4 is mentioned for the first time. Why use two abbreviations for the standard deviation, remove extra dot after CAM6.1 in L103.

L108: Homogenize the naming of the sections, either Sect. or Section.

L109: I would substitute semi-observation by more specific term(s).

L117: Is it CESM2.1 or CESM2? Please, keep consistency in the naming of the model versions along the document.

L125: Why is the iron solubility mentioned here?

L126: I would state in the introduction that the tests are to be conducted under present climate conditions, this will already justify using observations for the same period and then the clarification on the pre-industrial will not be needed here.

L138: Please, change "models" by model.

L139: Please, remove "generally".

L141: CESM2, CESM2.1? CLM?.

L153: Please, rephrase to specify the variable that is independent of the friction velocity (rather than the theory itself).

L154-155: As it is expressed now, the improvement in CAM4 size distribution is not informative to the reader. Please, either remove the part about the improvements or to briefly explain the difference between the approaches in previous CAM4 PSD and that derived from Kok (2011).

L156: Please, remove "other", and "of aerosols".

L179: Please, remove "the so-called".

L178: As mentioned above, please, select just one acronym for the standard deviation.

L189: Please, change "their ranges", by "its ranges".

L195: The reference to Scanza et al. (2015) was already included.

L221: Is the vertical transport modified per se? Or is it indirectly affected by changes in emission/size?

L231: What do the authors mean by "although even dust modeling with BRIFT can be improved if optimized against observations", is that optimization relevant for this specific study?

L328: Please, avoid repeating references unnecessarily (e.g. remove described in Sect. 2.2).

L333: There are two references for Kok et al. (2021), please, specify a or b.

L338: Please, change "could change", by the appropriate: does or does not change the model performance?

L338-343. May not be necessary to explain again the content of each sub-section.

L358: Please, explain what the binned method is.

L466 (and other locations in the text): Please, refer to the different *experiments* as such, instead of mentioning the *models.* If preferred by the authors, they could use *model versions.*

L369: Please, identify the reference with a or b.

L432: Typo: averages.

L439: Change "to the low" by "to the lower".

L475: Have the authors information on the precipitation evaluation for their own model?

L524: Please, include the coordinates of both stations or none.

L543: Does the super coarse dust start at 10 um? or larger diameters?

L614: Hematite and illite have a high iron content, feldspars not much. The sentence could be rephrased as ", including hematite and illite, and feldspar"

L636: I believe the increase is in wet deposition (not dry), please, verify.

L655: Please, include the full reference and then in parenthesis the values.

L683: Why is the calculation explicitly included there? It makes the text more difficult to read. I would avoid it (here and in other locations in the text below).

L699: The sentence "where the dust emission occurs in transport" is difficult to understand, please, clarify.

L869: Substitute "new model" by the appropriate model version name.

**Supplement:**

The caption in Figure S2 mentions the sub-regions being defined in Figure S3, but Figure S3 only lists the names of the regions, and those do not correspond to the legend in Figure S2. Please, review.

**Tables and figures**

*Table 1*: I would order the modes from smaller to larger in size. I believe this table could be included in the supplement and leave in the text exclusively the default and new configuration for the coarse mode.

*Table 4:* Why is the dust SSA for NEW_EMIS_SIZE missing?

Please, homogenize the naming convention for the different experiments, here tagged in Table 4 as NEW_EMIS, NEW_EMIS_SIZE, etc. In Table 2 and sections 2.1 and 2.2 they were listed also as EXP01, EXP02, etc. In Table 4 caption CAM6S5 and CAM6S6 are mentioned, which were not identified nor described before.

*Table 5*: Could the locations be represented in a map, together with the other observations location?

*Figure 1:* Which is the metric used to define the improvement (+) or worsening (-) of the comparison? Remove the comment on Figure S3 from the caption, and if needed, clarify in the text (line 392) the information presented in main paper and in the supplement.

*Figure 2:* Could the re-scaling factors now explained in the caption be included also in the figure legend (e.g. above each map)?

*Figure 5*: Please, review the caption: remove "and" in the third line, remove "for the abbreviation for other models", either explain them there or leave just the reference, specify what do we understand by semi-observations. Please, do not refer to other figures in figure captions unless they are needed to understand the figure contents.

*Figure 6*: What do the maps represent? Is it the ratio? Or the differences over the reference?

*Figure 7:* Please, use the same naming convention for the different experiments along the manuscript, otherwise is very confusing.

*Figure 8*: Homogenize the experiment names with the rest of the document, review the seasons listed in the caption, the inserted map below is not shown in this document version.

---

## Referee Comment (RC2)

**Review of "Importance of different parameterization changes for the updated dust cycle modelling in the Community Atmosphere Model (version 6.1)" by Li *et al.**

In this paper, Li *et al.* investigate the sensitivity of dust in the CESM2-CAM6.1 climate model to various parameterised processes: the emissions scheme, the dry deposition scheme, the fixed geometric width of the coarse mode, and the assumption of spherical/aspherical particles. Using a wealth of validatory observations and many simulations, they find that changing dust emissions and the coarse mode width have the greatest impact on the dust metrics, followed by the dry deposition scheme and then asphericity. They also propose a new version of CAM (CAM6.α) which improves on many dust metrics relative to CAM6.1 and incorporates some of the listed process changes.

The paper is well written and contains a wealth of useful information, including the most comprehensive database of dust observations yet (Table 3). The introduction is highly readable, and the conclusions are generally supported by the analysis. However, this paper rather feels like 3 independent studies convoluted together, namely, (1) a new and improved version of the dust scheme in CAM (CAM6.1 versus CAM6.α), (2) a study of the sensitivity of simulated dust to certain processes, and (3) a study of the merits of separating dust into its mineralogical components in CAM. I think the paper would benefit from being split into 2 or 3 separate papers, which I expand on below in the General Comments. In short, I think that the study needs a redesign before it is published, which may require major revisions (i.e., new simulations and a re-write) and/or splitting into separate papers.

**General comments**

Firstly, I think that the simulation design is incorrect for exploring the sensitivity of dust to the altered processes. For example, the new dry deposition scheme is only tested in conjunction with the other altered processes (CAM6.α and CAM6.α_MINE) and never on its own. Conversely, the new emissions scheme is tested by itself for both BULK and MINE dust models, whilst the size and shape of the particles are tested in conjunction with the new emissions scheme but using BULK and MINE dust respectively. In short, it's very difficult to attribute the impacts on the dust metrics to the individual processes.

I would suggest concentrating on either the BULK dust scheme or the MINE dust scheme, unless you plan to directly compare them. The study would be much cleaner if the processes were tested in isolation using either BULK or MINE and then compared to CAM6.1 (see Table below). In its current form, it is very difficult to disentangle which dust impacts emanate from which altered process.

Suggested simulations:

| Simulation | Name | Description |
|---|---|---|
| 1 | CAM6.1 | Standard model |
| 2 | NEW_EMISS | CAM6.1 with BRIFT emissions |
| 3 | NEW_SIZE_S5 | CAM6.1 with CAM5 size assumptions |
| 4 | NEW_SIZE_S6σ5 | CAM6.1 with CAM5 assumptions except coarse σ from CAM6.1 |
| 5 | NEW_DRYDEP | CAM6.1 with PZ10 dry deposition |
| 6 | NEW_SHAPE | CAM6.1 with aspherical dust |
| 7 | CAM6.1_MINE | Equivalent to MINE_BASE but may use CAM6.α as BASE simulation |
| 8 | CAM6.α | CAM6.1 with all of the relevant model changes |

Another issue that I had with the simulation design was the arbitrary tuning of dust optical depth (DOD) to 0.03 in some simulations but not in others (L289). This made it very difficult to quantify the impact of the altered processes and forced the authors to add caveats throughout the text e.g., L590 *"differences between the global annual mean dust deposition in BRIFT and DEAD would become smaller, if we rescaled the value according to the same DOD criteria"*. I suggest only tuning CAM6.1 and CAM6.α to 0.03 and using the tuned CAM6.1 as the BASE model in which to add the different processes incrementally. I see no need to rescale DOD in the sensitivity simulations and it would be interesting to see the impact of the different processes on the global-mean DOD as a derived product of the models. Tuning to 0.03 is arbitrary and also misses the fact that much of the dust mass is in the super coarse mode which is missing from the model, and therefore the model may be wrongly tuned to 0.03.

It is also confusing for the reader that some simulations have emissions scaled by 1/f_clay whilst others have the scaling as 1, and so the impact of this change is difficult to disentangle using the current suite of simulations. It would be better if this factor is consistent across the simulations or tested in isolation.

I gather from the text (L649) that the impact of asphericity on the dust mass extinction efficiency (MEE) is represented in *all* of these simulations. This is rather confusing, as it suggests some representation of asphericity is incorporated even when dust is assumed to be spherical (?). Please clarify this for the reader. In particular, please state whether the impact of asphericity on MEE is only applied in the simulation with dust asphericity or in all simulations (which seems inherently wrong). Really these details should be included in the Methods (L98, L224) and not in the result section.

In terms of the presentation of the results, I thought that comparing CAM6.α with CAM6.1 before looking at the individual processes was confusing, as much of the analysis of the impacts of individual processes could have been used to explain differences between the dust metrics in CAM6.1 and CAM6.α. Additionally, the authors say the following in Section 2.5:

*"It is worth noting that dust burdens and deposition fluxes would be comparable, if the bulk and speciated dust models have similar DOD. But the dust optical properties (e.g., single scattering albedo) in the bulk and speciated dust simulations differ, resulting in considerably different dust direct radiative effects and direct radiative effect efficiencies. Therefore, we state the difference in the dust DRE and DRE efficiency estimate in Sect. 6, but do not document the comparison of dust loadings/deposition/DOD between the bulk and speciated dust simulations."*

Given that DOD is tuned to be similar in these simulations, I do not see why the differences in optical properties should be used as an excuse not to compare BULK with MINE. This would be a very interesting study in its own right, and possibly the authors should omit MINE simulations in this paper as without comparing BULK with MINE, it is difficult to understand why MINE is used at all. Is the additional mineralogical detail in MINE useful for a better dust simulation? What is the additional computational expense of MINE over BULK? Is MINE being considered for inclusion in a future of CAM or is this rather an interesting pedagogical study? Currently, MINE is frivolously used in this study and is unnecessary without further analysis and comparison.

In summary, I would highly recommend that the authors run further simulations with each of the processes applied separately as the current simulation design is not conducive or particularly supportive of the results presented in the manuscript.

**Specific comments**

[L75]    Is it worth introducing the DEAD and BRIFT acronyms here?

[L84]    The fine mode is described as d < 1um whilst the coarse mode is d > 5um. Normally, the coarse mode is adjacent to the fine mode so I wonder what the authors would define the intermediate aerosol (1 < d< 5um) as?

[L91]    *"one of the changes from CAM5 to CAM6.1 was replacing the size distribution of aerosols in the coarse mode in CAM5 with the one that has a much narrower width in CAM6.1"*- this seems nonsensical to me, or completely without consideration for actual coarse mode dust widths (e.g., Ryder et al, 2013, 2018, 2019 suggest σ ∈ [1.6, 2] rather than 1.2). Why was it decided to favour stratospheric sulfate over tropospheric mineral dust when sulfate is more episodic (e.g. volcanic eruptions) and has less of an impact over tropospheric climate? Also, the authors seem to recommend that the coarse mode width be reverted to 1.8 as in CAM5 (I agree), but do not comment on the impact of resetting the coarse mode width on stratospheric sulfate. Seeing as this was the initial motivation for contracting σ, I think that some comment is appropriate.

[Table 1] I think that GMD should be labelled as "initialisation GMD" as this is more descriptive. Or is the initial GMD at source calculated online? It is difficult to tell from the text what the initial GMS is. This also refers to L179.

[Table 1] Why is the order of the modes Accumulation, Aitken, Coarse, then Primary? Surely it should be in ascending size order: Primary, Aitken, Accumulation then Coarse

[Table 1] Why was the accumulation mode width changed in CAM6.1? What are the impacts of reverting it? I can't see this detail in the text

[L109]   The term 'semi-observation' is undefined and is confusing

[L115]   *"show the final summarization in Section 7"*. This is an unusual way to say "Discussion and conclusions are provided in Section 7" or something to that effect

[L120]   This is one of the places in the text where it is unclear as to: (1) whether the impact of dust asphericity on MEE is represented at all, (2) if it is represented then in what way (methods), and (3) which simulations include it?

[L137]   Sentence beginning *"We consider the default DEAD scheme"* should explicitly acknowledge that it refers to emissions

[L143]   How confident are the authors in the critical LAI threshold? Should this assumption be discussed in the Discussion section?

[L152]   The mass is distributed as 0.1 %, 1 % and 98.9 % between the Aitken, accumulation, and coarse modes. Surely these ratios should change depending on the assumed coarse mode width?

[L160]   Many dust schemes treat dust as initially insoluble and then permitted to age via coagulation and condensation wherein it becomes soluble and internally mixed (e.g., dust in UKESM1). The authors should comment on their assumption of internally mixing dust, which may artificially enhance dust deposition near source regions? Would you expect similar results if dust is assumed to be insoluble?

[L165] The Neale et al (2010) reference is an internal document, which I can't find online. Can the authors please provide a URL for downloading the report, or alternatively, relevant peer-reviewed papers with the same information.

[L172] *"The wet deposition rate thus depends on the hygroscopicity of dust (=0.068; Scanza et al., 2015) as CCN/INPs and the prescribed scavenging coefficient (=0.1; Neale et al., 2010), both of which are currently constant with respect to the dust size (and composition for speciated dust) in CAM6.1."* – I assume the hygroscopicity of dust will evolve as dust is transported through the atmosphere so I question the use of a single spatially uniform constant for this parameter. The below cloud scavenging coefficient (0.1), if it is in units of $s^{-1}$, seems 2 orders of magnitude too high (Wang et al., 2010, doi:10.5194/acp-10-5685-2010). Wang et al (2010) for instance, suggest it's somewhere between $10^{-6}$ for accumulation mode aerosol and $10^{-3}$ for coarse mode aerosol depending on scavenging rate. The authors should comment more on the assumptions made in the model and the implications of those assumptions.

[L180] *"Note that the current default CAM6.1 employs a narrow coarse-mode size distribution but a broad boundary width (high bound minus low bound), likely resulting in the GMD bounds less in effect, compared to that in CAM5".* – what are the impacts of changing the coarse mode width on sea-salt emissions and sea-salt AOD? Surely this change will impact more than dust alone, which may be confounding other results presented in the study (e.g., the DRE).

[L210] *"The wet size due to growth of aerosol particles by adsorbing water vapor follows the κ-Kohler theory with a time-invariant hygroscopicity for each aerosol species (Petters and Kreidenwei, 2007)".* – is it worth listing these hygroscopicity parameters to aid in the replicability of the simulations?

[L215] *"here and hereafter unless stated otherwise"* – this phrase, in parentheses, doesn't seem to apply to anything or make sense

[L224] This is another place in the text where the impact of asphericity on the MEE is tantalisingly hinted at without further detail as to whether its on and how its incorporated

[L276] "In addition, the meteorology field (horizontal wind, air temperature T, and relative humidity) was nudged" – the results will obviously be changed if the model is free running then. For instance, the coarse dust will absorb LW radiation, warming the surface and destabilising the atmosphere. Perhaps this assumption (fixed meteorology) should be discussed in the Discussion section

[L285] *"Therefore, we state the difference in the dust DRE and DRE efficiency estimate in Sect. 6, but do not document the comparison of dust loadings/deposition/DOD between the bulk and speciated dust simulations."* – Avoiding comparing BULK and MINE seems like a massive oversight and is one of the first things I'd query as a reader. Does speciation between minerals improve the simulation compared to assuming dust as a bulk quantity? Simply saying that as the dust properties are different (of course they will be), this reduces comparability, is a little bit absurd and a bit of a cop out. I think this comparison should be made in a follow-on paper. To be honest, it doesn't seem worth including the MINE simulations if they not appropriately analysed.

[L289] Choosing to tune some models to DOD = 0.03 but not others is very peculiar. The authors say *"Dust tuning was not applied to EXP03 and EXP04 (bulk dust simulations), in which the*

*dust emission was identical to EXP02, in order to see how changes in the transported dust size distribution affects the DOD calculation".* – Well surely all of the individual sensitivity simulations (emissions, dry deposition, asphericity) would have benefitted from the same analysis? I guess that some parameters in the emissions and dry deposition algorithm need to tuned in some way (so using DOD might be a reasonable approach) as the parameters have a huge degree of uncertainty, but the asphericity probably did not need changing.

[L289]   My other issue with this paragraph is that the tuning is not described in any detail. Which parameters were tuned and what are their values in the baseline simulation? How was tuning conducted and why was global-mean DOD chosen as the target? Simply saying 'tuned the model following Albani et al (2014)' is not sufficient, and it would be impossible to replicate these simulations without further detail

[Table 3] This table seems very large, and I'm not sure whether the list of acronyms should be at then end of the table or in the caption. Would it be better to have 1 table for each metric?

[Results] The difference between CAM6.1 and CAM6.α i.e., the control and the simulation with all changes added (except mineralogy) comes before the dissection of impacts of individual processes. Why is this? Surely it would be better to investigate the impacts of the individual processes and then use them to explain why CAM6. α a is different to CAM6.1?

[L378]   *"CAM6.1 may overestimate the contribution of high-latitude dust emissions to the global dust total (8.0%)."* – is this referring to the dust burden? It's rather ambiguous as is

[L391]   *"Overall, all models reproduced the climatology of DOD from AERONET retrievals, the surface concentration, and deposition within a factor of ten (Fig. 1 and Fig. S3)"* – this doesn't seem to be the case from looking at Fig. 1 b, c, e, f, h, and i. It seems that both models exhibit at least one measurement outside the range of 1/10x and 10x.

[Fig. 2]   Why is the new dust emissions scheme smoother in terms of emissions, rather than the delta function (almost) in DEAD? I couldn't easily find this information in the text

[Fig. 3]   Isn't the Ridley et al (2016) DOD dataset constrained by MODIS (either through assimilation or using it as a baseline? If so, aren't Figs 3a and 3b effectively showing the same results?

[L436]   capture -> captures

[L437]   Taklamakan (as in the desert) is spelt wrong throughout

[L440]   *"Comparing with both datasets suggests that the new model may overestimate the regional DOD over North Africa and Middle East within a factor of two."* – this is especially odd considering the omission of dust mass for super-coarse particles from the model. Incidentally, would you expect the super-coarse dust to contribute much to 550nm DOD?

[Fig. 4]   Great figure

[L498]   S5i -> S5e

[Fig. 5]   This plot, especially Fig. 5a, is very confusing. There are too many colours and it is difficult to pick out the CAM models. It may be worth plotting a non-CAM multi-model mean with max/min as shaded in grey, and then have just the CAM models in colour

[L542]   Why is the size distribution for the fine dust fraction better captured by CAM6.α?

[L548] Sentence beginning *"Overall, CAM6.α better reproduced the size distribution"*. It would be worth adding the caveat here that the Otto et al and Ryder et al measurements are from single campaigns or flights and thus may not reflect the long-term mean dust properties at those altitudes, locations, and times

[L558] Section 4.2.1 – why are the mineralogy experiments used to test BRIFT vs DEAD rather than the BULK simulations? There doesn't appear to be any reasoning behind this

[L559] MIINE_NEW_EMIS -> MINE_NEW_EMIS

[L646] Paragraph on asphericity – I'm still confused even after reading the text as to whether the assumption of asphericity is applied to the dust MEE in every simulation run here or just the MINE_NEW_EMIS_SHAPE simulation?

[L683] *"(0.030-0.019)/0.030*100)"* – I don't think this formula needs to be written. See also L686 and L759

[L693] Paragraph beginning *"The lifetime of dust"*. Should this paragraph be in Section 4.2.4? It doesn't seem to mention asphericity or apply to the MINE_NEW_EMIS_SHAPE simulation

[L705] Why is MINE_NEW_EMIS referred to as the reference case? It's a sensitivity simulation, isn't it? Surely the only reference cases are CAM6.1 and possibly MINE_BASE?

[L733] *"NEW_EMIS_SIZE"* -> MINE_NEW_EMIS_SIZE. Also, this paragraph seems to be the only place where BULK and MINE are explicitly compared. I think the comparison should extend to all the dust metrics

[L798] *"Overall, replacing the size distribution of dust aerosol and the dust emission scheme with new ones (PZ10 and BRIFT, respectively)"* – replacing the size distribution is referred to here as PZ10 but this is the dry deposition scheme

[L821] The term "space volume" is ambiguous. Possibly "colocation in space"?

[L833] *"which can get mixed with dust aerosol particles during the transport and may not be completely excluded in the measurements."* This seems a little lazy, do you have any estimates of how much contamination leads to errors in measuring dust? At the moment, this point isn't backed up by evidence.

[L859] *"… followed by the enhanced dust mass extinction efficiency at the visible band by ~30% to account for the enhancement by dust asphericity"* – the asphericity applied to the MEE has not been shown to be the second most important change affected. Rather Fig. 10 shows that asphericity has a negligible impact on dust. Or is the asphericity in the MEE applied separately to the asphericity in the deposition rate? This is very confusing.

[L869] "Overall, the new model can:" – is the new model, referred to in this sentence, CAM6.α? If so, why has CAM6.α_MINE been neglected? The addition of MINE to this study makes little sense as it is peripheral. Additionally, is this "new model" already adopted for the next revision of CAM6 or is this the plan for the future?

---

## Author Comment (AC1)

Dear Juan A. Añel,

Thanks for your notice!

We have archived the materials used for our study, including the model results, observations, and model code in Zenodo with an assigned DOI:10.5281/zenodo.6989502.

We have also revised the 'Code and Data Availability' section accordingly, removing from the text the mentions to GitHub and providing a link to the Zenodo repository.

Best regards,
Longlei

---

## Author Comment (AC2)

- **Point-by-point response to the reviews**
- 3 Referee 1
- 4

5 We thank this reviewer very much for the so detailed constructive comments on this work. We 6 have made changes to the manuscript accordingly. We colored our response in blue. Text from 7 the manuscript is quoted with double quotation marks and new text is shown in *italics*.

8

**9 General comments**

10

This article presents multiple developments included in the dust cycle representation within the CAM6.1 model and assesses their impact on relevant variables, such as the dust surface concentration, deposition, size distribution, optical depth and direct radiative effect. The work conducted provides relevant information beyond the dust modeling community, as dust has impacts on different features of the atmospheric dynamics and chemistry, the climate and the Earth System. As such, I believe this article is well within the scope of the Geoscientific Model Development journal, it presents novel results, and it deserves publication.

18

20

- 19 Many thanks for the positive comments.
- However, in my view, in its current form the reader has to put in a considerable effort to follow the details of the massive amount of work presented.
- 23

24 Thanks for the comments and time in reviewing the manuscript.

25

The authors present nine different experiments: five defining dust as a bulk species and four experiments considering speciated dust. This involves a duplication of experiments in which one (or several) of the new developments are tested, and adds an additional variable to the analysis, making it harder to focus on the specific impact of the new aspects included in the model.

30

31 With respect to the experiments design, the authors could better clarify the criteria used to 32 include the new features in the tests. Instead of relying on a baseline (e.g. CAM6.1), and adding 33 separately to that configuration the different developments (on the emissions scheme, dry 34 deposition, size, or asphericity), the authors combine multiple developments in the different 35 experiments. I believe these combinations could hinder a clean comparison of the effect of each 36 development (e.g. looking at Table 4 it is difficult to know which pair of experiments allows 37 disentangling the effect of shape and deposition changes). This issue is accentuated by the fact 38 that the experiments are referenced along the manuscript by different names or acronyms, which 39 further complicates tracing them.

41 There are a couple of different methods to estimate the effect of each development, such as the 42 one we used and the one the reviewers suggested. Strictly speaking, either method cannot 43 totally exclude the possible influence of the parametrizations that had already been included and can affect the dust cycle modeling in the base model, such as the advection scheme and cloud 44 45 processing. The reason is that there likely exists a nonlinear "interaction" between the existing parameterizations and the newly introduced one, which seems weak though. We acknowledge 46 47 that adding one by one seems clearer than the original experiment design, but it requires more simulations and thus more computational resources while yielding a similar estimate of the 48 49 impact of each development (Fig. R1), compared to what we had presented based on our own's 50 experiment. We had selected our own's set of experiments, because adding the modification on 51 top of the previous change can help understand how the simulated dust cycle evolves while updating the model (MINE BASE) toward the most advanced one (CAM6.α MINE). 52

53

54

Fig. R1. Influence of changing to PZ10 on the simulated dry deposition fluxes in the dustspeciated model (change to the global annual mean of dry dust deposition: ~70 Tg) based on our experiment (a) and the suggested experiment by the reviewers (b). Quantified change to the global annual mean of dry dust deposition equals ~70 Tg by either method.

59

The BULK runs were constructed to investigate how the mistakenly set dust size distribution influences the dust cycle modeling and the estimate of dust DRE. This inappropriate size distribution has been employed in studies using the officially released BULK CAM6 and not in any study using the dust-speciated CAM. So, we do not have a good reason to perform size sensitivity tests in the MINE runs. What's more important is that quantifying the impact of

65 individual processes, based on the base CAM6.1 that uses an inappropriate dust size distribution, seems not that meaningful: it makes more sense to make such quantification using models with 66 the "correct" size distribution. That is why in all the MINE runs designed for that purpose we 67 revert the narrow coarse-mode size distribution to the broad one. Also, following the reviewer's 68 69 design would change little to the results obtained from our experiments on the dust cycle modeling. This is because the offline dynamics and the dust tuning employed ensures quite 70 similar dust cycles modeled by BULK and MINE with different developments (Fig. R2 and Fig. 71 72 R3), if the size distribution is also set identical, since the sum of the mass fraction for each of the 73 eight minerals always equals unity. We had pointed this similarity out in our originally submitted 74 manuscript: "It is worth noting that with the dust tuning applied toward the similar global mean 75 DOD ~0.03 the modeled dust cycle (i.e., burdens, concentrations, loadings, deposition fluxes) would be similarly comparable between the bulk and speciated dust models using the same 76 77 offline dynamics and dust size distribution". Repeating the set of simulations using BULK instead 78 to quantify the impact of each altered process would then yield similar results that we presented 79 in the manuscript.

---

## Author Comment (AC3)

**Point-by-point response to the reviews**

**Referee 2**

We thank this reviewer very much for the so detailed constructive comments on this work. We have made changes to the manuscript accordingly. We colored our response in blue. Text from the manuscript is quoted with double quotation marks and new text is shown in *italics*.

In this paper, Li *et al.* investigate the sensitivity of dust in the CESM2-CAM6.1 climate model to various parameterized processes: the emissions scheme, the dry deposition scheme, the fixed geometric width of the coarse mode, and the assumption of spherical/aspherical particles. Using a wealth of validatory observations and many simulations, they find that changing dust emissions and the coarse mode width have the greatest impact on the dust metrics, followed by the dry deposition scheme and then asphericity. They also propose a new version of CAM (CAM6.α) which improves on many dust metrics relative to CAM6.1 and incorporates some of the listed process changes.

The paper is well written and contains a wealth of useful information, including the most comprehensive database of dust observations yet (Table 3). The introduction is highly readable, and the conclusions are generally supported by the analysis. However, this paper rather feels like 3 independent studies convoluted together, namely, (1) a new and improved version of the dust scheme in CAM (CAM6.1 versus CAM6.α), (2) a study of the sensitivity of simulated dust to certain processes, and (3) a study of the merits of separating dust into its mineralogical components in CAM. I think the paper would benefit from being split into 2 or 3 separate papers, which I expand on below in the General Comments. In short, I think that the study needs a redesign before it is published, which may require major revisions (i.e., new simulations and a re-write) and/or splitting into separate papers.

We appreciate the positive comments very much. The reviewer correctly pointed it out that this is a paper convoluted together by independent studies. Our original plan, however, was to separately document the size change in BULK CAM6, and the improved emission and deposition parameterizations in CAM6. The merits of modeling dust as mineral components have been very well documented before in terms of the climatic impacts by mineral dust, so we do think it would not deserve a new paper on this. Since we tend to update separate processes in CAM6 and the new schemes have been detailed and tested offline or in previous versions of CAM (CAM4 and CAM5), it makes more senses to document in the same paper how the change to each process may affect the dust cycle modeling. Please see our reply to the comments below including that regarding to the experiment redesign.

Firstly, I think that the simulation design is incorrect for exploring the sensitivity of dust to the
altered processes. For example, the new dry deposition scheme is only tested in conjunction
with the other altered processes (CAM6.α and CAM6.α_MINE) and never on its own. Conversely,
the new emissions scheme is tested by itself for both BULK and MINE dust models, whilst the
size and shape of the particles are tested in conjunction with the new emissions scheme but
using BULK and MINE dust respectively. In short, it's very difficult to attribute the impacts on the
dust metrics to the individual processes.
I would suggest concentrating on either the BULK dust scheme or the MINE dust scheme, unless
you plan to directly compare them. The study would be much cleaner if the processes were
tested in isolation using either BULK or MINE and then compared to CAM6.1 (see Table below).
In its current form, it is very difficult to disentangle which dust impacts emanate from which
altered process.
Suggested simulations:

| Simulation | Name | Description |
|---|---|---|
| 1 | CAM6.1 | Standard model |
| 2 | NEW_EMISS | CAM6.1 with BRIFT emissions |
| 3 | NEW_SIZE_S5 | CAM6.1 with CAM5 size assumptions |
| 4 | NEW_SIZE_S6σ5 | CAM6.1 with CAM5 assumptions except coarse σ from CAM6.1 |
| 5 | NEW_DRYDEP | CAM6.1 with PZ10 dry deposition |
| 6 | NEW_SHAPE | CAM6.1 with aspherical dust |
| 7 | CAM6.1_MINE | Equivalent to MINE_BASE but may use CAM6.α as BASE simulation |
| 8 | CAM6.α | CAM6.1 with all of the relevant model changes |

In summary, I would highly recommend that the authors run further simulations with each of the
processes applied separately as the current simulation design is not conducive or particularly
supportive of the results presented in the manuscript.
This is a similar comment to what the first Reviewer raised. Below we paste our reply to the
comment by Reviewer # 1 (Line 41-169 in that report) as a response in the text.
There are a couple of different methods to estimate the effect of each development, such as the
one we used and the one the reviewers suggested. Strictly speaking, either method cannot
totally exclude the possible influence of the parametrizations that had already been included and
can affect the dust cycle modeling in the base model, such as the advection scheme and cloud
processing. The reason is that there likely exists a nonlinear "interaction" between the existing
parameterizations and the newly introduced one, which seems weak though. We acknowledge
that adding one by one seems clearer than the original experiment design, but it requires more
simulations and thus more computational resources while yielding a similar estimate of the impact of each development (**Fig. R1**), compared to what we had presented based on our own's
experiment. We had selected our own's set of experiments, because adding the modification on
top of the previous change can help understand how the simulated dust cycle evolves while
updating the model (MINE_BASE) toward the most advanced one (CAM6.α_MINE).

a. CAM6.α-MINE_NEW_EMIS_SHAPE: $-5.9 \times 10^{-14}$ kg m$^{-2}$ s$^{-1}$

b. PZ10-Z01: $-5.9 \times 10^{-14}$ kg m$^{-2}$ s$^{-1}$

$$-700 \quad -560 \quad -420 \quad -280 \quad -140 \quad 140 \quad 280 \quad 420 \quad 560 \quad 700$$
Dust DDF ($\times 10^{-14}$ kg m$^{-2}$ s$^{-1}$)

**Fig. R1.** Influence of changing to PZ10 on the simulated dry deposition fluxes in the dust-
speciated model (change to the global annual mean of dry dust deposition: ~70 Tg) based on
our experiment (a) and the suggested experiment by the reviewers (b; Simulation 5 – simulation
1). Quantified change to the global annual mean of dry dust deposition equals ~70 Tg by either
method.

The BULK runs were constructed to investigate how the mistakenly set dust size distribution
influences the dust cycle modeling and the estimate of dust DRE. This inappropriate size
distribution has been employed in studies using the officially released BULK CAM6 and not in
any study using the dust-speciated CAM. So, we do not have a good reason to perform size
sensitivity tests in the MINE runs. What's more important is that quantifying the impact of
individual processes, based on the base CAM6.1 that uses an inappropriate dust size distribution,
seems not that meaningful: it makes more sense to make such quantification using models with
the "correct" size distribution. That is why in all the MINE runs designed for that purpose we
revert the narrow coarse-mode size distribution to the broad one. Also, following the reviewer's
design would change little to the results obtained from our experiments on the dust cycle
modeling. This is because the offline dynamics and the dust tuning employed ensures quite
similar dust cycles modeled by BULK and MINE with different developments (Fig. R2 and Fig.

R3), if the size distribution is also set identical, since the sum of the mass fraction for each of the
eight minerals always equals unity. We had pointed this similarity out in our originally submitted
manuscript: "It is worth noting that with the dust tuning applied toward the similar global mean
DOD ~0.03 the modeled dust cycle (i.e., burdens, concentrations, loadings, deposition fluxes)
would be similarly comparable between the bulk and speciated dust models using the same
offline dynamics and dust size distribution". Repeating the set of simulations using BULK instead
to quantify the impact of each altered process would then yield similar results that we presented
in the manuscript.

[Figure]

**Fig. R2.** Surface dust emissions (a; global annual mean=2891 Tg) and deposition fluxes (b; global annual mean=2893 Tg) simulated by CAM6.α and their differences (c and d; both global annual mean=22 Tg) between MINE_ CAM6.α and CAM6.α.

[Figure]

**Fig. R3.** The same as **Fig. R2** but for DOD (a: global annual mean=0.030 and c: global mean difference=0.001) and dust burdens (b: global annual mean of dust mass=24 Tg and d: global mean difference≈0 Tg), respectively.

NEW_EMIS and MINE_NEW_EMIS appear like a duplication of experiment for testing the new dust emission scheme. But the estimate of dust DRE differs considerably between the two experiments.

To reflect the Reviewer's suggestion, we added the following in the section "2.6 Experiment design":

*"We investigate how the mistakenly set dust size distribution influences the dust cycle modeling and the estimate of dust DRE in the bulk-dust model rather the speciated-dust model, because this inappropriate size distribution has been employed in previous studies using the officially released bulk-dust CAM6 only and not in any study using the speciated-dust CAM. It is also reasonable to make all the quantifications in the model that use an appropriate dust size distribution. Therefore, we reverted the dust size distribution in all the speciated-dust runs to that configured in CAM5."*

*"It worth noting that with the dust tuning applied toward the similar global mean DOD ~0.03 the modeled dust cycle (i.e., burdens, concentrations, loadings, and deposition fluxes) would be*

*similarly comparable between the bulk- and speciated-dust models that nudged toward identical*
*offline dynamics and using the same dust size distribution (see Sect. 6). The quantified effect of*
*each of the modifications would thus be similar if using the bulk dust model instead (Fig. S2: R1*
*in this report), but the modeled dust optical properties (e.g., single scattering albedo) by the bulk*
*and speciated dust models differ considerably, resulting in considerably different dust DRE*
*(Scanza et al., 2015) and DRE efficiencies between NEW_EMIS (CAM6.α) and*
*MINE_NEW_EMIS (CAM6.α_MINE)."*

"*A comparison of the BULK and MINE models on simulating dust DRE had been previously*
*documented (Scanza et al., 2015). This study includes the MINE runs because we want to check*
*as well if the updates help improve reproducing the observed dust DRE efficiency in a model*
*that may more reasonably represent the regional variation of dust optical properties.* Note that
there are many ways to conduct sensitivity studies, which could lead to slightly different results.
*We added the modification on top of the previous change to understand how the simulated dust*
*cycle evolves while updating the model (MINE_BASE) toward the most advanced version*
*(CAM6.α_MINE). This may not hinder a clean comparison of the effect of each development,*
*since the 'interaction; between the existing and the newly introduced parameterizations appears*
*weak (Fig. S2: R1 in this report)."*

To clarify how we quantify influence of each development, we added two columns in the Table
4 pointing out the size distribution used and purpose of each experiment and added the following
text in the "Experiment design" section:

"*We separately compared the performance of PZ10 to Z01, aspherical to spherical dust, and*
*BRIFT to DEAD on the simulated dust cycle and quantified influence of each of those*
*modifications on the climatic-effect estimate by comparing the modeled dust cycle in the paired*
*simulations CAM6.α _MINE vs MINE_NEW_EMIS_SHAPE, MINE_NEW_EMIS_SHAPE vs*
*MINE_NEW_EMIS, and MINE_NEW_EMIS vs MINE_BASE, respectively."*

Finally, we added a separate new section to compare results from BULK with those from MINE:

"***6. Bulk- versus speciated-dust model***

*The bulk (CAM6.α) and dust-speciated model (CAM6.α_MINE) simulate a similar dust cycle with*
*the difference between the two types of models orders smaller than those simulated by the*
*former (Fig. 12 and 13: R2 and R3 in this report, respectively). This similarity results from the*
*dust tuning toward the global mean DOD of 0.03, the same meteorology dynamics both models*
*were nudged toward, and the design of the dust-speciated model that summing the mass fraction*
*of each dust species equals unity. With the same reasons, the influence of each of the*
*modifications on the modelled dust cycle quantified using the bulk model instead of the dust-*

*speciated model, as this study used, would be similarly comparable. The modelled dust optical properties, however, differs remarkably (i.e., dust SSA; Table 6) with the simulated global mean dust SSA by CAM6.α_MINE (0.896) lower than by CAM6.α (0.911) at the visible band centered at 0.53 μm. Note the dust DRE is sensitive to variation of the dust SSA. This lower dust SSA obtained here in the dust speciated model than in the bulk dust model is consistent to the finding of a previous study (Scanza et al., 2015) using an early model version (CAM5). Correspondingly, CAM6.α_MINE yields a reduced dust cooling (Table 6) and DRE efficiency (Fig. 9) than CAM6.α. For dust DRE efficiency (Fig. 9), speciating dust in CAM6 tends to reduce the RMSE while retaining the horizontally spatial correlation in either SW (CAM6.α: RMSE=11; R=0.26 versus CAM6.α_MINE: RMSE=10; R=0.20) or longwave (CAM6.α: RMSE=4; R=0.86 versus CAM6.α_MINE: RMSE=3; R=0.84) or both spectral ranges (CAM6.α: RMSE=7; R=0.93 versus CAM6.α_MINE: RMSE=6; R=0.92). This comparison suggests that modeling dust as component minerals with the dust size distribution in coarse mode of MINE_NEW_EMIS_SIZE helps improve the model performance relative to modeling dust as a bulk to reproduce the retrieved dust DRE efficiency (Fig. 9a). The improvement, however, could be artificial because of the combined use of imaginary complex refractive index of hematite volume (see Fig. 1b of Li et al., 2021) and the volume mixing used in the dust speciated model to compute the bulk-dust complex refractive index (Li et al. in prep.), leading to artificially more absorptive dust than in the bulk dust model (Fig. 9a and Table 6).*"

Another issue that I had with the simulation design was the arbitrary tuning of dust optical depth (DOD) to 0.03 in some simulations but not in others (L289). This made it very difficult to quantify the impact of the altered processes and forced the authors to add caveats throughout the text e.g., L590 "differences between the global annual mean dust deposition in BRIFT and DEAD would become smaller, if we rescaled the value according to the same DOD criteria". I suggest only tuning CAM6.1 and CAM6.α to 0.03 and using the tuned CAM6.1 as the BASE model in which to add the different processes incrementally. I see no need to rescale DOD in the sensitivity simulations and it would be interesting to see the impact of the different processes on the global-mean DOD as a derived product of the models. Tuning to 0.03 is arbitrary and also misses the fact that much of the dust mass is in the super coarse mode which is missing from the model, and therefore the model may be wrongly tuned to 0.03.

We had tuned CAM6.1 and CAM6.α as this reviewer also suggested toward 0.030. But we respectively do not agree with the reviewer that those are the only two simulations that need the retuning. For example, we must retune the model that uses the updated dust emission scheme. This is simply because if using the same tuning parameter value as in the model with DEAD, the global mean DOD would be >15 times higher than that in DEAD, reaching up to 0.45, which is undoubtedly unrealistic. We added the following in the manuscript.

*"MINE_BASE requires the dust tuning to use a much larger tuning parameter*
*(dust_emis_fact=3.64), compared to CAM6.1 (dust_emis_fact=0.91), because, otherwise, if*
*using the same dust_emis_fact as in DEAD, the dust emissions in BRIFT would lead to an*
*unrealistically high global mean DOD (>~0.5)."*

The MINE runs are not for sensitivity studies but used for quantifying how each of the
modifications affects the dust cycle modeling. We would obtain the same results if performing
BULK runs instead, because as stated, with the same model configurations set in this study, the
BULK and MINE simulations are nearly identical in terms of modeling the dust cycle. We clearly
pointed this out in the revised manuscript as below.

*"It is worth noting that with the dust tuning applied toward the similar global mean DOD ~0.030*
*the modeled dust cycle (i.e., burdens, concentrations, loadings, and deposition fluxes) would be*
*similarly comparable between the bulk and speciated dust models using identical offline*
*dynamics and dust size distribution. The quantified effect of each of the modifications would be*
*thus similar if using the bulk dust model instead."*

As to the dust mass distribution with respect to dust size, according to a recent study (Di Biagio
et al., 2020), for a total 39 Tg dust, approximately 33% (13) Tg dust are particles >10 μm, though
such estimates were obtained based on model simulations. However, this missing fraction of
"super-coarse" dust constitutes only a small fraction of the total DOD <2% which is even much
smaller than the uncertainty in the best estimate from Ridley et al. (2016). Therefore, we believe
missing that dust mass would not affect the accuracy of tuning dust toward DOD ~0.030.

In response to the reviewer's question about the dust tuning, we added some words to very
briefly explain why and how we tuned the model to get the global mean DOD ~0.030.

*"We prefer to tuning the model to reproduce the global mean DOD, 0.030, because DOD is*
*currently the best estimate of global dust quantities, compared to the others (i.e., dust*
*concentrations). It turns out that doing so can also reasonably reproduce the other quantities*
*with no need of a regional tuning. We tuned the dust model by modifying a namelist variable in*
*CAM, called soil_erod_factor, corresponding to $\lambda$ in Eq. (16)."*

Regarding the reviewer's suggestion to include the updates one by one, please see our
response to the previous comment on the experiment design (Line 64-159).

It is also confusing for the reader that some simulations have emissions scaled by $1/f\_clay$ whilst
others have the scaling as 1, and so the impact of this change is difficult to disentangle using
the current suite of simulations. It would be better if this factor is consistent across the
simulations or tested in isolation.

We thank the reviewer for the comment which makes us realize that our writing may be confusing.
This inversed clay fraction for the tuning factor b is not used in any of those simulations to
calculate the threshold gravimetric water content. To improve the readability and to not rely
excessively on external references, we introduced more the parameterization for both emission
schemes (please see Section 2.5 in the revised manuscript) and added a new column in Table
2 showing the b value used in each of those experiments.

[revised manuscript text omitted]

Note b is set to be 100*f$_{clay}$ as part of the DEAD emission scheme used in the default CAM6 but is set to be unity in BRIFT to well reproduce the observations:

*"An offline sensitivity test (Table S1: R1 in this report) supports the use of unity tuning factor to calculate the threshold gravimetric water content which we employed in the experiments for quantifying influence of each modification (speciated dust simulations listed in Table 2)."*

*"Table S1 (R1 in this report). Comparison of the three CESM simulations with the offline dynamics and different values of the tuning parameter (b) to calculate the threshold gravimetric water content in the new dust emission scheme, against measurements. The measurements include AERONET AOD climatology, surface dust concentrations, and dust deposition fluxes, as described in Section 3."*

| Parameter b | Correlation coefficient (RMSE) on climatology | | |
|---|---|---|---|
| | AERONET DOD | Surface dust concentrations (log space) | Dust deposition fluxes (log space) |
| 0.5 | 0.74 (0.13) | 0.83 (0.66) | 0.72 (0.93) |
| 1.0 | 0.68 (0.14) | 0.82 (0.72) | 0.77 (0.86) |
| 2.0 | 0.66 (0.14) | 0.83 (0.66) | 0.79 (0.82) |

I gather from the text (L649) that the impact of asphericity on the dust mass extinction efficiency
(MEE) is represented in *all* of these simulations. This is rather confusing, as it suggests some
representation of asphericity is incorporated even when dust is assumed to be spherical (?).
Please clarify this for the reader. In particular, please state whether the impact of asphericity on
MEE is only applied in the simulation with dust asphericity or in all simulations (which seems
inherently wrong). Really these details should be included in the Methods (L98, L224) and not
in the result section.

Such impact of the dust asphericity is included in all the simulations because we did not attempt
to quantify such effect in this study. To avoid of confusion and clarify this, we moved relevant
text from the result Section 4.2.3 (5.2.3 in the revised version) to Section 2.4.3, and added the
following to the "Experiment design" section:

"*The enhancement of the mass extinction efficiency of aerosol particles by dust asphericity is*
*included in all the simulations, since we do not attempt to quantify how this enhancement impacts*
*the simulated dust cycle, which has been previously well documented (Kok et al., 2021).*"

In terms of the presentation of the results, I thought that comparing CAM6.α _with CAM6.1
before looking at the individual processes was confusing, as much of the analysis of the impacts
of individual processes could have been used to explain differences between the dust metrics in
CAM6.1 and CAM6.α.

We think either doing what the reviewer suggested or keeping as what it was should work. In the
drafted manuscript, we had tried doing the same as the reviewer suggested but then reordered
the result section taking the "principle" that "the most important goes first", since the manuscript
is lengthy. In any order, the conclusions of this article would remain unchanged.

Additionally, the authors say the following in Section 2.5:

*"It is worth noting that dust burdens and deposition fluxes would be comparable, if the bulk and speciated dust models have similar DOD. But the dust optical properties (e.g., single scattering albedo) in the bulk and speciated dust simulations differ, resulting in considerably different dust direct radiative effects and direct radiative effect efficiencies. Therefore, we state the difference in the dust DRE and DRE efficiency estimate in Sect. 6, but do not document the comparison of dust loadings/deposition/DOD between the bulk and speciated dust simulations."*

Given that DOD is tuned to be similar in these simulations, I do not see why the differences in optical properties should be used as an excuse not to compare BULK with MINE. This would be a very interesting study in its own right, and possibly the authors should omit MINE simulations in this paper as without comparing BULK with MINE, it is difficult to understand why MINE is used at all. Is the additional mineralogical detail in MINE useful for a better dust simulation? What is the additional computational expense of MINE over BULK? Is MINE being considered for inclusion in a future of CAM or is this rather an interesting pedagogical study? Currently, MINE is frivolously used in this study and is unnecessary without further analysis and comparison.

We did not compare the modeled dust cycle between BULK and MINE runs, because this is science with secondary importance. We show the reason in the "Experiment design" section: "It is worth noting that with the dust tuning applied toward the similar global mean DOD ~0.03 the modeled dust cycle (i.e., burdens, concentrations, loadings, and deposition fluxes) would be similarly comparable between the bulk and speciated dust models using identical offline dynamics and dust size distribution. The quantified effect of each of the modifications would be thus similar if using the bulk dust model instead."

The different optical properties are not the reason for not making the comparison but are one of the reasons for why we included the MINE runs: we have shown in the text evaluations on the model performance of modeling the DRE efficiency and the influence of each modification on the DRE estimate, for which modeling the optical properties as accurately as possible is crucial. Therefore, dust speciated model is better to use to quantify such influence, as it simulates spatially varying dust optical properties, while the bulk dust model is using a globally constant dust optic.

We had tried to do this but found that having the potential impacts on the mineralogy by changing to the new dust emission scheme is not enough for a separate paper. Instead, we added more analysis on documenting results from the MINE runs, such that it makes more senses to have both BULK and MINE runs in this article.

"***6. Bulk versus speciated-dust model***

*The bulk (CAM6.α) and dust-speciated model (CAM6.α_MINE) simulate a similar dust cycle with the difference between the two types of models orders smaller than those simulated by the former (Fig. 12 and 13). This similarity results from the dust tuning toward the global mean DOD of 0.03, the same meteorology dynamics both models were nudged toward, and the design of the dust-speciated model that summing the mass fraction of each dust species equals unity. With the same reasons, the influence of each of the modifications on the modelled dust cycle quantified using the bulk model instead of the dust-speciated model, as this study used, would be similarly comparable. The modelled dust optical properties, however, differs remarkably (i.e., dust SSA; Table 6) with the simulated global mean dust SSA by CAM6.α_MINE (0.896) lower than by CAM6.α (0.911) at the visible band centered at 0.53 μm. Note the dust DRE is sensitive to variation of the dust SSA. This lower dust SSA obtained here in the dust speciated model than in the bulk dust model is consistent to the finding of a previous study (Scanza et al., 2015) using an early model version (CAM5). Correspondingly, CAM6.α_MINE yields a reduced dust cooling (Table 6) and DRE efficiency (Fig. 9) than CAM6.α. For dust DRE efficiency (Fig. 9), speciating dust in CAM6 tends to reduce the RMSE while retaining the horizontally spatial correlation in either SW (CAM6.α: RMSE=11; R=0.26 versus CAM6.α_MINE: RMSE=10; R=0.20) or LW (CAM6.α: RMSE=4; R=0.86 versus CAM6.α_MINE: RMSE=3; R=0.84) or both spectral ranges (CAM6.α: RMSE=7; R=0.93 versus CAM6.α_MINE: RMSE=6; R=0.92). This comparison suggests that modeling dust as component minerals with the dust size distribution in coarse mode of MINE_NEW_EMIS_SIZE helps improve the model performance relative to modeling dust as a bulk to reproduce the retrieved dust DRE efficiency (Fig. 9a). The improvement, however, could be artificial because of the combined use of imaginary complex refractive index of hematite volume (see Fig. 1b of Li et al., 2021) and the volume mixing used in the dust speciated model to compute the bulk-dust complex refractive index (Li et al. in prep.), leading to artificially more absorptive dust than in the bulk dust model (Fig. 9a and Table 6)."*

**Specific comments**

[L75] Is it worth introducing the DEAD and BRIFT acronyms here?

Introduced here.

[L84] The fine mode is described as d < 1um whilst the coarse mode is d > 5um. Normally, the coarse mode is adjacent to the fine mode so I wonder what the authors would define the intermediate aerosol (1 < d< 5um) as?

We just follow the definition normally used in the community. So, here is not a definition for the coarse mode aerosol. To avoid of possible confusions, we revised this statement as below:

"…*and slightly underestimating that of aerosols with diameter > 5.0μm…*".

[L91] *"one of the changes from CAM5 to CAM6.1 was replacing the size distribution of aerosols*
*in the coarse mode in CAM5 with the one that has a much narrower width in CAM6.1"*- this
seems nonsensical to me, or completely without consideration for actual coarse mode dust
widths (e.g., Ryder et al, 2013, 2018, 2019 suggest σ ∈ [1.6, 2] rather than 1.2). Why was it
decided to favour stratospheric sulfate over tropospheric mineral dust when sulfate is more
episodic (e.g. volcanic eruptions) and has less of an impact over tropospheric climate? Also, the
authors seem to recommend that the coarse mode width be reverted to 1.8 as in CAM5 (I agree),
but do not comment on the impact of resetting the coarse mode width on stratospheric sulfate.
Seeing as this was the initial motivation for contracting σ, I think that some comment is
appropriate.

That is right: we also think 1.2 is too narrow to use to represent size distribution of dust aerosol,
so we had decided to revert it to 1.8 in this work with which this reviewer also agree and
recommend using this broad size distribution in the future versions of CAM. In CAM6, the
volcanic sulfate is presented together with dust aerosol. The developers were focusing on the
volcanic sulfate while advancing the CAM model without noticing that the employed sigma is
inappropriate for dust aerosol.

We commented a little bit on this as below.

*"Our analysis suggests that the defaulted 1.2 for the geometric standard deviation of the*
*transported dust size distribution (coarse mode) may be too narrow to simulate the dust lifetime.*
In the next released model version, we recommend reverting the geometric standard deviation
to 1.8, as in CAM5, *which may require a split of representation of dust and the stratospheric*
*aerosols. It is this reversion that imposes the most important change among what we introduced*
*to CAM6.1 to the modeled dust cycle."*

[Table 1] I think that GMD should be labelled as "initialisation GMD" _as this is more descriptive.
Or is the initial GMD at source calculated online? It is difficult to tell from the text what the initial
GMS is. This also refers to L179.

Changed to "initialized GMD" here and elsewhere it is applicable. The reviewer is right that this
is initialized GMD.

[Table 1] Why is the order of the modes Accumulation, Aitken, Coarse, then Primary? Surely it
should be in ascending size order: Primary, Aitken, Accumulation then Coarse

The order in the table 1 is the same as that in the model which lists the accumulation mode ahead of the Aitken mode and primary mode at the last. In response, we reordered the list following the reviewer's suggestion.

**"Table 1.** Mode parameters for the Modal Aerosol Module version 4 (MAM4) used in CAM5 (CAM5 size) and CAM6.1 (CAM6 size) by default: geometric standard deviations (σ) and initialized geometric mean diameter (GMD) and its ranges. Values in parentheses if present are for CAM6.1 cells without parentheses are kept the same between CAM5 and CAM6.1."

| Mode (note order) | σ | *Initialized GMD (μm)* | Lower bound GMD (μm) | Upper bound GMD (μm) |
|---|---|---|---|---|
| Primary carbon (a4) | 1.6 | 0.050 | 0.010 | 0.10 |
| Aitken (a2) | 1.6 | 0.026 | 0.0087 | 0.052 |
| Accumulation (a1) | 1.8(1.6) | 0.11 | 0.054 | 0.44 |
| Coarse (a3) | 1.8(1.2) | 2.0(0.90) | 1.0(0.40) | 4.0(40) |

[Table 1] Why was the accumulation mode width changed in CAM6.1? What are the impacts of reverting it? I can't see this detail in the text

Good point. It is the same reason for this slight change as that in the coarse mode to accommodate the stratospheric aerosol (Mills et al., 2016), but our test simulations suggest negligible impacts on the dust cycle modeling when reverting it. We very briefly mentioned this in the revised manuscript (see the "Experiment design" section).

"*The other changes to the width of the accumulation mode and the bounds of the simulated GMD online impose negligible impacts on the dust cycle modeling, thus, we did not construct sensitivity tests on reverting them in this study.*"

[L109] The term 'semi-observation' is undefined and is confusing

We now specify both the observation and semi-observation as "measurements, retrievals, and model-observation integration" which brackets all the data used in this work.

[L115] "show the final summarization in Section 7". This is an unusual way to say "Discussion and conclusions are provided in Section 7" or something to that effect

We changed it to:

"*…limitations in the model-observation comparison in Sect. 5, and discussions and conclusions in Sect. 7.*"

[L120] This is one of the places in the text where it is unclear as to: (1) whether the impact of dust asphericity on MEE is represented at all, (2) if it is represented then in what way (methods), and (3) which simulations include it?

To avoid of confusion and clarify this, we removed "and optics" here, moved relevant text from the result Section 4.2.3 (5.2.3 in the revised manuscript) to Section 2.4.3, and added the following to the "Experiment design" section:

"*The enhancement of the mass extinction efficiency of aerosol particles by dust asphericity is included in all the simulations, since we do not attempt to quantify how this enhancement impacts the simulated dust cycle, which has been previously well documented (Kok et al., 2021).*"

[L137] Sentence beginning *"We consider the default DEAD scheme"* should explicitly acknowledge that it refers to emissions

Changed "scheme" to "dust emission scheme".

[L143] How confident are the authors in the critical LAI threshold? Should this assumption be discussed in the Discussion section?

The relationship of the bare soil fraction and LAI and the critical LAI threshold has been used as a standard for a while in CAM of different versions. It could be subject to change in the future, but the associated uncertainty would probably be smaller than that due to what we discussed in the Discussion section which are important missing pieces for modeling dust aerosols in CAM6. Still, we added one sentence in response to this good question.

"*This large uncertainty could partially result from the constants used in the parametrizations that affect the dust emission and transport processes, such as the critical LAI threshold, though it has been used during the past decade in different CAM versions.*"

[L152] The mass is distributed as 0.1 %, 1 % and 98.9 % between the Aitken, accumulation, and coarse modes. Surely these ratios should change depending on the assumed coarse mode width?

These values were obtained by applying the brittle fragmentation theory to the broad coarse-mode size distribution, so, they are applicable to the proposed new models. But the default CAM6.1 is using the same values while employing a much narrower coarse-mode size distribution, which could be problematic.

[L160] Many dust schemes treat dust as initially insoluble and then permitted to age via coagulation and condensation wherein it becomes soluble and internally mixed (e.g., dust in UKESM1). The authors should comment on their assumption of internally mixing dust, which may artificially enhance dust deposition near source regions? Would you expect similar results if dust is assumed to be insoluble?

The internal mixing assumption within each mode has been employed as an option in CAM since the version 5 and has been made in a huge number of studies, using CAM particularly. It worth pointing it out that dust aerosols are not completely internally mixed in MAM4 of CAM5/6: dust in different modes are externally mixed. But most dust mass is distributed in the coarse mode, which indicates that the assumption to the coarse-mode dust would be most influential on the dust cycle modeling, compared to that to the other modes. In this paper, we do not mean to document how different mixing assumptions affect the dust modeling in CAM6, since all our simulations stick to this assumption. So, we only try to briefly answer the question of this reviewer but will not expand it in the manuscript. From the view of the dust cycle modeling, we think the importance of dust hygroscopicity and its mixing with other aerosols is regionally dependent. For example, a different assumption of mixing with sea salt for South African dust can greatly change simulated deposition near the source and particularly in the downwind area. But for North African dust, it is not important near the source because both cloud fractions and sea salt concentrations are typically low. But from the view of modeling the optical properties and radiative effects of dust, the mixing states really matters.

[L165] The Neale et al (2010) reference is an internal document, which I can't find online. Can the authors please provide a URL for downloading the report, or alternatively, relevant peer-reviewed papers with the same information.

**RESPONSE:** It's a technical note. We put it in GitHub and a link in the manuscript where we cite this reference: https://github.com/L3atm/LLi2022GMD.

[L172] *"The wet deposition rate thus depends on the hygroscopicity of dust (=0.068; Scanza et al., 2015) as CCN/INPs and the prescribed scavenging coefficient (=0.1; Neale et al., 2010), both of which are currently constant with respect to the dust size (and composition for speciated dust) in CAM6.1."* _–_ I assume the hygroscopicity of dust will evolve as dust is transported through the atmosphere so I question the use of a single spatially uniform constant for this parameter. The below cloud scavenging coefficient (0.1), if it is in units of s-1, seems 2 orders of magnitude too high (Wang et al., 2010, doi:10.5194/acp-10-5685-2010). Wang et al (2010) for instance, suggest it's somewhere between 10-6 for accumulation mode aerosol and 10-3 for coarse mode aerosol depending on scavenging rate. The authors should comment more on the assumptions made in the model and the implications of those assumptions.

We appreciate the great comment and agree that the dust hygroscopicity would vary from one region to another and change during transport due to the dust ageing. But since the purpose of this paper is to document the changes, we made to CAM6.1, and how they work in effect to the dust cycle modeling, we tend to not spend too much text on commenting on all the parameterizations, such as the oversimplified hygroscopicity of dust in CAM6.1. Still, this is a very useful comment, as it could change the wet deposition rate. So, we very briefly pointed this out in the discussion section:

"*This large uncertainty could probably in part result from the constants used in the parametrizations that affect the dust emission and transport processes, such as the critical LAI threshold, the hygroscopicity of dust, and the prescribed scavenging coefficient, though the default values in the model has been used during the past decade in CAM of different versions.*"

[L180] *"Note that the current default CAM6.1 employs a narrow coarse-mode size distribution but a broad boundary width (high bound minus low bound), likely resulting in the GMD bounds less in effect, compared to that in CAM5".* – _what are the impacts of changing the coarse mode width on sea-salt emissions and sea-salt AOD? Surely this change will impact more than dust alone, which may be confounding other results presented in the study (e.g., the DRE).

This change does affect the emissions and optical depth of sea salt. We had included such impacts but then removed relevant text, since the focus of this study is on dust aerosol. Documenting sea salt seems somewhat distract the readers. To reflect this suggestion, now we mention sea salt a little bit in the last section.

"*This reverting may require a split of representation of dust and the stratospheric aerosols in the coarse mode, for which the narrow coarse-mode size distribution works better (Mills et al., 2016), and some changes to sea salt.*"

[L210] "The wet size due to growth of aerosol particles by adsorbing water vapor follows the κ-Kohler theory with a time-invariant hygroscopicity for each aerosol species (Petters and Kreidenwei, 2007)". – is it worth listing these hygroscopicity parameters to aid in the replicability of the simulations?

We will archive the model code which contains the values used for each of the aerosol species and is publicly available.

[L215] "here and hereafter unless stated otherwise" – this phrase, in parentheses, doesn't seem to apply to anything or make sense

Removed.

[L224] This is another place in the text where the impact of asphericity on the MEE is tantalisingly hinted at without further detail as to whether its on and how its incorporated

We clarified this in Section 2.5 of the revised manuscript as below, so, here removed "calculated mass extinction efficiency and".

"*The enhancement of the mass extinction efficiency of aerosol particles by dust asphericity is included in all the simulations, since we do not attempt to quantify how this enhancement impacts the simulated dust cycle, which has been previously well documented (Kok et al., 2021)*"

[L276] "In addition, the meteorology field (horizontal wind, air temperature T, and relative humidity) was nudged" – the results will obviously be changed if the model is free running then. For instance, the coarse dust will absorb LW radiation, warming the surface and destabilising the atmosphere. Perhaps this assumption (fixed meteorology) should be discussed in the Discussion section

The reviewer is right. If a free running is constructed, which we will do in the future, the results could be different. We had pointed it out that the results here are from simulations based on the use of offline dynamics in the first paragraph of the last section. To emphasis this, at some other places in the Discussion section, we inserted "offline dynamics":

"*It worth noting that the results obtained in this study rely on the models with the offline dynamics, which is subject to change while using the predicted meteorology field online.*"

"*…with the offline dynamics, the new model, CAM6.α…*"

[L285] *"Therefore, we state the difference in the dust DRE and DRE efficiency estimate in Sect. 6, but do not document the comparison of dust loadings/deposition/DOD between the bulk and speciated dust simulations."* – Avoiding comparing BULK and MINE seems like a massive oversight and is one of the first things I'd query as a reader. Does speciation between minerals improve the simulation compared to assuming dust as a bulk quantity? Simply saying that as the dust properties are different (of course they will be), this reduces comparability, is a little bit absurd and a bit of a cop out. I think this comparison should be made in a follow-on paper. To be honest, it doesn't seem worth including the MINE simulations if they not appropriately analysed.

The dust speciation helps better reproduce the observed DRE efficiency improvements compared to without the speciation, as presented in the Section 4.3.1 (5.3.1 in the revised text). For non-optical variable, summing over the eight minerals gives the total dust loadings/deposition/DOD similarly comparable to that from simulations without the dust
speciation. Per the suggestion of the reviewers, we added a new section "*6. Bulk- versus*
*speciated-dust model*" collecting information about the comparison between BULK and MINE
results that scattered in the text: s

"*This lower dust SSA obtained here in the dust speciated model than in the bulk dust model is*
*consistent to the finding of a previous study (Scanza et al., 2015) using an early model version*
*(CAM5). Correspondingly, CAM6.α_MINE yields a reduced dust cooling (Table 6) and DRE*
*efficiency (Fig. 9) than CAM6.α. For dust DRE efficiency (Fig. 9), speciating dust in CAM6 tends*
*to reduce the RMSE while retaining the horizontally spatial correlation in either SW (CAM6.α:*
*RMSE=11; R=0.26 versus CAM6.α_MINE: RMSE=10; R=0.20) or LW (CAM6.α: RMSE=4;*
*R=0.86 versus CAM6.α_MINE: RMSE=3; R=0.84) or both spectral ranges (CAM6.α: RMSE=7;*
*R=0.93 versus CAM6.α_MINE: RMSE=6; R=0.92). This comparison suggests that modeling*
*dust as component minerals with the dust size distribution in coarse mode of*
*MINE_NEW_EMIS_SIZE helps improve the model performance relative to modeling dust as a*
*bulk to reproduce the retrieved dust DRE efficiency (Fig. 9a). The improvement, however, could*
*be artificial because of the combined use of imaginary complex refractive index of hematite*
*volume (see Fig. 1b of Li et al., 2021) and the volume mixing used in the dust speciated model*
*to compute the bulk-dust complex refractive index (Li et al. in prep.), leading to artificially more*
*absorptive dust than in the bulk dust model (Fig. 9a and Table 6).*"

Also, we added RMSE and correlation coefficient in the DRE efficiency plot as shown below.

[Figure]

**"Figure 8.** Modelled and observed dust direct radiative effect efficiency in the shortwave
(SW)/longwave (LW) spectral ranges under clear conditions at the TOA over the sub-domains
(shown in the inserted map and location described below) *in April-June (AMJ), summer (JJA),*
*fall (NDJ), and September (Sep)* for the 2000s climate. The radiative effect efficiency is defined
as the ratio of the radiative effect to DOD, so has units of W m$^{-2}$ τ$^{-1}$. Included cases from left are
CAM6.1, CAM6.α, MINE_NEW_EMIS_SHAPE, CAM6.α _MINE. The field value/range are from
references listed in Table 5. *Colored numbers show correlation coefficient (R) and the root mean*
*square error (RMSE) between the model and retrievals in the SW (a) / LW (b) spectral ranges*
*or in both spectral ranges (numbers in parenthesis in Panel a).*"

[L289] Choosing to tune some models to DOD = 0.03 but not others is very peculiar. The authors
say *"Dust tuning was not applied to EXP03 and EXP04 (bulk dust simulations), in which the dust*
*emission was identical to EXP02, in order to see how changes in the transported dust size*
*distribution affects the DOD calculation".* – Well surely all of the individual sensitivity simulations
(emissions, dry deposition, asphericity) would have benefitted from the same analysis? I guess
that some parameters in the emissions and dry deposition algorithm need to tuned in some way
(so using DOD might be a reasonable approach) as the parameters have a huge degree of
uncertainty, but the asphericity probably did not need changing.

Though we did not tune EXP03 and EXZP04 which we had previously run, we scaled up DOD
and applied the same factor to the other dust quantities, as we stated in the text (the "Experiment
design" section). This rescaling makes senses, considering the roughly linear relationship
between those variables, though we acknowledge uncertainty may introduce by doing so. We
pointed this out in the Discussion section.

"…*though the linear assumption between DOD and the other dust quantities based on which we*
*rescaled up the concentrations, deposition, burdens, and DRE of dust in the size distribution*
*simulations.*"

In the emission and deposition schemes, we agree that there could maybe exist large uncertainty
in some parameters. But we would better not scale the non-tunable parameters within the dust
scheme to match the observational constraint of DOD=0.03, because the scaling factor exists
largely due to the missing sub-grid scale variability by 100-km grid-scale modeling, not because
of the uncertainty of parameters. Tuning those parameters to match the global constraint just
seems like errors compensating each other. The dust emission scheme in CAM contains a tuning
parameter "b", in the calculation of the threshold gravimetric water content, which can plausibly
range from less than 1 to the inversed clay fraction (can be > 3.0). Sensitivity tests by modifying
this tuning parameter among 0.5, 1.0, and 2.0 suggest that 1.0 is a good value to use (see Table
R1 in this report). We would not change parameters that are not introduced as tunable ones,
since they are observationally constrained. It is for this reason we did not modify parameters in the new dry deposition scheme considering that those are all non-tunable. We added the following in the "Experiment design" section and cited a new supplementary table (Table R1) there:

"*An offline sensitivity test (Table S1) supports the use of unity tuning factor to calculate the threshold gravimetric water content which we employed in the experiments for quantifying influence of each modification (speciated dust simulations listed in Table 2).*"

[L289] My other issue with this paragraph is that the tuning is not described in any detail. Which parameters were tuned and what are their values in the baseline simulation? How was tuning conducted and why was global-mean DOD chosen as the target? Simply saying 'tuned the model following Albani et al (2014)' _is not sufficient, and it would be impossible to replicate these simulations without further detail

We added the following to address this comment.

"*…we tuned the model following Albani et al., (2014) by modifying a namelist variable called soil_erod_factor, such that…*"

"*We prefer to tuning the model to reproduce the global mean DOD, 0.030, because this is currently the best estimate of global dust quantities, compared to the others (i.e., dust concentrations). It turns out that doing so can also reasonably reproduce the other quantities with no need of a regional tuning. We tuned the dust model by modifying a namelist variable in CAM, called soil_erod_factor.*"

[Table 3] This table seems very large, and I'm not sure whether the list of acronyms should be at then end of the table or in the caption. Would it be better to have 1 table for each metric?

We split this large table into 3 and list the acronyms in the caption:

"**Table** *3. Observed/retrieved cycle for dust model evaluations including optical depth, surface mass concentrations, surface deposition fluxes, and wet deposition percentages. AERONET: Aerosol Robotic Network; MODIS: Moderate Resolution Imaging Spectroradiometer; AOD: aerosol optical depth; DOD: dust optical depth.*"

"**Table** *4. Measured/retrieved dust size distribution for model evaluation. AERONET: Aerosol Robotic Network; DustCOMM: Dust Constraints from joint Observational-Modelling-experiMental analysis.*"

*"**Table** 5. Retrieved dust radiative effect efficiency for model evaluation. CERES: Clouds and the Earth's Radiant Energy System; TOA: top of the atmosphere; JJA: June, July, and August; AOD: aerosol optical depth; MISR: Multi-angle Imaging SpectroRadiometer; OMI: Ozone Monitoring Instrument; NDJ: November, December, and January; MODIS: Moderate Resolution Imaging Spectroradiometer; CALIPSO: Cloud-Aerosol Lidar and Infrared Pathfinder Satellite Observations; MFRSR: MultiFilter Rotating Shadowband Radiometer; SEVIRI: Spinning Enhanced Visible and Infrared Imager; GERB: Geostationary Earth Radiation Budget; AERONET: Aerosol Robotic Network; MPL: Micro-Pulse Lidar; AERI: Atmospheric Emitted Radiance Interferometer; SMART: Surface-sensing Measurements for Atmospheric Radiative Transfer; AMJ: April, May, and June."*

[Results] The difference between CAM6.1 and CAM6.α i.e., the control and the simulation with all changes added (except mineralogy) comes before the dissection of impacts of individual processes. Why is this? Surely it would be better to investigate the impacts of the individual processes and then use them to explain why CAM6. α is different to CAM6.1?

We think either doing what the Reviewer #2 suggested or keeping as what it was should work. In the drafted manuscript, we had tried doing the same as the reviewer suggested but then reordered the result section taking the "principle" that "the most important goes first". But in any order the conclusions of this article remain unchanged.

[L378] *"CAM6.1 may overestimate the contribution of high-latitude dust emissions to the global dust total (8.0%)."* – is this referring to the dust burden? It's rather ambiguous as is

This refers to the dust emission. We modified this sentence a little bit.

"CAM6.1 may overestimate the contribution of the high-latitude dust emission to the global dust total *emission* (8.0%)."

[L391] *"Overall, all models reproduced the climatology of DOD from AERONET retrievals, the surface concentration, and deposition within a factor of ten (Fig. 1 and Fig. S3)"* – this doesn't seem to be the case from looking at Fig. 1 b, c, e, f, h, and i. It seems that both models exhibit at least one measurement outside the range of 1/10x and 10x.

There are only 1, 4, and <10 point(s) of the 36, 47, and 108 points for DOD, surface concentrations, and deposition outside that range. That is, for over 90% of the points fall in the factor of 10. To be more accurate, we modified the sentence a little bit as follows.

*"Over 90% of the measurement sites, all models reproduced the climatology of DOD from AERONET retrievals, the surface concentration, and deposition within a factor of ten (Fig. 1 and Fig. S3)"*

[Fig. 2] Why is the new dust emissions scheme smoother in terms of emissions, rather than the delta function (almost) in DEAD? I couldn't easily find this information in the text

We added the following to answer this question.

*"The smoother distribution of the dust emission in BRIFT than DEAD is due primarily to the use of the source function in DEAD that shifts dust emissions toward the most erodible soil, while in BRIFT, the near-surface friction velocity frequently exceeds the calculated threshold wind fraction velocity, which seems low in the land model, causing dust to emit at more grid cells."*

[Fig. 3] Isn't the Ridley et al (2016) DOD dataset constrained by MODIS (either through assimilation or using it as a baseline? If so, aren't Figs 3a and 3b effectively showing the same results?

Good point. DOD of Ridley et al. (2006) "assimilated" MODIS retrievals: they corrected the bias present in MODIS retrievals (see Section 3 in the manuscript), so the former contains information of the latter, but the two datasets show considerably different results. For example, the globally averaged DOD from pure MODIS postprocessed by Pu et al. (2020) is significantly higher than the best estimate of Ridley et al. (2016) (0.025-0.035).

[L436] capture -> captures

Corrected.

[L437] Taklamakan (as in the desert) is spelt wrong throughout

Corrected.

[Fig. 4] Great figure

Thanks!

[L498] S5i -> S5e

Corrected.

[Fig. 5] This plot, especially Fig. 5a, is very confusing. There are too many colours and it is
difficult to pick out the CAM models. It may be worth plotting a non-CAM multi-model mean with
max/min as shaded in grey, and then have just the CAM models in colour
We removed non-CAM model results and cited relevant reference instead.

[Figure]

**"Figure 5.** Normalized size distribution of dust between 0.2 and 10 μm diameter in the global
average (a), near Canary Island (blue colors in b; dot: 2.5 km; x: 6-7 km; data for June/July 1997
from Otto et al., 2007), and near Cabo Verde (orange colors in c; dot: 2.5 km; x: 6-7 km; data for
August 2015 taken from Ryder et al., 2018). The default model, CAM6.1: (purple line); the new
model, CAM6.α: (red line); semi-observations: DustCOMM (black line) *inverted based on an*
*integration of a global model ensemble and quality-controlled observational constrains on the*
*transported dust size distribution, extinction efficiency, and regional DOD* with data taken
from Adebiyi et al. (2020). We chose the model layers and grid cells that are closest to the
location and atmospheric height, as well as the months, where and when the measurements
were made for comparison."
[L542] Why is the size distribution for the fine dust fraction better captured by CAM6.α?
We explained this in the revised manuscript.

*"CAM6.α can better capture due primarily to the more accurate gravitational settling velocity modeled by using the new dry deposition scheme."*

[L548] Sentence beginning "Overall, CAM6.α better reproduced the size distribution". It would be worth adding the caveat here that the Otto et al and Ryder et al measurements are from single campaigns or flights and thus may not reflect the long-term mean dust properties at those altitudes, locations, and times

Good point. We had introduced a separate section listing out limitations that are commonly presented in the model-data comparison which includes this point. But it's good to mention again at this place. So, we added the following.

*"It is worth noting that the measurements are from single campaigns or flights that may have representative issues not reflecting the climatological size and vertical distributions of dust aerosols (i.e., limited by the space and time coverage)."*

[L558] Section 4.2.1 – why are the mineralogy experiments used to test BRIFT vs DEAD rather than the BULK simulations? There doesn't appear to be any reasoning behind this

BULK and MINE runs were originally designed in two separate papers, but we ended up with this one. The results would be very similar between using BULK and MINE to test BRIFT vs DEAD and the other schemes. See our response to the general comment on BULK versus MINE (Line 61-190).

[L559] MIINE_NEW_EMIS -> MINE_NEW_EMIS

Done.

[L646] Paragraph on asphericity – I'm still confused even after reading the text as to whether the assumption of asphericity is applied to the dust MEE in every simulation run here or just the MINE_NEW_EMIS_SHAPE simulation?

We paste our response to the previous comments of this reviewer here:

Such impact of the dust asphericity is included in all the simulations because we did not attempt to quantify such effect in this study. To avoid of confusion and clarify this, we removed "and optics" here, moved relevant text from the result Section 4.2.3 (5.2.3 in the revised manuscript) to Section 2.4.3 (2.5.3 in the revised manuscript), and added the following to the "Experiment design" section:

*"The enhancement of the mass extinction efficiency of aerosol particles by dust asphericity is included in all the simulations, since we do not attempt to quantify how this enhancement impacts the simulated dust cycle, which has been previously well documented (Kok et al., 2021)."*

[L683] "(0.030-0.019)/0.030*100)" – I don't think this formula needs to be written. See also L686 and L759

Deleted.

[L693] Paragraph beginning *"The lifetime of dust"*. Should this paragraph be in Section 4.2.4? It doesn't seem to mention asphericity or apply to the MINE_NEW_EMIS_SHAPE simulation

We had included this paragraph here, because this is the section that we talked about the dust lifetime: one of the main impacts of the dust size change is on the dust lifetime. Sine this is a comparison between BRIFT and DEAD, we moved to Section 5.2.1 (revised manuscript) "Dust emission schemes: BRIFT versus DEAD".

[L705] Why is MINE_NEW_EMIS referred to as the reference case? It's a sensitivity simulation, isn't it? Surely the only reference cases are CAM6.1 and possibly MINE_BASE?

It is not a sensitivity simulation. The coarse-mode size distribution of dust in CAM6.1 was wrongly put. Thus, it seems not make a lot of senses to use CAM6.1 as the baseline simulation when quantifying the impact of each of the modifications. Please see our response to the general comment on the experiment design (Line 61-190).

[L733] *"NEW_EMIS_SIZE"* -> MINE_NEW_EMIS_SIZE. Also, this paragraph seems to be the only place where BULK and MINE are explicitly compared. I think the comparison should extend to all the dust metrics

As stated in our responses to previous comments, with the dust tuning and offline dynamics applied, speciating dust does not yield considerably different dust quantities (i.e., dust concentrations, burdens, and deposition) from BULK runs in the current climate. We added a new section to compare BULK and MINE runs:

"**6. Bulk- versus speciated-dust model**

*The bulk (CAM6.α) and dust-speciated model (CAM6.α_MINE) simulate a similar dust cycle with the difference between the two types of models orders smaller than those simulated by the former (Fig. 12 and 13: R2 and R3 in this report, respectively). This similarity results from the*

*dust tuning toward the global mean DOD of 0.03, the same meteorology dynamics both models were nudged toward, and the design of the dust-speciated model that summing the mass fraction of each dust species equals unity. With the same reasons, the influence of each of the modifications on the modelled dust cycle quantified using the bulk model instead of the dust-speciated model, as this study used, would be similarly comparable. The modelled dust optical properties, however, differs remarkably (i.e., dust SSA; Table 6) with the simulated global mean dust SSA by CAM6.α_MINE (0.896) lower than by CAM6.α (0.911) at the visible band centered at 0.53 μm. Note the dust DRE is sensitive to variation of the dust SSA. This lower dust SSA obtained here in the dust speciated model than in the bulk dust model is consistent to the finding of a previous study (Scanza et al., 2015) using an early model version (CAM5). Correspondingly, CAM6.α_MINE yields a reduced dust cooling (Table 6) and DRE efficiency (Fig. 9) than CAM6.α. For dust DRE efficiency (Fig. 9), speciating dust in CAM6 tends to reduce the RMSE while retaining the horizontally spatial correlation in either SW (CAM6.α: RMSE=11; R=0.26 versus CAM6.α_MINE: RMSE=10; R=0.20) or LW (CAM6.α: RMSE=4; R=0.86 versus CAM6.α_MINE: RMSE=3; R=0.84) or both spectral ranges (CAM6.α: RMSE=7; R=0.93 versus CAM6.α_MINE: RMSE=6; R=0.92). This comparison suggests that modeling dust as component minerals with the dust size distribution in coarse mode of MINE_NEW_EMIS_SIZE helps improve the model performance relative to modeling dust as a bulk to reproduce the retrieved dust DRE efficiency (Fig. 9a). The improvement, however, could be artificial because of the combined use of imaginary complex refractive index of hematite volume (see Fig. 1b of Li et al., 2021) and the volume mixing used in the dust speciated model to compute the bulk-dust complex refractive index (Li et al. in prep.), leading to artificially more absorptive dust than in the bulk dust model (Fig. 9a and Table 6).*"

[L798] "Overall, replacing the size distribution of dust aerosol and the dust emission scheme with new ones (PZ10 and BRIFT, respectively)" – replacing the size distribution is referred to here as PZ10 but this is the dry deposition scheme

"(PZ10 and BRIFT, respectively)" removed.

[L821] The term "space volume" is ambiguous. Possibly "colocation in space"?

Changed.

[L833] *"which can get mixed with dust aerosol particles during the transport and may not be completely excluded in the measurements."* _This seems a little lazy, do you have any estimates of how much contaminations leads to errors in measuring dust? At the moment, this point isn't backed up by evidence.

We deleted these texts, because 1) the second half of the sentence reads more like a repeat of the first half which the references we had cited serve well to support, and 2) they do not convey vital elements (we compiled the dust measurements from previous publication and use them here to evaluate the model performance. This section discussed the "Limitation in the model-observation comparison", noticing the readers that such kind of error exists in the dust measurements would be fine). Sentence now reads:

*"Finally, the modelled dust mass is for dust with our own defined mineralogy composition only (Li et al., 2021; Scanza et al., 2015), the measured mass could likely also include non-dust particles, such as sea salt (Kandler et al., 2011; Zhang et al., 2006), sulfate (Kandler et al., 2007), biomass burning aerosols (Ansmann et al., 2011; Johnson et al., 2008), or other air pollution aerosol (Huang et al., 2010; Yuan et al., 2008)."*

[L859] "… followed by the enhanced dust mass extinction efficiency at the visible band by ~30% to account for the enhancement by dust asphericity" – the asphericity applied to the MEE has not been shown to be the second most important change affected. Rather Fig. 10 shows that asphericity has a negligible impact on dust. Or is the asphericity in the MEE applied separately to the asphericity in the deposition rate? This is very confusing.

We do not plan to estimate the optical effect of the dust asphericity. That is why we include such effect in all the simulations. In response, we removed text relevant to the optical effect of the dust asphericity in the conclusion section. We also clarified how we dealt with the enhanced dust mass extinction efficiency at the visible band in the simulations in the "Experiment design" section:

**"***The enhancement of the mass extinction efficiency of aerosol particles by dust asphericity is included in all the simulations, since we do not attempt to quantify how this enhancement impacts the simulated dust cycle.***"**

[L869] "Overall, the new model can:" – is the new model, referred to in this sentence, CAM6.α? If so, why has CAM6.α_MINE been neglected? The addition of MINE to this study makes little sense as it is peripheral. Additionally, is this "new model" _already adopted for the next revision of CAM6 or is this the plan for the future?

We specified the new model. As to the modeled dust cycle, CAM6.α_MINE and CAM6.α show almost identical results. Please see our response to the general comment (Line 61-190). The modifications made to CAM6.1 to get CAM6.α is on the table. But the dust speciation is not planned yet to be included in a future CAM version.

---

## Author Response (AR1)

**Point-by-point response to the reviews**

**Referee 1**

We thank this reviewer very much for the detailed and constructive comments on this work. We have made changes to the manuscript accordingly. We colored our response in blue. Text from the manuscript is quoted with double quotation marks and new text is shown in *italics*.

**General comments**

This article presents multiple developments included in the dust cycle representation within the CAM6.1 model and assesses their impact on relevant variables, such as the dust surface concentration, deposition, size distribution, optical depth and direct radiative effect. The work conducted provides relevant information beyond the dust modeling community, as dust has impacts on different features of the atmospheric dynamics and chemistry, the climate and the Earth System. As such, I believe this article is well within the scope of the Geoscientific Model Development journal, it presents novel results, and it deserves publication.

Many thanks for the positive comments.

However, in my view, in its current form the reader has to put in a considerable effort to follow the details of the massive amount of work presented.

Thanks for the comments and time in reviewing the manuscript.

The authors present nine different experiments: five defining dust as a bulk species and four experiments considering speciated dust. This involves a duplication of experiments in which one (or several) of the new developments are tested, and adds an additional variable to the analysis, making it harder to focus on the specific impact of the new aspects included in the model.

With respect to the experiments design, the authors could better clarify the criteria used to include the new features in the tests. Instead of relying on a baseline (e.g. CAM6.1), and adding separately to that configuration the different developments (on the emissions scheme, dry deposition, size, or asphericity), the authors combine multiple developments in the different experiments. I believe these combinations could hinder a clean comparison of the effect of each development (e.g. looking at Table 4 it is difficult to know which pair of experiments allows disentangling the effect of shape and deposition changes). This issue is accentuated by the fact that the experiments are referenced along the manuscript by different names or acronyms, which further complicates tracing them.

We acknowledge that adding new developments one by one seems clearer than our
original experiment design. But it requires more simulations and thus more
computational resources while yielding a similar estimate of the impact of each
development (Fig. R1) compared to what we had presented based on our original
experiments. We had selected the original set of experiments, because adding a
modification on top of a previous change can help understand how the simulated dust
cycle evolves while updating the model (MINE_BASE) toward the most advanced one
(CAM6.α_MINE).

[Figure]

a. CAM6.α-MINE_NEW_EMIS_SHAPE: $-5.9 \times 10^{-14}$ kg m$^{-2}$ s$^{-1}$

b. PZ10-Z01: $-5.9 \times 10^{-14}$ kg m$^{-2}$ s$^{-1}$

$-700$  $-560$  $-420$  $-280$  $-140$  $140$  $280$  $420$  $560$  $700$
Dust DDF ($\times 10^{-14}$ kg m$^{-2}$ s$^{-1}$)

**Fig. R1.** Influence of change to PZ10 on the simulated dry deposition fluxes in the
dust-speciated model (change to the global annual mean of dry dust deposition: ~70
Tg) based on our experiment (a) and the suggested experiment by the reviewers (b).
Quantified change to the global annual mean of dry dust deposition equals to ~70 Tg
by either method.
The BULK runs were constructed to investigate how the incorrect dust size distribution
influences the dust cycle modeling and the estimate of dust DRE. This incorrect size
distribution has been employed in studies using the officially released BULK CAM6
and not in any study using the dust-speciated CAM. So, we do not have a good reason
to perform sensitivity tests on dust size distribution in the speciated-dust (MINE) runs.
What's more important is that quantifying the impact of individual processes, based
on the base CAM6.1 that uses an incorrect dust size distribution, seems not that
meaningful: it makes more senses to use the model with the "correct" size distribution.
That is why in all the MINE runs designed for that purpose we revert the narrow coarse-
mode size distribution to the broad one. Also, following the reviewer's experiment
design would change little to the results obtained from our experiments on the dust
cycle modeling. The reason is that the offline dynamics and the employed dust tuning ensure quite similar dust cycles modeled by BULK and MINE with different developments (Fig. R2 and Fig. R3), if the size distribution is also set to be identical, since the sum of the mass fraction for each of the eight minerals always equals unity. We had pointed out this similarity in our originally submitted manuscript: "It is worth noting that with the dust tuning applied toward the similar global mean DOD of ~0.03, the modeled dust cycle (i.e., burdens, concentrations, loadings, deposition fluxes) would be similarly comparable between the bulk and speciated dust models using the same offline dynamics and dust size distribution". Repeating the set of simulations using BULK instead to quantify the impact of each altered process would then yield similar results to what we presented in the manuscript.

[Figure]

**Fig. R2.** Surface dust emissions (a; global annual mean=2891 Tg) and deposition fluxes (b; global annual mean=2893 Tg) simulated by CAM6.α and their differences (c and d; both global annual mean=22 Tg) between CAM6.α_MINE and CAM6.α.

[Figure]

**Fig. R3.** The same as Fig. R2 but for DOD (a: global annual mean=0.030 and c: global mean difference=0.001) and dust burdens (b: global annual mean of dust mass=24 Tg and d: global mean difference≈0 Tg), respectively.

Following the Reviewer's suggestion, we added the following in the section "2.6 Experiment design**"**:

"*We quantify the impacts of the incorrect dust size distribution using the bulk-dust model because the incorrect size distribution has been employed in previous studies using the officially released bulk-dust CAM6 only but not the speciated-dust model. It is also reasonable to make all the quantifications in the model that use a correct dust size distribution. Therefore, we reverted the dust size distribution in all the speciated-dust runs to that configured in CAM5.*"

"*It is worth noting that with the dust tuning applied toward the similar global mean DOD of ~0.03, the modeled dust cycle (i.e., burdens, concentrations, loadings, and deposition fluxes) would be similarly comparable between the bulk- and speciated-dust models that nudged toward identical offline dynamics and using the same dust size distribution (see Sect. 6). The quantified effect of each of the modifications would thus be similar if using the bulk dust model instead (Fig. S2:* R1 *in this document), but the modeled dust optical properties (e.g., single scattering albedo) by the bulk and speciated dust models differ considerably, resulting in considerably different dust DRE (Scanza et al., 2015) and DRE efficiencies between NEW_EMIS (CAM6.α) and MINE_NEW_EMIS (CAM6.α_MINE).*"

*"A comparison of the bulk- and speciated-dust models on simulating dust DRE had been previously documented (Scanza et al., 2015). This study includes the speciated dust runs because we want to verify if the updates help improve the agreement with the observed dust DRE efficiency in the dust-speciated model, which could better represent the spatial variation of the dust optical properties."*

*"Note that there are many ways to conduct sensitivity studies, which could lead to slightly different results. We added the modification on top of the previous change to understand how the simulated dust cycle evolves while updating the model (MINE_BASE) toward the most advanced version (CAM6.α_MINE). This may not hinder a clean comparison of the effect of each development since the 'interaction' between the existing and newly introduced parameterizations seems weak (Fig. S2: R1 in this document)."*

To clarify how we quantify the effect of each development, we added two columns in Table 4 pointing out the size distribution used and purpose of each experiment and added the following text in the "Experiment design" section:

"We quantified the impact of each of the modifications (Z01 to PZ10, spherical to aspherical dust, and DEAD to BRIFT) on the simulated dust cycle and DRE by differentiating corresponding results in the paired simulations that contain identical developments except for the targeted modification. Specifically, we quantified the impact of changing (1) Z01 to PZ10 by taking the difference between the simulation with Z01 (MIN_NEW_EMIS_SHAPE) and that with PZ10 (CAM6.α_MIN), (2) spherical to aspherical dust between the simulation with special dust (MINE_NEW_EMIS) and that with spherical dust (MIN_NEW_EMIS_SHAPE), and, (3) DEAD to BRIFT between the simulation using DEAD (MINE_NEW_EMIS) and that using BRIFT (MINE_BASE)."

To easily trace the experiments, we now refer to them using their case names instead of EXP# all through the text.

Finally, we added a separate new section to compare results from BULK with those from MINE:

**"6. Bulk- versus speciated-dust model**

*The bulk (CAM6.α) and dust-speciated models (CAM6.α_MINE) simulate a similar dust cycle with the difference between the two types of models orders of magnitude smaller than the dust cycle itself modeled either by CAM6.α or CAM6.α_MIN (e.g., Fig. 12 and 13: R2 and R3 in this document, respectively). This similarity results from several factors.*

*1) tuning the dust cycle to a global mean DOD of 0.03;*

*2) nudging both models towards the same meteorology dynamics;*

*and 3) conserving the dust mass when speciating the dust-aerosols such that*
*summing the mass fraction of each dust species equals unity. For the same reasons,*
*the influence of each of the modifications on the modelled dust cycle quantified using*
*the bulk model instead of the dust-speciated model, as this study used, would be*
*similarly comparable.*

*What differs remarkably is the modeled dust optical properties between the speciated-*
*and bulk-dust simulations. For example, the speciated-dust model (CAM6.α_MIN)*
*yields a lower global-mean dust SSA than the bulk-dust model (CAM6.α): 0.896 versus*
*0.911 (Table 6) at the visible band centered at 0.53 μm. Note that the dust DRE is*
*sensitive to variation of the dust SSA. This lower dust SSA obtained here in the dust*
*speciated model than in the bulk dust model is consistent with the finding of a previous*
*study (Scanza et al., 2015) using an earlier model version (CAM5). Correspondingly,*
*CAM6.α_MINE yields a reduced dust cooling (Table 6) and DRE efficiency (Fig. 9)*
*relative to CAM6.α.*

*For dust DRE efficiency (Fig. 9), speciating dust in CAM6 tends to reduce the RMSE*
*while retaining the horizontal spatial correlation in either SW (CAM6.α: RMSE=11 W*
*$m^{-2}$ $τ^1$; R=0.26 versus CAM6.α_MINE: RMSE=10 W $m^{-2}$ $τ^1$; R=0.20) or longwave*
*(CAM6.α: RMSE=4.0 W $m^{-2}$ $τ^1$; R=0.86 versus CAM6.α_MINE: RMSE=3.0 W $m^{-2}$ $τ^1$;*
*R=0.84) or both spectral ranges (CAM6.α: RMSE=7.0 W $m^{-2}$ $τ^1$; R=0.93 versus*
*CAM6.α_MINE: RMSE=6.0 W $m^{-2}$ $τ^1$; R=0.92). This comparison suggests that*
*modeling dust as component minerals with the dust size distribution in coarse mode*
*of MINE_NEW_EMIS_SIZE helps improve the model performance relative to*
*modeling dust as a bulk to reproduce the retrieved dust DRE efficiency (Fig. 9a).*

*The improvement in reproducing the retrieved dust DRE efficiency, however, could be*
*artificial because of the combined use of the imaginary part of the complex refractive*
*index of hematite and the volume mixing rule used in the dust speciated model to*
*compute the bulk-dust complex refractive index (Li et al. in prep.). This combination*
*could lead to more absorptive dust than the bulk dust model (Fig. 9a and Table 6)."*

Then, I believe that a fundamental piece of this article is the variety of observations,
retrievals, model-derived products and model results that are used for the model
evaluation. The modelling community could greatly benefit from the effort done here
to compile that information and produce a benchmark for dust properties evaluation at
the global scale (in present climate). Unfortunately, these are only presented in the
article in a summarized manner (through a table). I would recommend adding in the
manuscript at least a discussion on the variables available, their usefulness for
modelled dust evaluation and their limitations.

We moved the supplementary sections to Section 3 in the revised main text and added more descriptions accordingly.

Added subsections in Section 3 include (please see contents of each of these subsections in the revised manuscript):

"*3.1 Surface dust concentrations and dust aerosol optical depth from AERONET*",

"*3.2 Surface dust deposition fluxes*",

"*3.3 Size distributions of dust aerosol*",

"*3.4 The direct radiative effect efficiency of dust*",

"*3.5 Other datasets*", and, a section to describe the metrics used for model assessment

"*4 Model assessment metrics*".

We also oriented the readers to the discussion section 7 for in-common limitations before Section 3.1:

"*Due to limitations in precisely matching the period and locations between model results and data, the evaluations focus on checking if models can capture overall features of the measured/observed/retrieved dust cycle and the corresponding dust DRE efficiency. In addition to this mismatch, we summarize limitations common in all the model-data comparisons in Sect. 7.*"

In order to lighten up the contents of the paper, I would recommend splitting the results in two different articles, one focusing on the current developments and their impact on the bulk dust cycle, and another focusing on those improvements that potentially have an impact on the mineralogy (e.g. the changes on the emission scheme).

We had tried to do this but found that having the potential impacts on the mineralogy by changing to the new dust emission scheme is not enough for a separate paper. Instead, we added a new section briefly documenting results from the MINE runs such that it makes more sense to have both BULK and MINE runs in this article.

"***6. Bulk- versus speciated-dust model***

*The bulk (CAM6.α) and dust-speciated models (CAM6.α_MINE) simulate a similar dust cycle with the difference between the two types of models orders of magnitude*

*smaller than the dust cycle itself modeled either by CAM6.α or CAM6.α_MIN (e.g., Fig.*
*12 and 13:* R2 *and* R3 *in this document, respectively). This similarity results from*
*several factors.*
*1) tuning the dust cycle to a global mean DOD of 0.03;*
*2) nudging both models towards the same meteorology dynamics;*
*and 3) conserving the dust mass when speciating the dust-aerosols such that*
*summing the mass fraction of each dust species equals unity. For the same reasons,*
*the influence of each of the modifications on the modelled dust cycle quantified using*
*the bulk model instead of the dust-speciated model, as this study used, would be*
*similarly comparable.*
*What differs remarkably is the modeled dust optical properties between the speciated-*
*and bulk-dust simulations. For example, the speciated-dust model (CAM6.α_MIN)*
*yields a lower global-mean dust SSA than the bulk-dust model (CAM6.α): 0.896 versus*
*0.911 (Table 6) at the visible band centered at 0.53 μm. Note that the dust DRE is*
*sensitive to variation of the dust SSA. This lower dust SSA obtained here in the dust*
*speciated model than in the bulk dust model is consistent with the finding of a previous*
*study (Scanza et al., 2015) using an earlier model version (CAM5). Correspondingly,*
*CAM6.α_MINE yields a reduced dust cooling (Table 6) and DRE efficiency (Fig. 9)*
*relative to CAM6.α.*
*For dust DRE efficiency (Fig. 9), speciating dust in CAM6 tends to reduce the RMSE*
*while retaining the horizontal spatial correlation in either SW (CAM6.α: RMSE=11 W*
*$m^{-2}$ $\tau^1$; R=0.26 versus CAM6.α_MINE: RMSE=10 W $m^{-2}$ $\tau^1$; R=0.20) or longwave*
*(CAM6.α: RMSE=4.0 W $m^{-2}$ $\tau^1$; R=0.86 versus CAM6.α_MINE: RMSE=3.0 W $m^{-2}$ $\tau^1$;*
*R=0.84) or both spectral ranges (CAM6.α: RMSE=7.0 W $m^{-2}$ $\tau^1$; R=0.93 versus*
*CAM6.α_MINE: RMSE=6.0 W $m^{-2}$ $\tau^1$; R=0.92). This comparison suggests that*
*modeling dust as component minerals with the dust size distribution in coarse mode*
*of MINE_NEW_EMIS_SIZE helps improve the model performance relative to*
*modeling dust as a bulk to reproduce the retrieved dust DRE efficiency (Fig. 9a).*
*The improvement in reproducing the retrieved dust DRE efficiency, however, could be*
*artificial because of the combined use of the imaginary part of the complex refractive*
*index of hematite and the volume mixing rule used in the dust speciated model to*
*compute the bulk-dust complex refractive index (Li et al. in prep.). This combination*
*could lead to more absorptive dust than the bulk dust model (Fig. 9a and Table 6)."*
Finally, I would recommend modifying the organization of some of the contents, and
re-writing or improving some parts of the text. Also, in some sections, the authors rely
excessively on external references, making it difficult to follow the discussion with the information provided in the paper itself. My recommendation would be to restructure
or adapt the article contents, such that:
(1) the previous status of the model is clearly defined and the motivation to improve
or change the specific dust representation is justified.
We slightly restructured the Introduction and did not add more content since
Reviewer #2 thinks the Introduction is highly readable (please see their comment:
Line 1357-1358 below). Please see our detailed response below (Line 363-421).
(2) the new developments are described in the current paper in a comprehensive
manner (i.e. not trusting excessively on the reader to go and check the external
references).
We introduced the key formulas used in the parameterizations, so that the readers
do not have to check those references.
(3) the evaluation methodology is explained before the presentation of results, for
instance adapting current section 3. It would be particularly useful to identify the
multiple metrics that are going to be used for the model evaluation and their purpose
(i.e. regional variability, temporal variability, etc.), comment on the dust tuning
methodology and its impact on the evaluation metrics (if any), as well as to merge
the description of the observations with the comments on section 5 about the
limitations of the datasets. Section 5 could be kept to provide an overall assessment
of the observations limitations on the main conclusions of the article.
We added a new section briefly describing the metrics used to assess the model
performance, and we kept the original Section 5 (new Section 7 in the revised
manuscript) as it was but oriented the readers to it in this section before Section 3.1:
"*Due to limitations in precisely matching the period and locations between model*
*results and data, the evaluations focus on checking if models can capture overall*
*features of the measured/observed/retrieved dust cycle and the corresponding dust*
*DRE efficiency. In addition to this mismatch, we summarize limitations common in all*
*the model-data comparisons in Sect. 7.*"
The new section reads as:
**"4 Model assessment metrics**
Metrics used to evaluate the model performance against observations include the root
mean square error (RMSE) and correlation efficient (Kendall's τ or Spearman's
Correlation). Kendall's τ and Spearman's Correlation are non-parametric methods that
do not require an assumption of data distribution, such as Gaussian or normal. For
dust deposition and loadings, correlations calculated are to assess how well models
reproduce both their regional climatology mean or one-time observation and the
seasonal cycles. Because of a lack of reliable monthly data, assessments for the dust

DRE efficiency, DOD from Rideley et al. (2016), and percentages of wet deposition in the total deposition are on spatial variability based on the regional climatology mean or one-time observations. We tested the correlation significance of the metrics at the statistical confidence level of 95%. For the dust DRE efficiency and percentages of wet deposition, some domains only have a range available, such as the Sahara Desert (15º-30ºN, 10ºW-30ºE) in the longwave spectral range. For those domains, a mean of the low and high boundaries of the range is used in the calculation of the Spearman's Correlation and the corresponding significance test."

Comments on the dust tuning methodology are now given in the "Experiment design" section, such as:

"by modifying a CAM namelist variable, dust_emis_fact, such that the simulated global mean DOD is ~0.030 at the visible band…".

Values for the tuning parameters are given in the revised Table 2.

I believe that with these changes, the article would be much easier to follow and it would reach a broader audience.

Thanks for the constructive suggestions!

**Introduction**

I believe this section could be slightly re-structured, particularly to better clarify the current model status, justify the need for improvement in the specific aspects that are dealt with in this work, and briefly explain how these are going to be approached.

We restructured the introduction to reflect these excellent suggestions:

"*As one of the widely used climate models, the Community Atmosphere Model (CAM) contains several weaknesses in modeling the dust cycle. For example,*

*1) the default scheme in CAM6.1 (Zender et al., 2003; Dust Entrainment And Deposition DEAD model, referred as DEAD) relies on an empirical geomorphic dust source function, created based on satellite retrievals of dust source regions, to model dust emissions;*

*2) the current default CESM2.1 uses the dry deposition scheme Zhang et al. (2001; Z01 hereafter) developed for particle deposition over smooth and non-vegetated surfaces. This scheme, however, underemphasizes the interception loss, the mechanism of which is less influential over the other surfaces, such as grassland. The use of the Z01 in the current default CESM2.1 is, thus, very likely overestimating the dry deposition velocity of fine-sized aerosols (diameter < 1.0 µm; referring to the*

*geometric diameter herein unless stated otherwise) and slightly underestimating that of coarse-sized aerosols (diameter > 5.0μm) (Wu et al., 2018), especially over non-vegetated surfaces (Petroff and Zhang et al., 2010);*

*3) one of the changes from CAM5 to CAM6.1 was that CAM6.1 replaced the size distribution of coarse-mode aerosols with a much narrower one (Table 1). This change was to accommodate stratospheric aerosols in the coarse mode (e.g., volcanic sulfate) compared to an early officially released version of this model (Mills et al., 2016). A recent model evaluation against satellite retrievals (Wu et al., 2020) suggests that CESM2.1-CAM6.1 worsened the dust cycle representation and stands out in simulating the relative importance of wet to dry deposition, compared with the other global climate models or model versions, such as CESM1-CAM5, due partially to the narrow coarse geometric standard deviation;*

*4) dust aerosol are typically aspherical particles in shape. The dust asphericity could lengthen the dust lifetime by ~20% compared to modeling dust as spherical particles (Huang et al., 2020). Still, CAM6.1 simulates dust as spherical particles, though the impact of dust asphericity on optical depth and resulting radiative effect of dust (Kok et al., 2017) has been previously introduced to CAM6.1 (Li et al., 2021).*

*Correspondingly, this paper describes several updates to the dust representation in CAM6.1 on the four aspects and evaluates whether and for what conditions they improve the dust model comparison to observations in the present climate. Specifically, we*

*1) replace DEAD with a new more physically based dust emission scheme, Kok et al., (2014a) previously developed for the climate models within the framework of DEAD. This scheme performs well against observations in CESM-CAM4 (Kok et al., 2014b) without the aid of the empirical geomorphic dust source function;*

*2) replace Z01 by the dry deposition scheme Petroff and Zhang et al., (2010) developed (PZ10 hereafter) to mediate the overestimation of the dry deposition velocity of fine-sized aerosols;*

*3) revert size distribution of dust aerosol particles in the coarse mode to the one previously employed in CAM5;*

*4) account for the lifetime effect of dust asphericity by decreasing the modeled gravitational settling velocity.*

*These updates are based on up-to-date knowledge of the dust properties/processes and are thus more physically realistic than the default dust parameterizations in CAM6.1/Community Land Model (version 5; CLM5)."*

**2. Model descriptions**

I would recommend starting by describing the aerosol representation in CAM6.1, as it affects both bulk dust, speciated dust and other aerosols simulated in the model.

Excellent suggestion. Per this specific comment, in the revised manuscript, we created a new section titled "*Aerosol representation*" ahead of Section 2.2 ("Bulk dust modeling"). We moved text relevant to the general aerosol representation from the "Bulk dust modeling" section to this new section.

Please, see my general comment above. Which is the added value of conducting two set of simulations (with bulk and speciated dust) for the purpose of this article (assessing changes due to deposition, emission, size distribution and shape)? If this is not justified, I would focus on this article in the bulk dust experiments, and present the speciated dust experiments elsewhere.

Thanks for this question. Please see our response to that general comment by this reviewer on the experiment design (Line 45-192).

**2.3. Dust optical properties and radiation flux diagnostic**

Please, take advantage of this section to explain aspects related to the calculation of optical properties and/or radiative variables that are currently explained in the results section (see my comments below on sections 4.3 onwards.

Done. We moved up text from Section 4.3 (5.3 in the revised version): "We augmented the longwave radiative effect from the model by 51% to account for dust scattering (Dufresne et al., 2002)", and defined the DRE efficiency in this section: "*The DRE efficiency, which we used to evaluate the model performance on simulating the dust optical properties, is defined as the ratio of dust DRE to dust optical depth (DOD) under clear conditions*".

**2.4.2. Dry deposition schemes**

The original dry deposition scheme is partly described here and partly in the introduction. I would use this section to describe the details on both the previous and the new proposed scheme. At least, I would include here the references to both schemes, and clarify if the empirical coefficients are updated in the new scheme.

Added the reference to the default Z01 scheme. These two schemes differ from each other greatly. For example, PZ10 considers additional processes, such as turbulent impaction, and accounts for more morphological characteristics of the canopy than Z01. Even for processes described in both schemes, the parameterizations are very different, such as the aerodynamic resistance (See Equation 4 of Petroff and Zhang, vs Equation 4 of Zhang et al., 2001) and Brownian diffusion (See Equation 4 of Petroff and Zhang, 2010 vs Equation 6 of Zhang et al., 2001). Consequently, these two schemes are employing two different sets of empirical coefficients. Please check the references cited here in the main text.

We provided key formulas for both parametrizations in the revised text and added descriptions of the coefficients such that the readers do not have to check external references.

**2.4.3. Dust asphericity**

Being this one of the developments listed in the article, it would be worth to include in this section at least the main characteristics of the development (e.g. factor varying according to the source region, and ranging from X to X).

We thank the reviewer for their help in improving the readability of the manuscript. Although such information was presented in the supplementary, to make it clearer, we moved some text to this section and made a revision.

"*In this calculation, we assume that the dust shape parameters are independent of the size of dust aerosol particles. Therefore, a constant revision (Eq. 35) of the dust gravitational settling velocity (the calculation in the model by default is for spherical aerosols) due to dust asphericity was applied to dust species in the three modes that contain dust aerosols (Aitken, accumulation, and coarse). The size independence assumption of dust asphericity follows the recent observational evidence that there is no statistically significant relationship between the shape parameters (aspect ratio and height-to-width ratio) and dust sizes (Huang et al., 2020). Because of highly limited measurements of dust shape parameters, we subjectively divided the dust coverage into "close-to-source", "short-range", and "long-range" zones and calculated the asphericity factor γ for each zone. The global map of the asphericity factor is shown in Fig. S1, with the value ranging between 0.82 and 0.93. We acknowledge limitation of the methodology here to account for the lifetime effect of dust asphericity, anticipating improvements on modeling this effect when more high-quality dust shape measurements are available.*"

Also, similarly, we now provide key formulas used in our calculations in the revised text. Please see "2.5.3 Dust asphericity".

Also, the authors mention the impact of the dust asphericity on optical properties (line 119). In section 4.2.3, they state that CESM2 does not include the enhancement in mass extinction efficiency due to asphericity, but that it is considered in this study (section 4.2.3). I believe the approach used to consider asphericity in the mass extinction efficiency should be clarified and described in this section (2.4.3).

We moved relevant text from the result Section 4.2.3 (Section 5.2.3 in the new version) to Section 2.4.3 (Section 2.5.3 in the new version).

We also added the following in the "Experiment design" section for clarity:

"*The enhancement of the mass extinction efficiency of aerosol particles by dust asphericity is included in all the simulations since we do not attempt to quantify how this enhancement impacts the simulated dust cycle.*"

**2.5. Experiment design**

Please, see my general comments related to the experiments' design.

I would recommend to describe first the common model configuration amongst experiments (i.e. configuration of the model components, spatial resolution, period simulated, etc.), and then identify the experiments designed to test the different developments.

Reordered the description to reflect this suggestion.

**4. Results**

Please, review and re-structure this section, see my general comment above.

We think doing what the reviewer suggested or keeping it as it was would be fine. In the drafted manuscript, we have tried doing the same as the reviewer suggested but reordered the result section taking the "principle" that "the most important things go first" since the manuscript is lengthy. In any order, the conclusions of this article would remain unchanged.
I believe using the same set of experiments to discuss all the modifications (either bulk or speciated dust) would help.

Please see our response to the comment by this reviewer on BULK versus MINE runs (Line 45-192).

In addition, a discussion focusing on the different variables, combining the multiple datasets used as a reference, rather than a separate explanation for each comparison could be of benefit. Another strategy to make easier the discussion for the reader, could be to "qualify" the sites / observations by their characteristic trait when explaining the details, e.g. source region, remote station, etc., rather than leaving it to the reader to figure out where the site is or its characteristics.

All the variables share some shortcomings in common. That explains why we have a separate Section 7 ("Limitation in the model-observation comparison") to discuss the model-data comparison. To reflect the suggestion and to make the discussion in the result sections lighter, we described more of the variables in Section 3 (" Observational datasets for model evaluations").

Added subsections in Section 3 include (please see contents of each of these subsections in the revised manuscript):

"*3.1 Surface dust concentrations and dust aerosol optical depth from AERONET*",

"*3.2 Surface dust deposition fluxes*",

"*3.3 Size distributions of dust aerosol*",

"*3.4 The direct radiative effect efficiency of dust*",

"*3.5 Other datasets*", and, a section to describe the metrics used for model assessment

"*4 Model assessment metrics*".

We also oriented the readers to the discussion section 7 for in-common limitations before Section 3.1:

"*Due to limitations in precisely matching the period and locations between model results and data, the evaluations focus on checking if models can capture overall features of the measured/observed/retrieved dust cycle and the corresponding dust DRE efficiency. In addition to this mismatch, we summarize limitations common in all the model-data comparisons in Sect. 7.*"

**4.1.1. Dust emissions**

Why compare the total emission burden with model estimates that go beyond CAM6.1 simulated size range? I believe it would be useful to include comparisons with models that use the same range (e.g. some of the AEROCOM phase I models, Huneeus et al. 2011).

Good point, though not all models that participated in AEROCOM use the same size range. As we pointed out the different size range between ours and that of Kok et al. (2021a), it would be ok to keep this small signpost: the estimate of Kok et al. (2021a), which is a constraint by available observations. Please check the references cited here in the main text.

The revised sentence reads as:

"To achieve the global mean DOD of ~0.03, CAM6.α requires a dust emission of 2891 Tg a$^{-1}$ (Table 6), which falls below the estimate of 3400-9100 Tg a$^{-1}$ by Kok et al. (2021a; their Table 1) that accounts for dust between 0.1-20 μm in diameter *and above the median, 1123 Tg a$^{-1}$, reported in AEROCOM phase I (Huneeus et al., 2011).*"

**4.1.2. Climatology annual means of [...]**

The discussion here will greatly benefit from a previous definition of the statistics, metrics, and evaluation, which I would suggest including in Section 3. In that way, the authors could make the discussion in this section lighter.

Good point. A definition of these is now included in Section 4 ("Model assessment metrics").

"*Kendall's τ and Spearman's Correlation are non-parametric methods that do not require an assumption of data distribution, such as Gaussian or normal. For dust deposition and loadings, correlations are calculated to assess how well models reproduce their regional climatology, mean or one-time observation, and seasonal cycles. Because of a lack of reliable monthly data, assessments for the dust DRE efficiency, DOD from Rideley et al. (2016), and percentages of wet in the total deposition are on spatial variability based on the regional climatology mean or one-time observations. We tested the correlation significance of the metrics at the statistical confidence level of 95%. For the dust DRE efficiency and percentages of wet deposition, some domains only have a range available, such as the Sahara Desert (15º-30ºN, 10ºW-30ºE) in the longwave spectral range. For those domains, a mean of the low and high boundaries of the range is used in the calculation of the Spearman's Correlation and the corresponding significance test.*"

The authors mention the tuning as a factor affecting the comparison of modelled DOD to MODIS and Ridley et al. (2016) products; however, this is not taken into consideration when AERONET information is used as a target. Could the tuning also have an effect on those results?

Good point. We added the following in this paragraph.

"*This overestimated DOD in the model near the source regions resulting from the tuning method may also partly explain the imperfect match between the modeled and AERONET-based DOD (Fig. 1a).*"

Does the dust wet vs. dry deposition balance in their model change with the improvements on size distribution? Could this partly be explained by an overestimation of the finer dust fractions? Or is the representation of modal internal mixtures more relevant to this process?

We added two more columns showing results from the size tests and the following sentence. To better quantify the assessment, we also added RMSE and correlation efficient for each simulation shown in the revised Table 7.

"*The models tend to overestimate the observed percentages of the wet deposition (Table 7). This overestimation could be due partly to the internal mixing assumption of dust aerosol with sea salts which increases hygroscopicity of the aerosol mixture during transport. Correcting the coarse-mode distribution, as we suggest (Table 1), does not help improve the model performance (Table 7).*"

"Table 7. Percentage (%) of wet deposition. Observations compiled by Mahowald et al., (2011b) from data at Bermuda (Jickells et al., 1998), Amsterdam Island, Cape Ferrat, Enewetak Atoll (R.Arimoto et al., 1985), Samoa; New Zealand sites (Arimoto et al., 1990); North Pacific sites (Uematsu et al., 1985); Greenland Dye 3 (Hillamo et al., 1993), Coastal Antarctica (Wagenbach et al., 1998), and Dome C of Antarctica (Wolff et al., 2006). *RMSE: root mean square error; R: Spearman's Correlation.*"

| Location | CAM6.1 [RMSE=39%; R=-0.38] | NEW_EMIS [RMSE=39%; R=-0.52] | NEW_EMIS_SIZE [RMSE=37%; R=-0.63] | CAM6.α [RMSE=37%; R=-0.31] | MINE_BASE [RMSE=34%; R=-0.45] | MINE_NEW_EMIS [RMSE=35%; R=-0.29] | CAM6.α_MINE [RMSE=36%; R=-0.38] | Observations |
|---|---|---|---|---|---|---|---|---|
| Bermuda [32ºN, 65ºW] | 92 | 91 | 81 | 87 | 81 | 85 | 87 | 17-70 |
| Amsterdam Island [38ºS, 78ºE] | 88 | 88 | 73 | 81 | 78 | 80 | 83 | 35-53 |
| Cape Ferrat [43ºN, 7ºE] | 92 | 94 | 89 | 86 | 87 | 84 | 86 | 35 |
| Enewetak Atoll [12ºN, 162ºE] | 79 | 73 | 52 | 66 | 58 | 56 | 64 | 83 |
| Samoa [14ºS, 152ºW] | 91 | 91 | 83 | 86 | 83 | 81 | 85 | 83 |
| New Zealand [35ºN, 173ºE] | 89 | 92 | 82 | 87 | 80 | 85 | 88 | 53 |
| North Pacific[a] [4º-28ºE, 162º-158ºW] | 62-90 | 71-91 | 48-80 | 53-85 | 46-80 | 48-80 | 56-84 | 75-85 |
| Greenland [65ºN, 44ºE] | 82 | 87 | 82 | 86 | 75 | 86 | 84 | 65-80 |
| Coastal Antarctica [76ºN, 25ºW] | 96 | 92 | 68 | 93 | 82 | 87 | 88 | 90 |
| Dome C. Antarctica[b] [75ºN, 123ºE] | 97 | 97 | 95 | 96 | 88 | 89 | 91 | 20[b] |

a shown are minimum and maximum of the annual wet percent among the four sites
b Non sea salt-sulfate

**4.1.4. Size distribution of transported dust**

Why is the comparison with AERONET presented in the supplement?

Thanks for the comment. We moved the figure in the supplement to the main text (Fig. 5):

"*Figure 5. Modelled and observed atmospheric size-resolved dust mass in the geometric diameter range of 1-10 µm at AERONET stations. Numbers in each plot indicate the Kendall's τ coefficient between model and observations (blue bars). The model runs here include the one using the old model with the mode size parameters from CAM6 by default (CAM6.1 in cyan) and the other one using the new model with the mode size parameters from CAM5 (CAM6.α in black). Both runs were using the offline dynamics.*"

**4.2.1. Dust emission schemes**

Please, avoid relying on excessively on external references to explain features observed among the experiments (e.g. lines 561 to 563), summarize them directly here.

We provided formulas for the new and old parameterizations in the revised text (please see Section 2.5) and cited them accordingly instead of relying on external references:

"…the dust emission coefficient in BRIFT (*Eq. 10*) and the new method of calculating the threshold gravimetric water content of the topsoil layer (*Eq. 9; see values for the tuning parameter "b" in Table 2*) shifts the main dust emission in…"

What is the impact of the dust tuning on the results? According to section 2.5, both EXP06 (MINE_BASE) and EXP07 (MINE_EMIS) were tuned to match a global DOD of around 0.03. Was that not the case? What does the re-scaling of the DOD mentioned on line 591 refer to?

The dust tuning, via a namelist variable, ensures that the global mean of the simulated DOD equals 0.030, which is one of the "best" estimates of the global dust quantities. The dust emission shown in this section is required in the model with different dust emission schemes to reach that criterion. To make this clearer and the discussion here lighter, we added the following in the revised "Experiment design" section.

"*We prefer to tuning the model to reproduce the global mean DOD of 0.030, because DOD is currently the best estimate of global dust quantities, compared to the others (i.e., dust concentrations). It turns out that doing so can also reasonably reproduce the other quantities with no need of a regional tuning. MINE_NEW_EMIS requires the dust tuning to use a much larger tuning parameter (dust_emis_fact=3.6; Table 2), than MINE_BASE (dust_emis_fact=1.6), because, otherwise, if using the same dust_emis_fact as in DEAD, the dust emissions in BRIFT would lead to an unrealistically high global mean DOD (>~0.5).*"

On Line 591 (original manuscript), the global DOD in BRIFT is lower than in DEAD (0.035 versus 0.029), because we did not retune the model to have the global DOD equal exactly to 0.030. Rescaling the dust deposition and loadings according to the factor making both global DOD equal exactly 0.030 would further reduce the difference between the dust deposition and loadings in the two experiments.

To make this clearer, we revised the sentence a little bit:

"*…differences between the global annual mean dust deposition in BRIFT and DEAD would become smaller if we rescaled the global annual mean dust deposition and*

 *loadings offline using factors to make the global mean DOD in the two experiments*
 *exactly equal 0.030."*

 **4.2.3. Dust asphericity**

 The authors state that the dust asphericity could mediate the overestimated dust
 emission from source regions, is this shown in their experiments?

 No, it is not a direct result of the experiment, but the result indicates the probably
 mediated effect. Since the dust tuning is to have global mean DOD of ~0.03,
 introducing the lifetime effect of dust asphericity to the model is expected to have the
 potential to reduce the dust emission level. We added the following text to explain it a
 little bit.

 "…dust asphericity could potentially mediate the overestimated dust emission from
 source regions (e.g., North Africa)*, because dust asphericity could enlengthen the*
 *lifetime in the atmosphere. Thus, it reduced the dust mass to have the same dust*
 *loadings and DOD as the spherical shape assumption needs.*"

 Does the asphericity factor affect differently fine vs coarse particles?

 No, the asphericity factor is the same over the three modes. This is based on the
 finding of Huang et al. (2020) that there is no statistically significant dependence of
 dust asphericity on the dust size. To clarify, we revised a sentence in the "Dust
 asphericity" section:

 "… *the asphericity factor γ (defined as the ratio of the gravitational settling velocity of*
 *aspherical dust to that of spherical dust) offline, which is independent of the dust size,*
 *based on …*"

 **4.2.4. Dust size representation**

 This section is difficult to follow, please, revise.

 Please see the revised Section 5.2.4 below:

 "The removal rates of dust aerosol particles by both dry and wet deposition highly
 depends on their size (Mahowald et al., 2014). Since most of dust loadings are in the
 coarse mode, changing parameters of the coarse-mode size distribution (σ, initialized
 GMD, and the prescribed minimal and maximum boundaries within which the modeled
 GMD can vary, Table 1) from σ=1.2 to 1.8 halved the lifetime of dust (lifetime=4.9 days
 versus 2.4 days; Table 6). This reduced dust lifetime is primarily due to the change in
 σ of the coarse mode (Fig. 8b) rather than the initialized GMD and its boundaries, as
 we obtained almost the same dust lifetime (~2.4 days) between *experiments with*

*different parameters for dust size distribution but identical σ=1.8 (NEW_EMIS_SIZE*
*versus NEW_EMIS_SIZE_WIDTH; Table 6).*

*We also notice a different DOD simulated by NEW_EMIS_SIZE (DOD=0.013) and*
*NEW_EMIS_SIZE_WIDTH (DOD=0.019). The prescribed GMD boundaries do not*
*affect the simulated dust loadings and DOD because the predicted GMD in the model*
*varies little. We can, therefore, derive that the initialized GMD itself is also relevant to*
*simulated DOD (relative change=20%) but second to changing the coarse-mode σ.*
*Thus, it is the increased σ of the coarse mode that explains the reduced dust loadings*
*(22 versus 11 Tg in NEW_EMIS and NEW_EMIS_SIZE, respectively; Table 6; Fig. 8b)*
*and DOD (0.030 versus 0.013 Tg in NEW_EMIS and NEW_EMIS_SIZE, respectively;*
*Table 6).*

The impact of changing the coarse-mode σ is greater than the other modifications (e.g.,
speciating dust or changing the dust emission scheme from DEAD to BRIFT) on the
simulated dust lifetime, which appears trivial (e.g., dust lifetime increased by 0.6 days
only by changing to the new emission scheme). Correspondingly, given a similar
emission rate, changing the coarse-mode σ affects DOD most, among the
modifications we made."

**4.3. Dust direct ratiative effect.**

Details such as the LW increase by 51% could be explained in section 2.3. I would
only mention this again here if the approach used in the different experiments would
differ, and thus affect the comparison.

Mentioned it now:

"*We augmented the longwave radiative effect from the model by 51% to account for*
*dust scattering (Dufresne et al., 2002).*"

**4.3.1. Dust direct radiative effect efficiency.**

Please, use also section 2.3 to define the net DRE efficiency.

Done: "*The DRE efficiency, which we used to evaluate the model performance on*
*simulating the dust optical properties, is defined as the ratio of dust DRE to dust optical*
*depth (DOD) under clear conditions.*"

What is the metric used here to define the model performance?

We have only several points included in this comparison. And for some, only ranges
are provided in the corresponding reference. So, we had not used any statistical metric
to measure the distance between the model and observations. But as a response, we included the correlation coefficient and RMSE with the assumption made for points where there is only a range that we use the mean in the calculations.

[Figure]

"Figure 8. Modelled and observed dust direct radiative effect efficiency in the shortwave (SW) and longwave (LW) spectral ranges under clear conditions at the TOA over the sub-domains (shown in the inserted map and location described below) in summer, fall, and September for the 2000s climate. The radiative effect efficiency is defined as the ratio of the radiative effect to DOD, so has units of W m$^{-2}$ $\tau^{-1}$. Included cases from left are CAM6.1, CAM6.α, MINE_NEW_EMIS_SHAPE, CAM6.α _MINE. *The field value/range are from references listed in Table 3. Colored numbers show correlation coefficient (R) and the root mean square error (RMSE) between the model and retrievals in the SW (a) and LW (b) spectral ranges or in both spectral ranges (numbers in parenthesis in Panel a).*"

The difference between the experiments with speciated and bulk dust is not exclusively dependent on the developments presented here, but, as the authors mention, attributed to the resulting optical properties for the different representation on the dust.

Please see our response (Line 45-192) to the general comment by this reviewer on bulk dust versus dust-speciated model. We added a new section ("6. Bulk- versus speciated-dust model": see Line 150-192 above) to compare results from the two types of models.

Does the model diagnose all sky or clear sky DRE (line 730)? Please, clarify this in
section 2.3.

We had stated that this is DRE under all sky conditions in that section: "The direct
radiative effect by dust aerosols is then determined by calculating the difference of the
net radiative flux with and without dust at the top of the atmosphere *under all-sky*
*conditions*".

To make it clearer, we revised this sentence a little bit, so now it reads as

"The direct radiative effect of dust aerosols *under all-sky conditions* is determined by
calculating the difference in the net radiative flux with and without dust at the top of
the atmosphere under all-sky conditions".

**Conclusions**

The authors mention the effect of dust asphericity on mass extinction efficiency as one
of the aspects that produces a larger change in the results, as mentioned above, it is
unclear to the reader which is the approach followed to introduce this in the model
and/or if it's introduced at all.

As stated in the original manuscript, all simulations here, including the base CAM6.1
and MINE_BASE, have considered the enhancement of dust asphericity on the mass
extinction efficiency. Previous studies have well documented such an effect. Thus, this
study does not aim at investigating it. To avoid possible confusion, we removed
relevant statements in conclusions and added a sentence in the "Experiment design"
section.
"*The enhancement of the mass extinction efficiency of aerosol particles by dust*
*asphericity is included in all the simulations since we do not attempt to quantify how*
*this enhancement impacts the simulated dust cycle.*"

I believe it would be useful to include a brief discussion on the implications of reverting
the standard deviation changes in the coarse mode for the stratospheric aerosols. If
the change was initially introduced to better accommodate those, which would be the
recommendation of the authors for the model version to be issued?

The solution could be to have a coarse mode for dust separate from the stratospheric
aerosols.

We revised relevant contents as the following in response to this suggestion.

"*Our analysis suggests that reverting the geometric standard deviation of the*
*transported dust size distribution (coarse mode) from the default 1.2 to 1.8 imposes*

*the biggest change to the modeled dust cycle among what we introduced to CAM6.1.*
*Note that the linear assumption between DOD and the other dust quantities based on*
*which we rescaled up the concentrations, deposition, burdens, and DRE of dust in the*
*size distribution simulations introduces uncertainty. Since the defaulted 1.2 is too*
*narrow to simulate the dust lifetime, in the next released model version, we*
*recommend reverting the geometric standard deviation to 1.8, as in CAM5. This*
*reverse may require to split representation of dust and the stratospheric aerosols in*
*the coarse mode, for which the narrow coarse-mode size distribution works better*
*(Mills et al., 2016), and some changes to sea salt."*

The authors comment on potential ways of improving further the dust cycle, however,
it is unclear for the reader if those stem from the work performed in this article. I would
recommend to highlight the weaknesses detected in this study concerning the dust
cycle representation (even after all the improvements included), and link to the
appropriate suggested next step to solve that issue.

We revised this part to better connect it with what we present in previous sections:

"1) *for the dust emission parameterization,* the threshold friction velocity calculated in
both BRIFT and DEAD does not account for…"

"2) …in the northern high-latitude regions *(Sect. 5.1.1),*…"

and added more text:

"*3) comparisons with the constrained global dust size distribution and measurements*
*downwind of North Africa suggest that the model underestimates dust aerosols in the*
*coarse mode with the geometric diameter > 5 µm and misses aerosol particles with*
*the geometric diameter > 10 µm (Fig. 6). The former happens likely due to an*
*underestimate of dust aerosol particles in that size range upon emissions and/or the*
*removal rate of those particles being too high during transport in the model (Adebiyi*
*and Kok, 2020b), the reason for which is still under exploration. For the latter,*
*extending the dust size range to include particles with the geometric diameter > 10 µm*
*in CAM6 is a worthy endeavor, such as in Ke et al. (2022).*

*4) as previously noted (Wu et al., 2018), some of the variables in the dry deposition*
*parameterizations could vary in different seasons for certain land cover and land use*
*types, such as the roughness length, $Z_0$, in Z01 and the displacement height of the*
*canopy, h, in PZ10, for which a fixed climatological mean is used in the models. How*
*accounting for the seasonal variation of those variables in the model can affect the*
*dust cycle modeling deserves further exploration.*

*5) compared to bulk dust, modeling dust aerosol as component minerals could better*
*reproduce the observed spatiotemporal variability of dust optical properties and thus*

 *the dust DRE efficiency (Fig. 9), while retaining the accuracy of modeling the dust*
*cycle with the offline dynamics in the present day. But the current atlas of soil*
*mineralogy and the optical properties of key minerals (i.e., iron oxides) contain large*
*uncertainties which should be better quantified in the future, such as that planned*
*in the Earth Surface Mineral Dust Source Investigation (EMIT) and in our ongoing work*
*(Li et al., in prep), respectively."*

**Technical corrections**

Please, find below a list of technical corrections that could be applied to the current manuscript version.

Thanks a lot for these technical corrections. We made corresponding changes in the revised manuscript.

L19. Either refer to the CAM6 model in the abstract (as it is in the article title) or change the title to include the CESM model.

We mentioned the CAM6 model in the abstract.

"*The Community Atmosphere Model (CAM6.1), the atmospheric component of the Community Earth System Model (CESM; version 2.1), simulates the lifecycle (emission, transport, and deposition) of mineral dust…*"

L23-24. If possible, outline the main changes included in the different parameterizations (emission, dry deposition, size distribution and dust particle shape).

We mentioned these in the revised text.

L26-27. Is it the effect of the size distribution change as large as the change in the dust emission scheme?

Great point. Changing the size distribution is more influential than changing the dust emission scheme in modeling the dust lifetime, burden, and DOD, for instance (see Fig. 10), no matter if we retune the model to have the simulated global dust AOD ~0.3. We added some words to reflect this comparison.

"In comparison, *the other modifications induced small changes to the modeled dust cycle and model-observation comparisons, except the size distribution of dust in the coarse mode, which can be even more influential than that of replacing the dust emission scheme.*"

L46. Is shape also a factor affecting the uncertainty in dust direct radiative effect?

The primary influence of shape is on the dust asymmetry factor and extinction. In our global model, we tune dust emissions to a level at which the mean DOD is around 0.03. Since the direct radiative effect roughly linearly depends on DOD, the irregular shape of dust particles would not impose an influence comparable with those we stated on this line. To make that statement more scientifically rigorous, we slightly revised this sentence, pointing out that those are primary factors.

"*These uncertainties in the dust cycle modeling, as well as uncertainties in optical properties due primarily to dust size and mineral composition…*"

L63-64. Is it necessary to mention the previous CAM and CESM versions?

In the revised manuscript, we deleted this paragraph.

L71. Why do the authors mention now the Community Land Model version 5 (CAM6.1/CLM5)? Please, use the same acronym/naming convention all along the article, either CAM6.1 or CAM6.1/CLM5, or at least, mention the full name the first time it appears and explain that from then on it will be referenced as CAM6.1.

CAM6.1 refers to the atmosphere component only of CESM, while CLM5 refers to the land component. Correspondingly, when mentioning CAM6.1, it means modifications to the atmosphere component only. But incorporating the new dust emission scheme requires us to modify both atmosphere and land components. Therefore, we need to mention both CAM6.1 and CLM5 to be scientifically rigorous.

L102 (Table 1 caption): MAM4 is mentioned for the first time. Why use two abbreviations for the standard deviation, remove extra dot after CAM6.1 in L103.

Thanks! In the revised version, we spelled MAM4 out, deleted GSD, and removed the extra dot after CAM6.1 on that line.

L108: Homogenize the naming of the sections, either Sect. or Section.

This seems a requirement by the journal: when beginning with the word, Section, one should use the full name, but in a sentence, one should use Sect. to refer.

L109: I would substitute semi-observation by more specific term(s).

We now specify both the observation and semi-observation as "*measurements, retrievals, and model-observation integration*" which should bracket all the data used in this work.

L117: Is it CESM2.1 or CESM2? Please, keep consistency in the naming of the model versions along the document.

We now use CESM2.1 all throughout the manuscript.

L125: Why is the iron solubility mentioned here?

It is redundant information, so deleted. In the original version, we also included
modeling of iron from dust, fire, and so on. But we had decided to delete it from this
manuscript since it is already a long article.

L126: I would state in the introduction that the tests are to be conducted under present
climate conditions, this will already justify using observations for the same period and
then the clarification on the pre-industrial will not be needed here.

Great suggestion. We very briefly mentioned this in the revised introduction.

"…*and for what conditions they improve the dust model comparison to observations*
*in the present climate*…"

"…*and the experiment we conducted (Sect. 2.6) under present climate conditions to*…"

L138: Please, change "models" by model.

Done.

L139: Please, remove "generally".

Done.

L141: CESM2, CESM2.1? CLM?.

Changed CESM2 to CESM2.1. As stated in previous comment on CAM6.1/CLM, here
we think CLM is better to be kept as it was.

L153: Please, rephrase to specify the variable that is independent of the friction
velocity (rather than the theory itself).

Good point. The revised statement now reads as:

"*The size distribution of the emitted dust is derived using the brittle fragmentation*
*theory developed by Kok (2011b) distributing 0.1%, 1.0%, and 98.9% percentage of*
*dust mass into Aitken, accumulation, and coarse modes, respectively, independent of*
*the friction velocity upon dust emissions (Kok, 2011a).*"

L154-155: As it is expressed now, the improvement in CAM4 size distribution is not informative to the reader. Please, either remove the part about the improvements or to briefly explain the difference between the approaches in previous CAM4 PSD and that derived from Kok (2011).

Deleted.

L156: Please, remove "other", and "of aerosols".

Done.

L179: Please, remove "the so-called".

Done.

L178: As mentioned above, please, select just one acronym for the standard deviation.

Using only one now.

L189: Please, change "their ranges", by "its ranges".

Done.

The reference to Scanza et al. (2015) was already included.

Reference deleted.

L221: Is the vertical transport modified per se? Or is it indirectly affected by changes in emission/size?

We did not modify it per se. The change is indirect due to changes in dust emissions and size. In the first version of this manuscript, we also perturbed the vertical layers, which can affect vertical transport more efficiently, but we deleted that part after. In response to the reviewer's question, here we removed "vertical transport" to avoid possible confusion.

L231: What do the authors mean by "although even dust modeling with BRIFT can be improved if optimized against observations", is that optimization relevant for this specific study?

A typo caused this confusion. We corrected the typo in the revised manuscript and replaced BRIFT with DEAD. It means improvements could likely happen using other methods, such as statistical optimizations (Kok et al., 2021), rather than employing the new dust emission scheme.

Please check the references cited here in the main text.

L328: Please, avoid repeating references unnecessarily (e.g. remove described in
Sect. 2.2).

Repeated references removed.

L333: There are two references for Kok et al. (2021), please, specify a or b.

Done.

L338: Please, change "could change", by the appropriate: does or does not change
the model performance?

Paragraph removed in response to the next comment.

L338-343. May not be necessary to explain again the content of each sub-section.

Removed the navigation paragraph.

L358: Please, explain what the binned method is.

It is a terminology that the dust community widely uses without a definition. Also, this
study does not employ the binned method. So, we believe it would be fine without
explaining it here as well.

L466 (and other locations in the text): Please, refer to the different experiments as
such, instead of mentioning the models. If preferred by the authors, they could use
model versions.

We changed "models" to "*all experiments*" here and at other locations in the text as
well.

L369: Please, identify the reference with a or b.

Done. Changed to "Kok et al. (2021a)".

L432: Typo: averages.

Corrected. Thanks!

L439: Change "to the low" by "to the *lower*".

Done. Thanks!

L475: Have the authors information on the precipitation evaluation for their own model?

No, but CAM6 had been fully evaluated over aspects including precipitation.

L524: Please, include the coordinates of both stations or none.

The coordinates of both stations included.

L543: Does the super coarse dust start at 10 um? or larger diameters?

There is no clear boundary between coarse and super coarse particles. Here we refer
to particles >10 µm in diameter, not including the 10 µm. Since Table 1 lists the coarse
dust and does not define "super coarse dust" clearly defined, we removed "super
coarse" to avoid possible confusion. So, now only keep expressions like "dust coarser
than 10 µm in diameter" here and elsewhere in the text.

For example, we changed "the super coarse dust particles are also…" to "*dust*
*particles in this size range are also…*".

L614: Hematite and illite have a high iron content, feldspars not much. The sentence
could be rephrased as ", including hematite and illite, and feldspar"

Rephrased. Thanks!

L636: I believe the increase is in wet deposition (not dry), please, verify.

Fig. 6c suggests the increase is in the dry deposition. This increase could probably
stem from the release of fine-mode particles by evaporation of the cloud-borne dust.
We revised the statement, such as it reads now as:

"…*which then become cloud-borne. The increased cloud-borne particles in turn*
*increase the possibility of horizontal transport and release of particles by the cloud*
*droplet evaporation, leading to an increase of…*"

L655: Please, include the full reference and then in parenthesis the values.

Done. It reads now as:

"…*between the global mean DOD in Aerosol Comparisons between Observations and*
*Models (AEROCOM; median: 0.023) (Huneeus et al., 2011) and that in Ridley et al.*
*(2016) (0.03±0.005) near the visible band.*"

L683: Why is the calculation explicitly included there? It makes the text more difficult to read. I would avoid it (here and in other locations in the text below).

We removed this kind of expressions everywhere.

L699: The sentence "where the dust emission occurs in transport" is difficult to understand, please, clarify.

Changed it to "*the importance of accurately simulating convergence-related convection (i.e., haboob) (Marsham et al., 2011) and where the dust emission occurs for dust transport modeling…*"

L869: Substitute "new model" by the appropriate model version name.

The new model version name inserted.

Table 1: I would order the modes from smaller to larger in size. I believe this table could be included in the supplement and leave in the text exclusively the default and new configuration for the coarse mode.

The order in Table 1 is the same as that in the model. We included this table in the main text because we wanted to inform the readers about the mode information, for which they may search while reading through the main text, especially considering that the mode change is one of the main changes we made to the model.

Table 4: Why is the dust SSA for NEW_EMIS_SIZE missing?

When designing and performing simulations, we did not attempt to address the impacts of these changes on the dust radiative effect. So, we had not requested model output for this variable in that single experiment. According to dust SSA from the other experiments shown in this Table, we speculate a value around 0.90 for this experiment. But, since we did not show the model-data comparison for dust SSA, we believe the missing of dust SSA in this single experiment may not influence the overall merit of this work.

Please, homogenize the naming convention for the different experiments, here tagged in Table 4 as NEW_EMIS, NEW_EMIS_SIZE, etc. In Table 2 and sections 2.1 and 2.2 they were listed also as EXP01, EXP02, etc. In Table 4 caption CAM6S5 and CAM6S6 are mentioned, which were not identified nor described before.

The case names are all consistent throughout the text now. We also revised the "size" column in Table 2 since those notions are no longer in use.

Table 5: Could the locations be represented in a map, together with the other
observations location?

We provided such information in the revised table (first column). But we did not show
that for each set of the observations in a map together with location information of the
others since the map would be super busy and very confusing, considering the number
of data sets we have included in this work.

Figure 1: Which is the metric used to define the improvement (+) or worsening (-) of
the comparison? Remove the comment on Figure S3 from the caption, and if needed,
clarify in the text (line 392) the information presented in main paper and in the
supplement.

The citation of Fig. S3 removed, and the metric used clarified in the figure.

[Figure]

"Figure 1. Model-observation (AERONET) comparison for DOD (dust optical depth) at
the visible band centered at 0.53 µm (a, b, and c), dust surface concentrations (d, e,
and f), and surface deposition fluxes (g, h, and i). Colored dots in a, d, and g show the
difference between the proposed new model (CAM6.α) and observations. White
symbols indicate the new model CAM6.α improves (plus sign) or worsens (minus sign)
the model-observation comparison over that between the default model (CAM6.1) and
observations *with the metric included in the bottom right-hand corner of the figure*.
Numbers listed in a, d, and g are counts of the number of improved or worsen stations.
The spatial correlation coefficients between model (CAM6.1: b, e, and h; CAM6.α: c, f, and i) and observations were calculated based on the annual mean values in log space (the log of each model and observational value was taken before calculating the correlation coefficient, since the values span several orders of magnitude except DOD). Dash lines in the scatter plot show 10:1 or 1:10 lines."

Figure 2: Could the re-scaling factors now explained in the caption be included also in the figure legend (e.g. above each map)?

Added. Please see Fig. 2 in the revised manuscript.

Figure 5: Please, review the caption: remove "and" in the third line, remove "for the abbreviation for other models", either explain them there or leave just the reference, specify what do we understand by semi-observations. Please, do not refer to other figures in figure captions unless they are needed to understand the figure contents.

Removed. We also added the following in the caption:

"…*semi-observations: DustCOMM (black line) inverted based on an integration of a global model ensemble and quality-controlled observational constrains on the transported dust size distribution, extinction efficiency, and regional DOD*"

Figure 6: What do the maps represent? Is it the ratio? Or the differences over the reference?

We believe the caption for Panel a-h is clear on this. "Figure 6. Impacts of the dust emission scheme (a and b: **ratio of BRIFT to DEAD**), aerosol dry deposition scheme (c-f: **ratio of PZ10 to Z01**), and dust shape (g and h: **ratio of ellipsoidal to spherical dust**) on the modeled dust deposition (total: a, d, and g; fine mode: c), and dust loading (total: b, f, and h; fine mode: e)."

Figure 7: Please, use the same naming convention for the different experiments along the manuscript, otherwise is very confusing.

Done.

Figure 8: Homogenize the experiment names with the rest of the document, review the seasons listed in the caption, the inserted map below is not shown in this document version.

Changed relevant text to:

"Figure 8. Modelled and observed dust direct radiative effect efficiency in the shortwave (SW) and longwave (LW) spectral ranges *under clear conditions at the TOA over the sub-domains (location described as [lat, lon]) in April-June (AMJ), summer*

*(JJA), fall (NDJ), and September (Sep) for the 2000s climate.* The radiative effect efficiency is defined as the ratio of the radiative effect to DOD, so has units of W m$^{-2}$ $\tau^{-1}$. Included cases from left are CAM6.1, CAM6.α, MINE_NEW_EMIS_SHAPE, CAM6.α _MINE. The field value/range are from references listed in Table 3. *Colored numbers show correlation coefficient (R) and the root mean square error (RMSE) between the model and retrievals in the SW (a) and LW (b) spectral ranges or in both spectral ranges (numbers in parenthesis in Panel a)."*

**Referee 2**

We thank this reviewer very much for the detailed and constructive comments on this work. We have made changes to the manuscript accordingly. We colored our response in blue. Text from the manuscript is quoted with double quotation marks and new text is shown in *italics*.

In this paper, Li *et al.* investigate the sensitivity of dust in the CESM2-CAM6.1 climate model to various parameterized processes: the emissions scheme, the dry deposition scheme, the fixed geometric width of the coarse mode, and the assumption of spherical/aspherical particles. Using a wealth of validatory observations and many simulations, they find that changing dust emissions and the coarse mode width have the greatest impact on the dust metrics, followed by the dry deposition scheme and then asphericity. They also propose a new version of CAM (CAM6.α) which improves on many dust metrics relative to CAM6.1 and incorporates some of the listed process changes.

The paper is well written and contains a wealth of useful information, including the most comprehensive database of dust observations yet (Table 3). The introduction is highly readable, and the conclusions are generally supported by the analysis. However, this paper rather feels like 3 independent studies convoluted together, namely, (1) a new and improved version of the dust scheme in CAM (CAM6.1 versus CAM6.α), (2) a study of the sensitivity of simulated dust to certain processes, and (3) a study of the merits of separating dust into its mineralogical components in CAM. I think the paper would benefit from being split into 2 or 3 separate papers, which I expand on below in the General Comments. In short, I think that the study needs a redesign before it is published, which may require major revisions (i.e., new simulations and a re-write) and/or splitting into separate papers.

We appreciate the positive comments very much. The reviewer correctly pointed out that this is convoluted by independent studies. Our original plan, however, was to separately document the size change in BULK CAM6 and the improved emission and deposition parameterizations in CAM6. Previous studies have shown the merits of modeling dust as mineral components in terms of the climatic impacts of mineral dust, so we think it would not deserve a new paper on this. Since we tend to update separate processes in CAM6 and the new schemes have been detailed and tested offline or in previous versions of CAM (CAM4 and CAM5), it makes more sense to document in the same paper how the change to each process may affect the dust cycle modeling. Please see our reply to the comments below.

Firstly, I think that the simulation design is incorrect for exploring the sensitivity of dust to the altered processes. For example, the new dry deposition scheme is only tested in conjunction with the other altered processes (CAM6.α and CAM6.α_MINE) and never on its own. Conversely, the new emissions scheme is tested by itself for both

BULK and MINE dust models, whilst the size and shape of the particles are tested in conjunction with the new emissions scheme but using BULK and MINE dust respectively. In short, it's very difficult to attribute the impacts on the dust metrics to the individual processes.

I would suggest concentrating on either the BULK dust scheme or the MINE dust scheme, unless you plan to directly compare them. The study would be much cleaner if the processes were tested in isolation using either BULK or MINE and then compared to CAM6.1 (see Table below). In its current form, it is very difficult to disentangle which dust impacts emanate from which altered process.

Suggested simulations:

| Simulation | Name | Description |
|---|---|---|
| 1 | CAM6.1 | Standard model |
| 2 | NEW_EMISS | CAM6.1 with BRIFT emissions |
| 3 | NEW_SIZE_S5 | CAM6.1 with CAM5 size assumptions |
| 4 | NEW_SIZE_S6σ5 | CAM6.1 with CAM5 assumptions except coarse σ from CAM6.1 |
| 5 | NEW_DRYDEP | CAM6.1 with PZ10 dry deposition |
| 6 | NEW_SHAPE | CAM6.1 with aspherical dust |
| 7 | CAM6.1_MINE | Equivalent to MINE_BASE but may use CAM6.α as BASE simulation |
| 8 | CAM6.α | CAM6.1 with all of the relevant model changes |

In summary, I would highly recommend that the authors run further simulations with each of the processes applied separately as the current simulation design is not conducive or particularly supportive of the results presented in the manuscript.

This is a similar comment to what the first Reviewer raised. Below we paste our reply to the comment by Reviewer # 1 as a response.

There are a couple of different methods to estimate the effect of each development, such as the one we used and the one the reviewers suggested. Strictly speaking, either method cannot totally exclude the possible influence of the parametrizations that had already been included and can affect the dust cycle modeling in the base model. The reason is that there likely exists a nonlinear "interaction" between the existing parameterizations and the newly introduced one, which seems weak though.

We acknowledge that adding new developments one by one seems clearer than our original experiment design. But it requires more simulations and thus more computational resources while yielding a similar estimate of the impact of each development (Fig. R1) compared to what we had presented based on our original experiments. We had selected the original set of experiments, because adding a modification on top of a previous change can help understand how the simulated dust cycle evolves while updating the model (MINE_BASE) toward the most advanced one
(CAM6.α_MINE).

[Figure]

a. CAM6.α-MINE_NEW_EMIS_SHAPE: -5.9x10$^{-14}$ kg m$^{-2}$ s$^{-1}$

b. PZ10-Z01: -5.9x10$^{-14}$ kg m$^{-2}$ s$^{-1}$

−700 −560 −420 −280 −140 140 280 420 560 700
Dust DDF (x10$^{-14}$ kg m$^{-2}$ s$^{-1}$)

**Fig. R1.** Influence of changing to PZ10 on the simulated dry deposition fluxes in the
dust-speciated model (change to the global annual mean of dry dust deposition: ~70
Tg) based on our experiment (a) and the suggested experiment by the reviewers (b;
Simulation 5 – simulation 1). Quantified change to the global annual mean of dry dust
deposition equals ~70 Tg by either method.
The BULK runs were constructed to investigate how the incorrect dust size distribution
influences the dust cycle modeling and the estimate of dust DRE. This incorrect size
distribution has been employed in studies using the officially released BULK CAM6
and not in any study using the dust-speciated CAM. So, we do not have a good reason
to perform sensitivity tests on dust size distribution in the speciated-dust (MINE) runs.
What's more important is that quantifying the impact of individual processes, based
on the base CAM6.1 that uses an incorrect dust size distribution, seems not that
meaningful: it makes more senses to use the model with the "correct" size distribution.
That is why in all the MINE runs designed for that purpose we revert the narrow coarse-
mode size distribution to the broad one. Also, following the reviewer's experiment
design would change little to the results obtained from our experiments on the dust
cycle modeling. The reason is that the offline dynamics and the employed dust tuning
ensure quite similar dust cycles modeled by BULK and MINE with different
developments (Fig. R2 and Fig. R3), if the size distribution is also set to be identical,
since the sum of the mass fraction for each of the eight minerals always equals unity.
We had pointed out this similarity in our originally submitted manuscript: "It is worth
noting that with the dust tuning applied toward the similar global mean DOD of ~0.03,
the modeled dust cycle (i.e., burdens, concentrations, loadings, deposition fluxes)

would be similarly comparable between the bulk and speciated dust models using the
same offline dynamics and dust size distribution". Repeating the set of simulations
using BULK instead to quantify the impact of each altered process would then yield
similar results to what we presented in the manuscript.

[Figure]

**Fig. R2.** Surface dust emissions (a; global annual mean=2891 Tg) and deposition
fluxes (b; global annual mean=2893 Tg) simulated by CAM6.α and their differences (c
and d; both global annual mean=22 Tg) between MINE_ CAM6.α and CAM6.α.

[Figure]

**Fig. R3.** The same as **Fig. R2** but for DOD (a: global annual mean=0.030 and c: global mean difference=0.001) and dust burdens (b: global annual mean of dust mass=24 Tg and d: global mean difference≈0 Tg), respectively.

Following the Reviewer's suggestion, we added the following in the section "2.6 Experiment design**"**:

"*We quantify the impacts of the incorrect dust size distribution using the bulk-dust model because the incorrect size distribution has been employed in previous studies using the officially released bulk-dust CAM6 only but not the speciated-dust model. It is also reasonable to make all the quantifications in the model that use a correct dust size distribution. Therefore, we reverted the dust size distribution in all the speciated-dust runs to that configured in CAM5.*"

"*It is worth noting that with the dust tuning applied toward the similar global mean DOD of ~0.03, the modeled dust cycle (i.e., burdens, concentrations, loadings, and deposition fluxes) would be similarly comparable between the bulk- and speciated-dust models that nudged toward identical offline dynamics and using the same dust size distribution (see Sect. 6). The quantified effect of each of the modifications would thus be similar if using the bulk dust model instead (Fig. S2: R1 in this document), but the modeled dust optical properties (e.g., single scattering albedo) by the bulk and speciated dust models differ considerably, resulting in considerably different dust DRE (Scanza et al., 2015) and DRE efficiencies between NEW_EMIS (CAM6.α) and MINE_NEW_EMIS (CAM6.α_MINE).*"

*"A comparison of the bulk- and speciated-dust models on simulating dust DRE had been previously documented (Scanza et al., 2015). This study includes the speciated dust runs because we want to verify if the updates help improve the agreement with the observed dust DRE efficiency in the dust-speciated model, which could better represent the spatial variation of the dust optical properties."*

*"*Note that there are many ways to conduct sensitivity studies, which could lead to slightly different results. *We added the modification on top of the previous change to understand how the simulated dust cycle evolves while updating the model (MINE_BASE) toward the most advanced version (CAM6.α_MINE). This may not hinder a clean comparison of the effect of each development since the 'interaction' between the existing and newly introduced parameterizations seems weak (Fig. S2:* R1 *in this document).*"

To clarify how we quantify the effect of each development, we added two columns in Table 4 pointing out the size distribution used and purpose of each experiment and added the following text in the "Experiment design" section:

"We quantified the impact of each of the modifications (Z01 to PZ10, spherical to aspherical dust, and DEAD to BRIFT) on the simulated dust cycle and DRE by differentiating corresponding results in the paired simulations that contain identical developments except for the targeted modification. Specifically, we quantified the impact of changing (1) Z01 to PZ10 by taking the difference between the simulation with Z01 (MIN_NEW_EMIS_SHAPE) and that with PZ10 (CAM6.α_MIN), (2) spherical to aspherical dust between the simulation with special dust (MINE_NEW_EMIS) and that with spherical dust (MIN_NEW_EMIS_SHAPE), and, (3) DEAD to BRIFT between the simulation using DEAD (MINE_NEW_EMIS) and that using BRIFT (MINE_BASE)."

To easily trace the experiments, we now refer to them using their case names instead of EXP# all through the text.

Finally, we added a separate new section to compare results from BULK with those from MINE:

**"6. Bulk- versus speciated-dust model**

*The bulk (CAM6.α) and dust-speciated models (CAM6.α_MINE) simulate a similar dust cycle with the difference between the two types of models orders of magnitude smaller than the dust cycle itself modeled either by CAM6.α or CAM6.α_MINE (e.g., Fig. 12 and 13:* R2 *and* R3 *in this document, respectively). This similarity results from several factors.*

*1) tuning the dust cycle to a global mean DOD of 0.03;*

*2) nudging both models towards the same meteorology dynamics;*

*and 3) conserving the dust mass when speciating the dust-aerosols such that summing the mass fraction of each dust species equals unity. For the same reasons, the influence of each of the modifications on the modelled dust cycle quantified using the bulk model instead of the dust-speciated model, as this study used, would be similarly comparable.*

*What differs remarkably is the modeled dust optical properties between the speciated- and bulk-dust simulations. For example, the speciated-dust model (CAM6.α_MIN) yields a lower global-mean dust SSA than the bulk-dust model (CAM6.α): 0.896 versus 0.911 (Table 6) at the visible band centered at 0.53 μm. Note that the dust DRE is sensitive to variation of the dust SSA. This lower dust SSA obtained here in the dust speciated model than in the bulk dust model is consistent with the finding of a previous study (Scanza et al., 2015) using an earlier model version (CAM5). Correspondingly, CAM6.α_MINE yields a reduced dust cooling (Table 6) and DRE efficiency (Fig. 9) relative to CAM6.α.*

*For dust DRE efficiency (Fig. 9), speciating dust in CAM6 tends to reduce the RMSE while retaining the horizontal spatial correlation in either SW (CAM6.α: RMSE=11 W $m^{-2}$ $τ^1$; R=0.26 versus CAM6.α_MINE: RMSE=10 W $m^{-2}$ $τ^1$; R=0.20) or longwave (CAM6.α: RMSE=4.0 W $m^{-2}$ $τ^1$; R=0.86 versus CAM6.α_MINE: RMSE=3.0 W $m^{-2}$ $τ^1$; R=0.84) or both spectral ranges (CAM6.α: RMSE=7.0 W $m^{-2}$ $τ^1$; R=0.93 versus CAM6.α_MINE: RMSE=6.0 W $m^{-2}$ $τ^1$; R=0.92). This comparison suggests that modeling dust as component minerals with the dust size distribution in coarse mode of MINE_NEW_EMIS_SIZE helps improve the model performance relative to modeling dust as a bulk to reproduce the retrieved dust DRE efficiency (Fig. 9a).*

*The improvement in reproducing the retrieved dust DRE efficiency, however, could be artificial because of the combined use of the imaginary part of the complex refractive index of hematite and the volume mixing rule used in the dust speciated model to compute the bulk-dust complex refractive index (Li et al. in prep.). This combination could lead to more absorptive dust than the bulk dust model (Fig. 9a and Table 6)."*

Another issue that I had with the simulation design was the arbitrary tuning of dust optical depth (DOD) to 0.03 in some simulations but not in others (L289). This made it very difficult to quantify the impact of the altered processes and forced the authors to add caveats throughout the text e.g., L590 "differences between the global annual mean dust deposition in BRIFT and DEAD would become smaller, if we rescaled the value according to the same DOD criteria". I suggest only tuning CAM6.1 and CAM6.α to 0.03 and using the tuned CAM6.1 as the BASE model in which to add the different processes incrementally. I see no need to rescale DOD in the sensitivity simulations and it would be interesting to see the impact of the different processes on the global-
mean DOD as a derived product of the models. Tuning to 0.03 is arbitrary and also
misses the fact that much of the dust mass is in the super coarse mode which is
missing from the model, and therefore the model may be wrongly tuned to 0.03.
We tuned CAM6.1 and CAM6.α, as this reviewer also suggested, toward 0.030. But
we must retune the model that uses the updated dust emission scheme simply
because if using the same tuning parameter value as in the model with DEAD, the
global mean DOD would be >15 times higher than that in DEAD, reaching up to 0.45,
which is undoubtedly unrealistic. We added the following to the manuscript.
"*MINE_NEW_EMIS requires the dust tuning to use a much larger tuning parameter*
*(dust_emis_fact=3.6; Table 2), than MINE_BASE (dust_emis_fact=1.6), because,*
*otherwise, if using the same dust_emis_fact as in DEAD, the dust emissions in BRIFT*
*would lead to an unrealistically high global mean DOD (>~0.5).*"
The MINE runs are not for sensitivity studies but for quantifying how each modification
affects the dust cycle modeling. We would obtain the same results if performing BULK
runs because, with the same model configurations set in this study, the BULK and
MINE simulations are nearly identical for modeling the dust cycle. We pointed this out
in the revised manuscript as below.
"*With the dust tuning applied toward the similar global mean DOD of ~0.030, the*
*modeled dust cycle (i.e., burdens, concentrations, loadings, and deposition fluxes)*
*would be similar between the bulk- and speciated-dust models that are nudged toward*
*identical offline dynamics and using the same dust size distribution (see Sect. 6). The*
*quantified effect of each of the modifications would thus be similar if using the bulk*
*dust model instead (Fig. S2),…*"
As to the dust mass distribution concerning dust size, according to a recent study (Di
Biagio et al., 2020), for a total of 39 Tg dust, approximately 33% (13) Tg dust are
particles >10 μm, though we obtain such estimates based on model simulations.
However, this missing fraction of "super-coarse" dust constitutes only a fraction of the
total DOD <2% which is even much smaller than the uncertainty in the best estimate
from Ridley et al. (2016). Therefore, we believe missing that dust mass would not
affect the accuracy of tuning dust toward DOD of ~0.030.
In response to the reviewer's question about dust tuning, we added some sentences
to explain why and how we tuned the model to get the global mean DOD of ~0.030.
"*We prefer to tuning the model to reproduce the global mean DOD of 0.030, because*
*DOD is currently the best estimate of global dust quantities, compared to the others*
*(i.e., dust concentrations). It turns out that doing so can also reasonably reproduce the*
*other quantities with no need of a regional tuning. We tuned the dust model by*

*modifying a namelist variable in CAM, called soil_erod_factor, corresponding to λ in*
*Eq. (16).*"

Regarding the reviewer's suggestion to include the updates one by one, please see
our response to the previous comment by this reviewer on the experiment design (Line
1404-1508).

It is also confusing for the reader that some simulations have emissions scaled by
1/f_clay whilst others have the scaling as 1, and so the impact of this change is difficult
to disentangle using the current suite of simulations. It would be better if this factor is
consistent across the simulations or tested in isolation.

We thank the reviewer for the comment, which makes us realize that our writing may
be confusing. The parameter b is set to be $1/f_{clay}$ as part of DEAD in the default CAM6
but is set to be unity in BRIFT to better reproduce the observations. There are also
other parameters not shared between the two schemes in addition to the different
values used for b.

As a response, we provided formulas for both emission schemes (please see Section
2.5 in the revised manuscript). In Table 2, we added a new column showing the b
value used in each experiment.

[revised manuscript text omitted]

*"Table S1 (*R1 *in this document). Comparison of the three CESM simulations with the*
*offline dynamics and different values of the tuning parameter (b) to calculate the*
*threshold gravimetric water content in the new dust emission scheme, against*
*measurements. The measurements include AERONET AOD climatology, surface dust*
*concentrations, and dust deposition fluxes, as described in Section 3."*

| Parameter b | Correlation coefficient (RMSE) on climatology | | |
|---|---|---|---|
| | AERONET DOD | Surface dust concentrations (log space) | Dust deposition fluxes (log space) |
| 0.5 | 0.74 (0.13) | 0.83 (0.66) | 0.72 (0.93) |
| 1.0 | 0.68 (0.14) | 0.82 (0.72) | 0.77 (0.86) |
| 2.0 | 0.66 (0.14) | 0.83 (0.66) | 0.79 (0.82) |

I gather from the text (L649) that the impact of asphericity on the dust mass extinction
efficiency (MEE) is represented in *all* of these simulations. This is rather confusing, as
it suggests some representation of asphericity is incorporated even when dust is
assumed to be spherical (?). Please clarify this for the reader. In particular, please
state whether the impact of asphericity on MEE is only applied in the simulation with
dust asphericity or in all simulations (which seems inherently wrong). Really these
details should be included in the Methods (L98, L224) and not in the result section.

All the simulations account for such an impact of dust asphericity. To avoid confusion,
we moved relevant text from the result Section 4.2.3 (Section 5.2.3 in the revised
version) to Section 2.4.3 and added the following to the "Experiment design" section:

*"The enhancement of the mass extinction efficiency of aerosol particles by dust*
*asphericity is included in all the simulations since we do not attempt to quantify how*
*this enhancement impacts the simulated dust cycle."*

In terms of the presentation of the results, I thought that comparing CAM6.α _with
CAM6.1 before looking at the individual processes was confusing, as much of the
analysis of the impacts of individual processes could have been used to explain
differences between the dust metrics in CAM6.1 and CAM6.α.

We think doing what the reviewer suggested or keeping it as it was would be OK. In the drafted manuscript, we have tried doing the same as the reviewer suggested but reordered the result section taking the "principle" that "the most important things go first" since the manuscript is lengthy. In any order, the conclusions of this article would remain unchanged.

Additionally, the authors say the following in Section 2.5:

*"It is worth noting that dust burdens and deposition fluxes would be comparable, if the bulk and speciated dust models have similar DOD. But the dust optical properties (e.g., single scattering albedo) in the bulk and speciated dust simulations differ, resulting in considerably different dust direct radiative effects and direct radiative effect efficiencies. Therefore, we state the difference in the dust DRE and DRE efficiency estimate in Sect. 6, but do not document the comparison of dust loadings/deposition/DOD between the bulk and speciated dust simulations."*

Given that DOD is tuned to be similar in these simulations, I do not see why the differences in optical properties should be used as an excuse not to compare BULK with MINE. This would be a very interesting study in its own right, and possibly the authors should omit MINE simulations in this paper as without comparing BULK with MINE, it is difficult to understand why MINE is used at all. Is the additional mineralogical detail in MINE useful for a better dust simulation? What is the additional computational expense of MINE over BULK? Is MINE being considered for inclusion in a future of CAM or is this rather an interesting pedagogical study? Currently, MINE is frivolously used in this study and is unnecessary without further analysis and comparison.

We did not compare the modeled dust cycle between BULK and MINE runs because this is a science with secondary importance. We show the reason in the "Experiment design" section: "With the dust tuning applied toward the similar global mean DOD of ~0.030, the modeled dust cycle (i.e., burdens, concentrations, loadings, and deposition fluxes) would be similar between the bulk- and speciated-dust models that are nudged toward identical offline dynamics and using the same dust size distribution (see Sect. 6). The quantified effect of each of the modifications would thus be similar if using the bulk dust model instead (Fig. S2),…"

The different optical properties are not the reason for not making the comparison but for including the MINE runs. We have shown in the text evaluations on the model performance of modeling the DRE efficiency and the influence of each modification on the DRE estimate, for which modeling the optical properties as accurately as possible is crucial. Therefore, we prefer to use the dust speciated model to quantify such influence, as it simulates spatially varying dust optical properties (the bulk dust model uses a globally constant dust optic).

We had tried to do this but found that having the potential impacts on the mineralogy by changing to the new dust emission scheme is not enough for a separate paper. Instead, we added more analysis on documenting results from the MINE runs such that it makes more sense to have both BULK and MINE runs in this article.

"*6. Bulk versus speciated-dust model*

*The bulk (CAM6.α) and dust-speciated models (CAM6.α_MINE) simulate a similar dust cycle with the difference between the two types of models orders of magnitude smaller than the dust cycle itself modeled either by CAM6.α or CAM6.α_MIN (e.g., Fig. 12 and 13:* R2 *and* R3 *in this document, respectively). This similarity results from several factors.*

*1) tuning the dust cycle to a global mean DOD of 0.03;*

*2) nudging both models towards the same meteorology dynamics;*

*and 3) conserving the dust mass when speciating the dust-aerosols such that summing the mass fraction of each dust species equals unity. For the same reasons, the influence of each of the modifications on the modelled dust cycle quantified using the bulk model instead of the dust-speciated model, as this study used, would be similarly comparable.*

*What differs remarkably is the modeled dust optical properties between the speciated- and bulk-dust simulations. For example, the speciated-dust model (CAM6.α_MIN) yields a lower global-mean dust SSA than the bulk-dust model (CAM6.α): 0.896 versus 0.911 (Table 6) at the visible band centered at 0.53 μm. Note that the dust DRE is sensitive to variation of the dust SSA. This lower dust SSA obtained here in the dust speciated model than in the bulk dust model is consistent with the finding of a previous study (Scanza et al., 2015) using an earlier model version (CAM5). Correspondingly, CAM6.α_MINE yields a reduced dust cooling (Table 6) and DRE efficiency (Fig. 9) relative to CAM6.α.*

*For dust DRE efficiency (Fig. 9), speciating dust in CAM6 tends to reduce the RMSE while retaining the horizontal spatial correlation in either SW (CAM6.α: RMSE=11 W $m^{-2}$ $\tau^{-1}$; R=0.26 versus CAM6.α_MINE: RMSE=10 W $m^{-2}$ $\tau^{-1}$; R=0.20) or longwave (CAM6.α: RMSE=4.0 W $m^{-2}$ $\tau^{-1}$; R=0.86 versus CAM6.α_MINE: RMSE=3.0 W $m^{-2}$ $\tau^{-1}$; R=0.84) or both spectral ranges (CAM6.α: RMSE=7.0 W $m^{-2}$ $\tau^{-1}$; R=0.93 versus CAM6.α_MINE: RMSE=6.0 W $m^{-2}$ $\tau^{-1}$; R=0.92). This comparison suggests that modeling dust as component minerals with the dust size distribution in coarse mode of MINE_NEW_EMIS_SIZE helps improve the model performance relative to modeling dust as a bulk to reproduce the retrieved dust DRE efficiency (Fig. 9a).*

*The improvement in reproducing the retrieved dust DRE efficiency, however, could be*
*artificial because of the combined use of the imaginary part of the complex refractive*
*index of hematite and the volume mixing rule used in the dust speciated model to*
*compute the bulk-dust complex refractive index (Li et al. in prep.). This combination*
*could lead to more absorptive dust than the bulk dust model (Fig. 9a and Table 6)."*
**Specific comments**
[L75] Is it worth introducing the DEAD and BRIFT acronyms here?
Introduced here.
[L84] The fine mode is described as d < 1um whilst the coarse mode is d > 5um.
Normally, the coarse mode is adjacent to the fine mode so I wonder what the authors
would define the intermediate aerosol (1 < d< 5um) as?
We followed the definition used in the community. So, it is not a definition for the coarse
mode aerosol. We revised this statement to avoid possible confusion.
"…*and slightly underestimating that of aerosols with diameter > 5.0µm*…".
[L91] *"one of the changes from CAM5 to CAM6.1 was replacing the size distribution*
*of aerosols in the coarse mode in CAM5 with the one that has a much narrower width*
*in CAM6.1"*- this seems nonsensical to me, or completely without consideration for
actual coarse mode dust widths (e.g., Ryder et al, 2013, 2018, 2019 suggest σ ∈ [1.6,
2] rather than 1.2). Why was it decided to favour stratospheric sulfate over
tropospheric mineral dust when sulfate is more episodic (e.g. volcanic eruptions) and
has less of an impact over tropospheric climate? Also, the authors seem to
recommend that the coarse mode width be reverted to 1.8 as in CAM5 (I agree), but
do not comment on the impact of resetting the coarse mode width on stratospheric
sulfate. Seeing as this was the initial motivation for contracting σ, I think that some
comment is appropriate.
That is right. We also think 1.2 is too narrow to represent the size distribution of dust
aerosol. So, we decided to revert it to 1.8, with which this reviewer also agrees and
recommends using this broad-size distribution in future versions of CAM. In CAM6,
the volcanic sulfate is presented together with dust aerosol. The developers focused
on the volcanic sulfate while advancing the CAM model without noticing that the
employed sigma is inappropriate for dust aerosol.
We commented a little bit on this as below.
*"Our analysis suggests reverting the geometric standard deviation of the transported*
*dust size distribution (coarse mode) from the default 1.2 to 1.8 imposes the biggest*

*change to the modeled dust cycle among what we introduced to CAM6.1. Note that*
*the linear assumption between DOD and the other dust quantities based on which we*
*rescaled up the concentrations, deposition, burdens, and DRE of dust in the size*
*distribution simulations introduces uncertainty. Since the defaulted 1.2 is too narrow*
*to simulate the dust lifetime, in the next released model version, we recommend*
*reverting the geometric standard deviation to 1.8, as in CAM5. This reverse may*
*require a splitting of representation of dust and the stratospheric aerosols in the coarse*
*mode, for which the narrow coarse-mode size distribution works better (Mills et al.,*
*2016), and some changes to sea salt.*"

[Table 1] I think that GMD should be labelled as "initialisation GMD" _as this is more descriptive. Or is the initial GMD at source calculated online? It is difficult to tell from the text what the initial GMS is. This also refers to L179.

Changed to "initialization GMD" here and where it is applicable. The reviewer is right that this is initialization GMD.

[Table 1] Why is the order of the modes Accumulation, Aitken, Coarse, then Primary? Surely it should be in ascending size order: Primary, Aitken, Accumulation then Coarse

The order in Table 1 is the same as that in the model. In response, we reordered the list following the reviewer's suggestion.

**Table 1.** Mode parameters for the Modal Aerosol Module version 4 (MAM4) used in CAM5 (CAM5 size) and CAM6.1 (CAM6 size) by default: geometric standard deviations (σ) and initialization geometric mean diameter (GMD) and its ranges. Values in parentheses if present are for CAM6.1 cells without parentheses are kept the same between CAM5 and CAM6.1."

| Mode (note order) | σ | *Initialization GMD (μm)* | Lower bound GMD (μm) | Upper bound GMD (μm) |
|---|---|---|---|---|
| Primary carbon (a4) | 1.6 | 0.050 | 0.010 | 0.10 |
| Aitken (a2) | 1.6 | 0.026 | 0.0087 | 0.052 |
| Accumulation (a1) | 1.8(1.6) | 0.11 | 0.054 | 0.44 |
| Coarse (a3) | 1.8(1.2) | 2.0(0.90) | 1.0(0.40) | 4.0(40) |

[Table 1] Why was the accumulation mode width changed in CAM6.1? What are the impacts of reverting it? I can't see this detail in the text

Good point. It is again to accommodate the stratospheric aerosol (Mills et al., 2016). Our test simulations suggest negligible impacts on the dust cycle modeling when reverting it. We briefly mentioned this in the revised manuscript (see the "Experiment design" section).

*"The other changes to the width of the accumulation mode and the bounds of the simulated GMD online impose negligible impacts on the dust cycle modeling, thus, we did not construct sensitivity tests on reverting them in this study."*

[L109] The term 'semi-observation' is undefined and is confusing

We now specify both the observation and semi-observation as "measurements, retrievals, and model-observation integration" which should bracket all the data used in this work.

[L115] "show the final summarization in Section 7". This is an unusual way to say "Discussion and conclusions are provided in Section 7" or something to that effect

We changed it to:

*"…limitations in the model-observation comparison in Sect. 5, and discussions and conclusions in Sect. 7."*

[L120] This is one of the places in the text where it is unclear as to: (1) whether the impact of dust asphericity on MEE is represented at all, (2) if it is represented then in what way (methods), and (3) which simulations include it?

To avoid confusion, we removed "and optics" here, moved relevant text from the result Section 4.2.3 (5.2.3 in the revised manuscript) to Section 2.4.3, and added the following to the "Experiment design" section:

*"The enhancement of the mass extinction efficiency of aerosol particles by dust asphericity is included in all the simulations since we do not attempt to quantify how this enhancement impacts the simulated dust cycle."*

[L137] Sentence beginning *"We consider the default DEAD scheme"* should explicitly acknowledge that it refers to emissions

Changed "scheme" to "dust emission scheme".

[L143] How confident are the authors in the critical LAI threshold? Should this assumption be discussed in the Discussion section?

The calculation of the critical LAI threshold has been a standard for a while in CAM of different versions. It could be subject to change in the future. But the associated uncertainty would probably be small compared to the missing pieces we mentioned in the discussion section for modeling dust aerosols in CAM6.

We added one sentence in response to this good question.

"*This large uncertainty could partially result from the constants used in the parametrizations that affect the dust emission and transport processes, such as the critical LAI threshold, the hygroscopicity of dust, and the prescribed scavenging coefficient, though the default values in the model have been used during the past decade in CAM of different versions.*"

[L152] The mass is distributed as 0.1 %, 1 % and 98.9 % between the Aitken, accumulation, and coarse modes. Surely these ratios should change depending on the assumed coarse mode width?

These values were obtained by applying the brittle fragmentation theory to the broad coarse-mode size distribution that is the same as used in this study. Thus, we can apply it to the proposed new models. But the default CAM6.1 uses the same values while employing a much narrower coarse-mode size distribution, which could be problematic.

[L160] Many dust schemes treat dust as initially insoluble and then permitted to age via coagulation and condensation wherein it becomes soluble and internally mixed (e.g., dust in UKESM1). The authors should comment on their assumption of internally mixing dust, which may artificially enhance dust deposition near source regions? Would you expect similar results if dust is assumed to be insoluble?

Since version 5, CAMs employ the internal mixing assumption within each mode as an option. It is worth pointing out that dust aerosols are not completely internally mixed in MAM4 of CAM5/6: dust aerosols in different modes are externally mixed. But most dust mass is distributed in the coarse mode, which indicates that the assumption made to the coarse-mode dust may be most influential on the dust cycle modeling compared to dust in Aitken and accumulation modes. In this paper, we do not attempt to document how different mixing assumptions affect the dust modeling in CAM6 since all our simulations stick to this assumption. So, we tried to answer this question but did not expand it in the manuscript.

From the view of the dust cycle modeling, we think the importance of dust hygroscopicity and its mixing with other aerosols is regionally dependent. For example, a different assumption of mixing with sea salt for South African dust can greatly change simulated deposition near the source, particularly in the downwind area. But near North Africa, they are not that influential because both cloud fractions and sea salt concentrations are typically low. But how the mixing state of aerosols is crucial for modeling the optical properties and radiative effects.

[L165] The Neale et al (2010) reference is an internal document, which I can't find online. Can the authors please provide a URL for downloading the report, or alternatively, relevant peer-reviewed papers with the same information.

**RESPONSE:** It's a technical note. We put it on GitHub and a link in the manuscript where we cite this reference: https://github.com/L3atm/LLi2022GMD.

[L172] *"The wet deposition rate thus depends on the hygroscopicity of dust (=0.068; Scanza et al., 2015) as CCN/INPs and the prescribed scavenging coefficient (=0.1; Neale et al., 2010), both of which are currently constant with respect to the dust size (and composition for speciated dust) in CAM6.1."* _– _I assume the hygroscopicity of dust will evolve as dust is transported through the atmosphere so I question the use of a single spatially uniform constant for this parameter. The below cloud scavenging coefficient (0.1), if it is in units of s-1, seems 2 orders of magnitude too high (Wang et al., 2010, doi:10.5194/acp-10-5685-2010). Wang et al (2010) for instance, suggest it's somewhere between 10-6 for accumulation mode aerosol and 10-3 for coarse mode aerosol depending on scavenging rate. The authors should comment more on the assumptions made in the model and the implications of those assumptions.

We appreciate the great comment and agree that the dust hygroscopicity would vary from region to region and change during transport due to dust aging. How to better treat the scavenging coefficient could be an excellent future study. The purpose of this paper is to document the changes and how they change the dust cycle modeling. We tend not to spend space commenting on all the parameterizations, such as the oversimplified hygroscopicity of dust in CAM6.1. Still, this comment points out important information for modeling dust aerosol, as it could change the wet deposition rate. So, we very briefly pointed this out in the discussion section:

"*This large uncertainty could probably in part result from the constants used in the parametrizations that affect the dust emission and transport processes, such as the critical LAI threshold, the hygroscopicity of dust, and the prescribed scavenging coefficient, though the default values in the model has been used during the past decade in CAM of different versions.*"

[L180] *"Note that the current default CAM6.1 employs a narrow coarse-mode size distribution but a broad boundary width (high bound minus low bound), likely resulting in the GMD bounds less in effect, compared to that in CAM5".* – _what are the impacts of changing the coarse mode width on sea-salt emissions and sea-salt AOD? Surely this change will impact more than dust alone, which may be confounding other results presented in the study (e.g., the DRE).

This change does affect the emissions and optical depth of sea salt. We had included such impacts but then removed relevant text since this study focuses on dust aerosol.

Documenting sea salt seems somewhat distracts the readers. Following this suggestion, we mentioned sea salt in the last section.

"*This reverse may require a splitting of representation of dust and the stratospheric aerosols in the coarse mode, for which the narrow coarse-mode size distribution works better (Mills et al., 2016), and some changes to sea salt.*"

[L210] "The wet size due to growth of aerosol particles by adsorbing water vapor follows the κ-Kohler theory with a time-invariant hygroscopicity for each aerosol species (Petters and Kreidenwei, 2007)". – is it worth listing these hygroscopicity parameters to aid in the replicability of the simulations?

We archived the model code, which contains the values used for each aerosol species and is publicly available.

[L215] "here and hereafter unless stated otherwise" – this phrase, in parentheses, doesn't seem to apply to anything or make sense

Removed.

[L224] This is another place in the text where the impact of asphericity on the MEE is tantalisingly hinted at without further detail as to whether its on and how its incorporated

We clarified this in Section 2.5 of the revised manuscript as below, so, here we removed "calculated mass extinction efficiency and".

"*The enhancement of the mass extinction efficiency of aerosol particles by dust asphericity is included in all the simulations since we do not attempt to quantify how this enhancement impacts the simulated dust cycle*"

[L276] "In addition, the meteorology field (horizontal wind, air temperature T, and relative humidity) was nudged" – the results will obviously be changed if the model is free running then. For instance, the coarse dust will absorb LW radiation, warming the surface and destabilising the atmosphere. Perhaps this assumption (fixed meteorology) should be discussed in the Discussion section

The reviewer is right. If a free running is constructed, which we will do in the future, the results could be different. We pointed out that the results here are from simulations based on the use of offline dynamics in the first paragraph of the last section. To emphasize this, at some other places in the Discussion section, we mentioned this information again:

*"It is worth noting that the results obtained in this study rely on the models with the offline dynamics, which is subject to change while using the predicted meteorology field online."*

*"…with the offline dynamics, the new model, CAM6.α…"*

[L285] *"Therefore, we state the difference in the dust DRE and DRE efficiency estimate in Sect. 6, but do not document the comparison of dust loadings/deposition/DOD between the bulk and speciated dust simulations."* – Avoiding comparing BULK and MINE seems like a massive oversight and is one of the first things I'd query as a reader. Does speciation between minerals improve the simulation compared to assuming dust as a bulk quantity? Simply saying that as the dust properties are different (of course they will be), this reduces comparability, is a little bit absurd and a bit of a cop out. I think this comparison should be made in a follow-on paper. To be honest, it doesn't seem worth including the MINE simulations if they not appropriately analysed.

The dust speciation helps reproduce the observed DRE efficiency improvements compared to without the speciation, as presented in Section 4.3.1 (5.3.1 in the revised text). For non-optical variable, summing over the eight minerals gives the total dust loadings/deposition/DOD similar results to simulations without the dust speciation. Per the suggestion of the reviewers, we added a new section "*6. Bulk- versus speciated-dust model*" collecting information about the comparison between BULK and MINE results that scattered in the text:

*"For dust DRE efficiency (Fig. 9), speciating dust in CAM6 tends to reduce the RMSE while retaining the horizontal spatial correlation in either SW (CAM6.α: RMSE=11 W $m^{-2}$ $\tau^{-1}$; R=0.26 versus CAM6.α_MINE: RMSE=10 W $m^{-2}$ $\tau^{-1}$; R=0.20) or longwave (CAM6.α: RMSE=4.0 W $m^{-2}$ $\tau^{-1}$; R=0.86 versus CAM6.α_MINE: RMSE=3.0 W $m^{-2}$ $\tau^{-1}$; R=0.84) or both spectral ranges (CAM6.α: RMSE=7.0 W $m^{-2}$ $\tau^{-1}$; R=0.93 versus CAM6.α_MINE: RMSE=6.0 W $m^{-2}$ $\tau^{-1}$; R=0.92). This comparison suggests that modeling dust as component minerals with the dust size distribution in coarse mode of MINE_NEW_EMIS_SIZE helps improve the model performance relative to modeling dust as a bulk to reproduce the retrieved dust DRE efficiency (Fig. 9a)."*

Also, we added RMSE and correlation coefficient in the DRE efficiency plot as shown below.

[Figure]

**Figure 8.** Modelled and observed dust direct radiative effect efficiency in the shortwave (SW) and longwave (LW) spectral ranges under clear conditions at the TOA over the sub-domains (shown in the inserted map and location described below) *in April-June (AMJ), summer (JJA), fall (NDJ), and September (Sep) for the 2000s climate. The radiative effect efficiency is defined as the ratio of the radiative effect to DOD, so has units of W m$^{-2}$ $\tau^{-1}$. Included cases from left are CAM6.1, CAM6.α, MINE_NEW_EMIS_SHAPE, CAM6.α _MINE. The field value/range are from references listed in Table 5. Colored numbers show correlation coefficient (R) and the root mean square error (RMSE) between the model and retrievals in the SW (a) and LW (b) spectral ranges or in both spectral ranges (numbers in parenthesis in Panel a).*"

[L289] Choosing to tune some models to DOD = 0.03 but not others is very peculiar. The authors say *"Dust tuning was not applied to EXP03 and EXP04 (bulk dust simulations), in which the dust emission was identical to EXP02, in order to see how changes in the transported dust size distribution affects the DOD calculation".* – Well surely all of the individual sensitivity simulations (emissions, dry deposition, asphericity) would have benefitted from the same analysis? I guess that some parameters in the emissions and dry deposition algorithm need to tuned in some way (so using DOD might be a reasonable approach) as the parameters have a huge degree of uncertainty, but the asphericity probably did not need changing.

Though not tuning EXP03 and EXZP04, we scaled up DOD and applied the same factor to the other dust quantities, as we stated in the text (the "Experiment design" section). This rescaling makes sense, considering the roughly linear relationship between those variables, though we acknowledge doing so may introduce uncertainty. We pointed this out in the Discussion section.

*"…though the linear assumption between DOD and the other dust quantities based on*
*which we rescaled up the concentrations, deposition, burdens, and DRE of dust in the*
*size distribution simulations."*

In the emission and deposition schemes, we agree that there could maybe exist
uncertainty in some parameters. But we would better not scale the non-tunable
parameters within the dust scheme to match the observational constraint of DOD=0.03,
because the scaling factor exists largely due to the missing sub-grid scale variability
by 100-km grid-scale modeling, not because of the uncertainty of parameters. Tuning
those parameters to match the global constraint just seems like errors compensating
each other. The dust emission scheme in CAM contains a tuning parameter "b", in the
calculation of the threshold gravimetric water content, which can plausibly range from
less than 1 to the inversed clay fraction (can be > 3.0). Sensitivity tests by modifying
this tuning parameter among 0.5, 1.0, and 2.0 suggest that 1.0 is a good value to use
(see Table R1 in this document). We tend not to change non-tunable parameters,
since they are observationally constrained. That explains why we did not modify those
parameters in the new dry deposition scheme. We added the following in the
"Experiment design" section and cited a new supplementary table (Table R1) there:

*"An offline sensitivity test (Table S1) supports the use of unity tuning factor to calculate*
*the threshold gravimetric water content which we employed in the experiments for*
*quantifying influence of each modification (speciated dust simulations listed in Table*
*2)."*

[L289] My other issue with this paragraph is that the tuning is not described in any
detail. Which parameters were tuned and what are their values in the baseline
simulation? How was tuning conducted and why was global-mean DOD chosen as the
target? Simply saying 'tuned the model following Albani et al (2014)' _is not sufficient,
and it would be impossible to replicate these simulations without further detail

We added the following to address this comment.

*"…we tuned the model following Albani et al., (2014) by modifying a namelist variable*
*called soil_erod_factor, such that…"*

*"We prefer to tuning the model to reproduce the global mean DOD of 0.030, because*
*this is currently the best estimate of global dust quantities, compared to the others (i.e.,*
*dust concentrations). It turns out that doing so can also reasonably reproduce the other*
*quantities with no need of a regional tuning. We tuned the dust model by modifying a*
*namelist variable in CAM, called soil_erod_factor."*

[Table 3] This table seems very large, and I'm not sure whether the list of acronyms
should be at then end of the table or in the caption. Would it be better to have 1 table
for each metric?

We split this large table into 3 and list the acronyms in the captions:

"*Table* 3. Observed/retrieved cycle for dust model evaluations including optical depth,
surface mass concentrations, surface deposition fluxes, and wet deposition
percentages. AERONET: Aerosol Robotic Network; MODIS: Moderate Resolution
Imaging Spectroradiometer; AOD: aerosol optical depth; DOD: dust optical depth."

"*Table* 4. Measured/retrieved dust size distribution for model evaluation. AERONET:
Aerosol Robotic Network; DustCOMM: Dust Constraints from joint Observational-
Modelling-experiMental analysis."

"*Table* 5. Retrieved dust radiative effect efficiency for model evaluation. CERES:
Clouds and the Earth's Radiant Energy System; TOA: top of the atmosphere; JJA:
June, July, and August; AOD: aerosol optical depth; MISR: Multi-angle Imaging
SpectroRadiometer; OMI: Ozone Monitoring Instrument; NDJ: November, December,
and January; MODIS: Moderate Resolution Imaging Spectroradiometer; CALIPSO:
Cloud-Aerosol Lidar and Infrared Pathfinder Satellite Observations; MFRSR:
MultiFilter Rotating Shadowband Radiometer; SEVIRI: Spinning Enhanced Visible and
Infrared Imager; GERB: Geostationary Earth Radiation Budget; AERONET: Aerosol
Robotic Network; MPL: Micro-Pulse Lidar; AERI: Atmospheric Emitted Radiance
Interferometer; SMART: Surface-sensing Measurements for Atmospheric Radiative
Transfer; AMJ: April, May, and June."

[Results] The difference between CAM6.1 and CAM6.α i.e., the control and the
simulation with all changes added (except mineralogy) comes before the dissection of
impacts of individual processes. Why is this? Surely it would be better to investigate
the impacts of the individual processes and then use them to explain why CAM6. α is
different to CAM6.1?

We think doing what the reviewer suggested or keeping it as it was would be fine. In
the drafted manuscript, we have tried doing the same as the reviewer suggested but
reordered the result section taking the "principle" that "the most important things go
first" since the manuscript is lengthy. In any order, the conclusions of this article would
remain unchanged.

[L378] "CAM6.1 may overestimate the contribution of high-latitude dust emissions to
the global dust total (8.0%)." – is this referring to the dust burden? It's rather
ambiguous as is

This refers to the dust emission. We modified this sentence a little bit.

"CAM6.1 may overestimate the contribution of the high-latitude dust emission to the global dust total *emission* (8.0%)."

[L391] *"Overall, all models reproduced the climatology of DOD from AERONET retrievals, the surface concentration, and deposition within a factor of ten (Fig. 1 and Fig. S3)"* – this doesn't seem to be the case from looking at Fig. 1 b, c, e, f, h, and i. It seems that both models exhibit at least one measurement outside the range of 1/10x and 10x.

There are 1, 4, and <10 point(s) of the 36, 47, and 108 points for DOD, surface concentrations, and deposition outside that range. That is, over 90% of the points fall in the factor of 10. To be more accurate, we modified the sentence.

*"Over 90% of the measurement sites,* all models reproduced the climatology of DOD from AERONET retrievals, the surface concentration, and deposition within a factor of ten (Fig. 1 and Fig. S3*)"*

[Fig. 2] Why is the new dust emissions scheme smoother in terms of emissions, rather than the delta function (almost) in DEAD? I couldn't easily find this information in the text

We added the following to answer this question.

"*The smoother distribution of the dust emission in BRIFT than DEAD is due primarily to the use of the source function in DEAD that shifts dust emissions toward the most erodible soil, while in BRIFT, the near-surface friction velocity frequently exceeds the calculated threshold wind fraction velocity, causing dust to emit at more grid cells.*"

[Fig. 3] Isn't the Ridley et al (2016) DOD dataset constrained by MODIS (either through assimilation or using it as a baseline? If so, aren't Figs 3a and 3b effectively showing the same results?

Good point. DOD of Ridley et al. (2006) "assimilated" MODIS retrievals: the authors corrected the bias present in MODIS retrievals (see Section 3 in the manuscript). So, their DOD reflects the information of pure MODIS DOD, but the two datasets show considerably different results. For example, the globally averaged DOD from pure MODIS post-processed by Pu et al. (2020) is significantly higher than the best estimate of Ridley et al. (2016) (0.025-0.035). Please check the references in the main text.

[L436] capture -> captures

Corrected.

[L437] Taklamakan (as in the desert) is spelt wrong throughout

Corrected.

[Fig. 4] Great figure

Thanks!

[L498] S5i -> S5e

Corrected.

[Fig. 5] This plot, especially Fig. 5a, is very confusing. There are too many colours and
it is difficult to pick out the CAM models. It may be worth plotting a non-CAM multi-
model mean with max/min as shaded in grey, and then have just the CAM models in
colour

We removed non-CAM model results and cited relevant references instead.

[Figure]

**Figure 5.** Normalized size distribution of dust between 0.2 and 10 μm diameter in the
global average (a), near Canary Island (blue colors in b; dot: 2.5 km; x: 6-7 km; data
for June/July 1997 from Otto et al., 2007), and near Cabo Verde (orange colors in c;
dot: 2.5 km; x: 6-7 km; data for August 2015 taken from Ryder et al., 2018). The default model, CAM6.1: (purple line); the new model, CAM6.α: (red line); semi-observations: DustCOMM (black line) *inverted based on an integration of a global model ensemble and quality-controlled observational constrains on the transported dust size distribution, extinction efficiency, and regional DOD* with data taken from Adebiyi et al. (2020). We chose the model layers and grid cells that are closest to the location and atmospheric height, as well as the months, where and when the measurements were made for comparison."

[L542] Why is the size distribution for the fine dust fraction better captured by CAM6.α?

We explained this in the revised manuscript.

"*But it greatly underestimated the fine dust fraction (diameter < 2 μm) which CAM6.α can better capture due primarily to the more correct gravitational settling velocity modeled by using the new dry deposition scheme.*"

[L548] Sentence beginning "Overall, CAM6.α better reproduced the size distribution". It would be worth adding the caveat here that the Otto et al and Ryder et al measurements are from single campaigns or flights and thus may not reflect the long-term mean dust properties at those altitudes, locations, and times

Good point. We introduced a separate section listing limitations common among the model-data comparison, including this point. But it's good to mention again at this place. So, we added the following.

"*It is worth noting that the measurements are from single campaigns or flights that may have representative issues not reflecting the climatological size and vertical distributions of dust aerosols (i.e., limited by the space and time coverage).*"

[L558] Section 4.2.1 – why are the mineralogy experiments used to test BRIFT vs DEAD rather than the BULK simulations? There doesn't appear to be any reasoning behind this

We planned to have BULK and MINE runs for two separate papers but put them together into this article. The comparison of BRIFT with DEAD in the BULK runs would be similar to those in the MINE runs. Please see our response to the general comment by this reviewer on BULK versus MINE (Line 1404-1558).

[L559] MIINE_NEW_EMIS -> MINE_NEW_EMIS

Done.

[L646] Paragraph on asphericity – I'm still confused even after reading the text as to whether the assumption of asphericity is applied to the dust MEE in every simulation run here or just the MINE_NEW_EMIS_SHAPE simulation?

We paste our response to the previous comments here:

To avoid confusion, we removed "and optics" here, moved relevant text from the result Section 4.2.3 (5.2.3 in the revised manuscript) to Section 2.4.3, and added the following to the "Experiment design" section:

"*The enhancement of the mass extinction efficiency of aerosol particles by dust asphericity is included in all the simulations since we do not attempt to quantify how this enhancement impacts the simulated dust cycle*."

[L683] "(0.030-0.019)/0.030*100)" – I don't think this formula needs to be written. See also L686 and L759

Deleted.

[L693] Paragraph beginning *"The lifetime of dust"*. Should this paragraph be in Section 4.2.4? It doesn't seem to mention asphericity or apply to the MINE_NEW_EMIS_SHAPE simulation

Since this is a comparison between BRIFT and DEAD, we moved to Section 5.2.1 (revised manuscript) "Dust emission schemes: BRIFT versus DEAD".

[L705] Why is MINE_NEW_EMIS referred to as the reference case? It's a sensitivity simulation, isn't it? Surely the only reference cases are CAM6.1 and possibly MINE_BASE?

It is not a sensitivity simulation. By default, CAM6.1 uses an incorrect coarse-mode size distribution of dust. Thus, it does not make sense to use CAM6.1 as the baseline simulation when quantifying the impact of each of the modifications. Please see our response to the general comment by this reviewer on the experiment design (Line 1404-1508).

[L733] *"NEW_EMIS_SIZE"* -> MINE_NEW_EMIS_SIZE. Also, this paragraph seems to be the only place where BULK and MINE are explicitly compared. I think the comparison should extend to all the dust metrics

As stated in our responses to previous comments, with the dust tuning and offline dynamics applied, speciating dust does not yield considerably different dust quantities (i.e., dust concentrations, burdens, and deposition) from BULK runs in the current climate. We added a new section to compare BULK and MINE runs:

**"6. Bulk- versus speciated-dust model**

*The bulk (CAM6.α) and dust-speciated models (CAM6.α_MINE) simulate a similar dust cycle with the difference between the two types of models orders of magnitude smaller than the dust cycle itself modeled either by CAM6.α or CAM6.α_MIN (e.g., Fig. 12 and 13: R2 and R3 in this document, respectively). This similarity results from several factors.*

*1) tuning the dust cycle to a global mean DOD of 0.03;*

*2) nudging both models towards the same meteorology dynamics;*

*and 3) conserving the dust mass when speciating the dust-aerosols such that summing the mass fraction of each dust species equals unity. For the same reasons, the influence of each of the modifications on the modelled dust cycle quantified using the bulk model instead of the dust-speciated model, as this study used, would be similarly comparable.*

*What differs remarkably is the modeled dust optical properties between the speciated- and bulk-dust simulations. For example, the speciated-dust model (CAM6.α_MIN) yields a lower global-mean dust SSA than the bulk-dust model (CAM6.α): 0.896 versus 0.911 (Table 6) at the visible band centered at 0.53 μm. Note that the dust DRE is sensitive to variation of the dust SSA. This lower dust SSA obtained here in the dust speciated model than in the bulk dust model is consistent with the finding of a previous study (Scanza et al., 2015) using an earlier model version (CAM5). Correspondingly, CAM6.α_MINE yields a reduced dust cooling (Table 6) and DRE efficiency (Fig. 9) relative to CAM6.α.*

*For dust DRE efficiency (Fig. 9), speciating dust in CAM6 tends to reduce the RMSE while retaining the horizontal spatial correlation in either SW (CAM6.α: RMSE=11 W $m^{-2}$ $\tau^{-1}$; R=0.26 versus CAM6.α_MINE: RMSE=10 W $m^{-2}$ $\tau^{-1}$; R=0.20) or longwave (CAM6.α: RMSE=4.0 W $m^{-2}$ $\tau^{-1}$; R=0.86 versus CAM6.α_MINE: RMSE=3.0 W $m^{-2}$ $\tau^{-1}$; R=0.84) or both spectral ranges (CAM6.α: RMSE=7.0 W $m^{-2}$ $\tau^{-1}$; R=0.93 versus CAM6.α_MINE: RMSE=6.0 W $m^{-2}$ $\tau^{-1}$; R=0.92). This comparison suggests that modeling dust as component minerals with the dust size distribution in coarse mode of MINE_NEW_EMIS_SIZE helps improve the model performance relative to modeling dust as a bulk to reproduce the retrieved dust DRE efficiency (Fig. 9a).*

*The improvement in reproducing the retrieved dust DRE efficiency, however, could be artificial because of the combined use of the imaginary part of the complex refractive index of hematite and the volume mixing rule used in the dust speciated model to compute the bulk-dust complex refractive index (Li et al. in prep.). This combination could lead to more absorptive dust than the bulk dust model (Fig. 9a and Table 6)."*

[L798] "Overall, replacing the size distribution of dust aerosol and the dust emission
scheme with new ones (PZ10 and BRIFT, respectively)" – replacing the size
distribution is referred to here as PZ10 but this is the dry deposition scheme

"(PZ10 and BRIFT, respectively)" removed.

[L821] The term "space volume" is ambiguous. Possibly "colocation in space"?

Changed.

[L833] *"which can get mixed with dust aerosol particles during the transport and may*
*not be completely excluded in the measurements."* _This seems a little lazy, do you
have any estimates of how much contaminations leads to errors in measuring dust?
At the moment, this point isn't backed up by evidence.

We deleted these (please see revised sentence below), because 1) the second half of
the sentence reads more like a repeat of the first half, which the references we had
cited serve well to support, and 2) they do not convey vital elements (we compiled the
dust measurements from previous publications using them here to evaluate the model
performance. This section discussed the limitation in the model-observation
comparison as a notice to readers that such error exists in the dust measurements.).

"Finally, the modelled dust mass is for dust with our own defined mineralogy
composition only (Li et al., 2021; Scanza et al., 2015), the measured mass could likely
also include non-dust particles, such as sea salt (Kandler et al., 2011; Zhang et al.,
2006), sulfate (Kandler et al., 2007), biomass burning aerosols (Ansmann et al., 2011;
Johnson et al., 2008), or other air pollution aerosol (Huang et al., 2010; Yuan et al.,
2008)."

[L859] "… followed by the enhanced dust mass extinction efficiency at the visible band
by ~30% to account for the enhancement by dust asphericity" – the asphericity applied
to the MEE has not been shown to be the second most important change affected.
Rather Fig. 10 shows that asphericity has a negligible impact on dust. Or is the
asphericity in the MEE applied separately to the asphericity in the deposition rate?
This is very confusing.

We do not plan to estimate the optical effect of the dust asphericity in this study. That
is why we include such an effect in all the simulations. In response, we removed text
relevant to the optical effect of the dust asphericity in the conclusion section. We also
clarified how we dealt with the enhanced dust mass extinction efficiency at the visible
band in the simulations in the "Experiment design" section:

**"*The enhancement of the mass extinction efficiency of aerosol particles by dust
aspheviticy is included in all the simulations since we do not attempt to quantify how
this enhancement impacts the simulated dust cycle.*"**
[L869] "Overall, the new model can:" – is the new model, referred to in this sentence,
CAM6.α? If so, why has CAM6.α_MINE been neglected? The addition of MINE to this
study makes little sense as it is peripheral. Additionally, is this "new model" _already
adopted for the next revision of CAM6 or is this the plan for the future?
We specified the new model. CAM6.α_MINE and CAM6.α show almost identical dust
cycles. Please see our response to the general comment by this reviewer (Line 1404-
1558). The modifications made to CAM6.1 to get CAM6.α is on the table. But the dust
speciation is not planned yet to be included in a future CAM version.